



# TOSCA - An Open-Source Finite-Volume LES Environment for Wind Farm Flows

Sebastiano Stipa[1], Arjun Ajay[1], Dries Allaerts[2], and Joshua Brinkerhoff[1]

[1]University of British Columbia, Okanagan Campus, CA
[2]Delft University of Technology, NL

**Correspondence:** Sebastiano Stipa (sebstipa@mail.ubc.ca)

**Abstract.** The growing number and size of wind energy projects coupled with the rapid growth in high-performance computing technology are driving researchers toward conducting large-scale simulations of the flow field surrounding entire wind farms. This requires highly parallel-efficient tools, given the large number of degrees of freedom involved in such simulations, and yields valuable insights on farm-scale physical phenomena, such as gravity wave interaction with the wind farm and farm-farm

wake interactions. In the current study, we introduce the open-source, finite-volume, large eddy simulation (LES) code TOSCA (Toolbox fOr Stratified Convective Atmospheres), and demonstrate its capabilities by simulating the flow around a finite-size wind farm immersed in a shallow, conventionally neutral boundary layer (CNBL), ultimately assessing gravity wave-induced blockage effects. Turbulent inflow conditions are generated using a new hybrid off-line/concurrent precursor method. Velocity is forced with a novel pressure controller that allows to prescribe a desired average hub-height wind speed while avoiding

inertial oscillations above the atmospheric boundary layer (ABL) caused by the Coriolis force, a known problem in wind farm LES studies. Moreover, to correct the dependency of the potential temperature profile evolution on the code architecture observed in previous studies, we propose a method to maintain the mean potential temperature profile constant throughout the precursor simulation. Furthermore, we highlight that different codes do not predict the same velocity inside the boundary layer under geostrophic forcing, owing to their intrinsically different numerical dissipation. The proposed methodology overcomes

these issues by ensuring that the same hub wind and thermal stratification are used in successor simulations, regardless of the adopted code or precursor run time. Finally, validation of actuator line and disk models, CNBL evolution, and velocity profiles inside a periodic wind farm are also presented to assess TOSCA's ability to model large-scale wind farm flows accurately and with high parallel efficiency.

## 1 Introduction

In 2018, Ørsted, a leading company in developing, constructing, and operating offshore and onshore wind farms, concluded a project aimed at understanding the limits of models and processes used for wind energy forecasts. The investigation pointed out that blockage and wake effects are currently neglected and underestimated respectively when performing wind power predictions (Ørsted, 2019). Blockage, also referred to as turbine/farm induction (Bleeg et al., 2018), is defined as the wind slowdown approaching the wind farm. On the other hand, wake losses are characterized by a power production deficit by



waked turbines, and are claimed to be underestimated both inside and especially between neighboring sites (Pedersen et al., 2022). While wind farm losses arising from individual turbine wakes have been the subject of extensive research, farm-farm wake effects gained importance only recently (Ahsbahs et al., 2020; Schneemann et al., 2020). Specifically, as more plants are constructed in the proximity of pre-existing ones, the evolution of neighboring farm wakes is an increasingly important aspect to account for and model (Nygaard et al., 2020).

Turbine-level induction has been researched for many years (Troldborg and Meyer Forsting, 2017; Gribben and Hawkes, 2019; Branlard and Gaunaa, 2014; Branlard et al., 2020; Branlard and Meyer Forsting, 2020; Segalini, 2021; Segalini and Dahlberg, 2020), and extensions to wind turbine clusters have been attempted using a linear superposition of individual effects. However, recent studies suggest that this could underestimate — if not totally misrepresent — wind farm-level blockage, which is heavily influenced by the mutual interaction between the wind farm and the density-stratified atmospheric boundary

layer (ABL) (Smith, 2010; Wu and Porté-Agel, 2017; Allaerts and Meyers, 2017, 2018, 2019; Centurelli et al., 2021). In fact, the flow deceleration in the wind farm displaces the capping inversion layer, and interfacial waves are formed. Subsequently, their energy is transported vertically and horizontally by atmospheric internal gravity waves. This mechanism triggers pressure disturbances inside the boundary layer, altering the velocity field around the wind farm.

In industry, wind energy predictions are made using low-cost but fast, often analytical, reduced-order models (Katić et al.,
1987; Ainslie, 1988; Larsen, 1988; Anderson, 2009; Openwind; Bastankhah and Porté-Agel, 2014; Niayifar and Porté-Agel, 2016), aimed at capturing the gross aerodynamic processes within the farm. While they have been used effectively for hundreds of wind energy projects, the majority of these models currently struggle in accurately reproducing wind farm blockage and farm-farm wake interactions. This is classified as an industry-wide issue, as over-predicting annual energy production can have a negative impact on all companies' financial estimates.

Reduced-order models need to be thoroughly validated, but comprehensive observation datasets are difficult to obtain. Numerical analyses, in particular large eddy simulations (LES), are able to provide such data, together with valuable insight into the physical processes. LES resolves the largest and most energetic turbulent eddies, while the smallest ones are modeled. Nevertheless, LES of large wind farms in the atmospheric boundary layer (ABL) is extremely challenging, given the breadth of scales involved, spanning from resolved turbulence eddies of a few meters, to gravity waves characterized by wavelengths
of several kilometers. In addition, many numerical aspects have to be carefully treated, such as wave reflections produced by the domain boundaries, and realistic inflow turbulence generation.

In order to efficiently and accurately simulate the flow around a finite-size wind farm under thermal stratification, an LES solver must possess good parallel efficiency, an optimized code input-output (I/O), and a method for defining initial and boundary conditions that accurately reflect the spatio-temporal ABL state. This includes realistic inflow turbulence modeling, and
a system to avoid gravity wave reflections at the physical boundaries. These two tasks can be achieved at once using the concurrent-precursor method, where a simulation without wind turbines (precursor) is advanced in sync with the wind farm simulation (successor). The latter features a fringe region, where body forces are used to damp gravity waves reflections and to restore the desired turbulent inflow. At each time step, such body forces are calculated based on the concurrent precursor instantaneous fields, leading to the precursor and successor solutions matching at the fringe region exit. More details on pre-





cursor techniques are given in Sec. 2.6, where our new hybrid method is also described.

Several LES codes have been developed by the research community so far (see Breton et al., 2017 for a review), among which only a few can effectively tackle the above-mentioned application. The KU-Leuven code SP-Wind, for example, has been successfully used for finite wind farm simulations capturing gravity wave effects (Lanzilao and Meyers, 2022b), but unfortunately is not open-source. Conversely, open-source tools, such as the PALM model (Maronga et al., 2015), developed by the Institute

of Meteorology and Climatology at Leibniz Universität of Hannover (Germany), or SOWFA (the Simulator fOr Wind Farm Applications), maintained by the National Renewable Energy Laboratory (NREL), do not implement the concurrent-precursor method, making it difficult to properly capture gravity waves effects and inlet/outlet reflections. In addition, although SOWFA has been used in several research studies in the last decade (Fleming et al., 2014; Matiz-Chicacausa and Lopez, 2018; Johlas et al., 2021; Zhang et al., 2022, to name a few), it is not sufficiently parallel-efficient when running with thousands of cores,

and the number of produced files increases with processor count. While some of these shortcomings have been addressed and solved in the NREL Exawind project (Min et al., 2022), the latter is not yet at a production stage.

For the aforementioned reasons, we have developed an open-source, finite-volume framework that is tailored for large-scale studies of wind farm-induced gravity waves, which would address at least two of the grand challenges facing wind energy in the present decade (Sprague et al., 2017), namely *simulating boundary layer turbulence over large areas* and the *simula-*

*tion of an entire wind farm under realistic atmospheric flow conditions*. The new framework is called TOSCA (Toolbox fOr Stratified Convective Atmospheres) and exploits state-of-the-art parallel libraries, such as OpenMPI (Gabriel et al., 2004), PETSc (Balay et al., 2022a), HYPRE (Falgout and Yang, 2002) and HDF5 (The HDF Group, 2000-2010) for the parallel solution of partial differential equations and handling of intense I/O operations. TOSCA is designed to enable LES simulations of large finite wind farms, operating in realistic atmospheric conditions, modeling wind turbines with mid-fidelity

actuator models such as the actuator line (Sørensen and Shen, 2002) and the actuator disk (Jimenez et al., 2007, 2008). As inlet-outlet boundary conditions produce a consistent and undesirable reflection of atmospheric gravity waves, we introduce a hybrid off-line precursor/concurrent-precursor methodology which, coupled with periodic boundary conditions, avoids artificial wave reflections while simultaneously reducing the computational cost associated with initializing the turbulent precursor. The concurrent-precursor method (Inoue et al., 2014) is to our knowledge not available in other finite-volume solvers, though

extensively used in pseudo-spectral methods (Wu and Porté-Agel, 2017; Allaerts and Meyers, 2017, 2018). For this reason, gravity waves studies to-date have been only performed using the latter discretization technique, which does not allow for grid refinement in the pseudo-spectral directions. This forces a uniform grid resolution, leading to high cell counts. Conversely, the finite-volume method allows for grid stretching, enabling to resolve larger domains with the same number of degrees of freedom while providing greater geometrical flexibility.

The present paper is organized as follows. First, in Sec. 2, we describe the developed LES framework. Next, Sec. 3 presents comparisons with existing numerical and experimental studies to validate TOSCA's actuator models, the evolution of thermally stratified ABLs, and wake interactions inside a periodic wind farm in neutral conditions. In Sec. 4, we compare results obtained from CNBL simulations using the newly developed velocity and temperature controlling techniques against the commonly used geostrophic forcing combined with a wind angle controller. In Sec. 5, we present the simulated flow field around a reference



100-turbine finite wind farm immersed in a turbulent CNBL, highlighting TOSCA's ability to accurately capture gravity wave
blockage effects. Finally, conclusions are outlined in Sec. 6.

## 2    Methodology

TOSCA is written in C, and it largely exploits the OpenMPI (Gabriel et al., 2004), PETSc (Balay et al., 2022a,b, 1997), HYPRE
(Falgout and Yang, 2002) and HDF5 (The HDF Group, 2000-2010) libraries to handle I/O operations in parallel and to solve
linear and non-linear systems arising after discretizing the governing equations. TOSCA is a finite-volume code, formulated in
generalized curvilinear coordinates, allowing it to take as input also non-Cartesian structured meshes. The governing equations
are described in Sec. 2.1, while the numerical method is presented in Sec. 2.2, together with a brief overview of generalized
curvilinear coordinates. Sec. 2.3 details the LES turbulence model. Wind turbines are represented using actuators models, de-
scribed in Sec. 2.4, but can also be fully resolved, together with complex terrain, through a sharp-interface immersed boundary
method (IBM) based on Haji Mohammadi et al. (2019), to be detailed in a follow-up paper.

Details on TOSCA's parallel efficiency are reported in Appendix B, where we analyze the time per iteration with increasing
number of Niagara (Loken et al., 2010; Ponce et al., 2019) nodes and mesh elements. TOSCA has also been used to run finite
wind farm simulations on the whole Niagara supercomputer (2024 nodes, 40 cores per node) and on all Cascade nodes of the
UBC-ARC Sockeye supercomputer, demonstrating its capability to handle massively-parallel computations.
In order to run ABL simulations, we developed a novel methodology, described in Sec. 2.5, that enforces a desired hub-
height wind speed while simultaneously avoiding inertial oscillations produced by the Coriolis force above the boundary layer.
In addition, we show that disagreement exists between different CFD codes in predicting the final mean potential temperature
profile inside the boundary layer. In this regard, we propose the use of a mean temperature controller which maintains a
prescribed average potential temperature profile, harmonizing the comparison of simulation results in future studies. Finally,
Sec. 2.6 details our hybrid off-line/concurrent precursor methodology, which saves computational resources when performing
the turbulence initialization in the precursor phase.

### 2.1    Governing Equations

Governing equations correspond to mass and momentum conservation for an incompressible flow with Coriolis forces and
Boussinesq approximation for the buoyancy term. The latter is calculated using the modified density $\rho_k$, evaluated by solving
a transport equation for the potential temperature. These equations, expressed in Cartesian coordinates using tensor notation
read

$$\frac{\partial u_i}{\partial x_i} = 0 \tag{1}$$

$$\frac{\partial u_i}{\partial t} + \frac{\partial}{\partial x_j}\left(u_j u_i\right) = -\frac{1}{\rho_0}\frac{\partial p}{\partial x_i} + \frac{\partial}{\partial x_j}\left[\nu_{\text{eff}}\left(\frac{\partial u_i}{\partial x_j} + \frac{\partial u_j}{\partial x_i}\right)\right] - \frac{1}{\rho_0}\frac{\partial p_\infty}{\partial x_i} + \frac{\rho_k}{\rho_0}g_i - 2\epsilon_{ijk}\Omega_j u_k + f_i + s_i^v + s_i^h \tag{2}$$

$$\frac{\partial \theta}{\partial t} + \frac{\partial}{\partial x_j}\left(u_j \theta\right) = \frac{\partial}{\partial x_j}\left(k_{\text{eff}}\frac{\partial \theta}{\partial x_j}\right) \tag{3}$$



where $u_i$ is the Cartesian velocity, $p/\rho_0$ is the kinematic pressure, $\theta$ is the potential temperature, defined as $\theta = T(p_0/p)^{R/c_p}$ ($T$ is the absolute temperature, $R$ is the gas specific constant, $c_p$ is the specific heat at constant pressure and $p_0$ is the reference pressure), $g_i$ is the gravitational acceleration vector, $\Omega_j$ is the rotation rate vector at an arbitrary location on the planetary surface (defined as $\omega \cos\phi\widehat{y} + \omega \sin\phi\widehat{z}$, where $\phi$ is the latitude, in a local reference frame having $\widehat{z}$ aligned and opposite to the gravitational acceleration vector, $\widehat{x}$ tangent to Earth's parallels and $\widehat{y}$ such that the frame is right-handed). Source terms $f_i$, $s_i^v$, and $s_i^h$ are body forces introduced by turbines, and by vertical and horizontal damping regions, respectively. Moreover, the modified density $\rho_k$ is defined as

$$\frac{\rho_k}{\rho_0} = 1 - \left(\frac{\theta - \theta_0}{\theta_0}\right) \tag{4}$$

where $\theta_0$ is a reference potential temperature, chosen as the ground temperature. Parameters $\nu_{\text{eff}}$ and $\kappa_{\text{eff}}$ are the effective viscosity and thermal diffusivity respectively. The former is the sum of the kinematic viscosity $\nu$ and the sub-grid scale viscosity $\nu_t$, while the latter is sum between the thermal diffusivity $\kappa = \nu/Pr$ and the turbulent thermal diffusivity $\kappa_t$. Both $\nu_t$ and $\kappa_t$ are defined in Sec. 2.3, while the Prandtl number $Pr$ is set to 0.7 in all simulations. The third term on the right-hand side of Eq. 2 is a uniform horizontal pressure gradient that balances turbulent stresses and the Coriolis force, allowing the boundary layer to reach a statistically steady state. This term is commonly referred to as velocity controller, and it is explained in Sec. 2.5.1.

## 2.2 Numerical Procedure

The adoption of generalized curvilinear coordinates allows for the computational mesh to follow terrain coordinates if required, or to be stretched and deformed with the only condition that the indexing remains structured. We denote a set of generalized curvilinear coordinates as $l_i$, with $i = 1, 2, 3$, by which points in a three-dimensional Euclidean space $E^3$ may be defined. Cartesian coordinates are a special case of such a generalization, and will be denoted as $x_i$, with $i = 1, 2, 3$. When using explicit notation, the three curvilinear directions will be identified by Greek symbols as $\xi$, $\eta$, and $\zeta$. With these definitions, and given the position vector $\mathbf{r}$ of a point P in Cartesian space, the covariant base vectors can be expressed as $\mathbf{g}_i = \partial\mathbf{r}/\partial l_i$ (with Cartesian components $(\mathbf{g}_i)_j = \partial x_j/\partial l_i$), while contravariant base vectors are given by $\mathbf{g}^i = \nabla l_i$ (with Cartesian components $(\mathbf{g}^i)_j = \partial l_i/\partial x_j$). As a result, the following relation holds between covariant and contravariant base vectors

$$\mathbf{g}^i = \frac{\mathbf{g}_{i+1} \times \mathbf{g}_{i+2}}{\mathbf{g}_i \cdot (\mathbf{g}_{i+1} \times \mathbf{g}_{i+2})} = J(\mathbf{g}_{i+1} \times \mathbf{g}_{i+2}) \tag{5}$$

where $J$ is the Jacobian of the transformation defining $l_i$ in terms of $x_j$, i.e. the determinant of the matrix of partial derivatives $\partial l_i/\partial x_j$. It is required that $J \neq 0$, which is equivalent to asking that covariant base vectors are not co-planar. Note that they are usually neither unit vectors nor orthogonal to each other. Given a set of curvilinear coordinates $l_i$, with covariant base vectors $\mathbf{g}_i$ and contravariant base vectors $\mathbf{g}^i$, it is possible to define the covariant and contravariant metric tensors through the scalar products

$$g_{ij} = \mathbf{g}_i \cdot \mathbf{g}_j = \frac{\partial x_k}{\partial l_i}\frac{\partial x_k}{\partial l_j}, \qquad g^{ij} = \mathbf{g}^i \cdot \mathbf{g}^j = \frac{\partial l_i}{\partial x_k}\frac{\partial l_j}{\partial x_k} \tag{6}$$

where the repeated index implies summation. Metric tensors satisfy $J = \sqrt{\det(g_{ij})}$ and $J^{-1} = \sqrt{\det(g^{ij})}$.



The use of generalized curvilinear coordinates allows differential operators on any structured mesh to be expressed using a Cartesian-like discretization along the curvilinear directions, which are chosen to be the local structured grid lines. Moreover, the quantities

$$\mathbf{S}^{\xi} = \frac{1}{J}\left(\frac{\partial \xi}{\partial x}\widehat{\mathbf{x}} + \frac{\partial \xi}{\partial y}\widehat{\mathbf{y}} + \frac{\partial \xi}{\partial z}\widehat{\mathbf{z}}\right), \qquad \mathbf{S}^{\eta} = \frac{1}{J}\left(\frac{\partial \eta}{\partial x}\widehat{\mathbf{x}} + \frac{\partial \eta}{\partial y}\widehat{\mathbf{y}} + \frac{\partial \eta}{\partial z}\widehat{\mathbf{z}}\right), \qquad \mathbf{S}^{\zeta} = \frac{1}{J}\left(\frac{\partial \zeta}{\partial x}\widehat{\mathbf{x}} + \frac{\partial \zeta}{\partial y}\widehat{\mathbf{y}} + \frac{\partial \zeta}{\partial z}\widehat{\mathbf{z}}\right) \tag{7}$$

are equal to face area vectors if evaluated at cell faces. Contravariant base vectors components $\partial l_i/\partial x_j$ in Eq. 7 are evaluated using Eq. 5, i.e. from the covariant base vectors, which are easily obtained exploiting finite differences.

Contravariant components of any vector field $\mathbf{u}$, function of position $\mathbf{r}$, can be expressed in terms of its Cartesian components as $u^i = \mathbf{u} \cdot \mathbf{g}^i$. If one instead uses $\mathbf{g}^i/J$ (namely the face area vector along the $i^{\text{th}}$ direction $\mathbf{S}^i$), and the relation is again evaluated at cell faces, contravariant fluxes are obtained as $V^i = \mathbf{u} \cdot \mathbf{S}^i$. In TOSCA, only the independent variables (positions)
are transformed in curvilinear coordinates using the chain rule and integration by parts, while dependent variables are retained in Cartesian coordinates. This partial transformation avoids computing the Christoffel symbols of the second kind, which are cumbersome to evaluate numerically. Moreover, they would increase the requirements of smoothness of the computational mesh, as they involve second-order derivatives of the transformation metrics. The momentum equation is finally dotted with the face area vectors, so that it can be partially written in terms of contravariant fluxes as

$$\frac{\partial V^q}{\partial l_q} = 0, \tag{8}$$

$$\frac{\partial V^q}{\partial t} + \frac{\partial l_q}{\partial x_i}\frac{\partial}{\partial l_r}(V^r u_i) = -\frac{1}{\rho_0}\frac{\partial p}{\partial l_r}g^{rq} + \frac{\partial l_q}{\partial x_i}\frac{\partial}{\partial l_r}\left[\frac{\nu_{eff}}{J}\left(\frac{\partial u_i}{\partial l_k}g^{rk} + \frac{\partial u_j}{\partial l_k}\frac{\partial l_r}{\partial x_j}\frac{\partial l_k}{\partial x_i}\right)\right] - \frac{1}{\rho_0}\frac{\partial p_\infty}{\partial l_r}g^{rq} +$$

$$+ \frac{1}{J}\frac{\partial l_q}{\partial x_i}\frac{\rho_k}{\rho_0}g_i - \frac{2}{J}\frac{\partial l_q}{\partial x_i}\epsilon_{ijk}\Omega_j u_k + \frac{1}{J}\frac{\partial l_q}{\partial x_i}\left(f_i + s_i^v + s_i^h\right), \tag{9}$$

$$\frac{\partial \theta}{\partial t} + \frac{\partial}{\partial l_r}(V^r \theta) = J\frac{\partial}{\partial l_r}\left(\frac{\kappa_{eff}}{J}\frac{\partial \theta}{\partial l_k}g^{rk}\right). \tag{10}$$

Eq. 9 is used to solve for contravariant fluxes, which are staggered at cell faces, while pressure is located at cell centers. In
contrast to a staggered formulation using a full transformation, where Cartesian velocity does not appear in the equations, in a partial transformation all Cartesian velocity components are required at each face center in order to discretize Eq. 9 with the same accuracy. One alternative would be to solve all components of the momentum equation at each face center, in order for the Cartesian velocity to be attainable without interpolation. Although this approach has been adopted in literature (Maliska and Raithby, 1984), it triples the computational cost. In TOSCA, we follow the approach of Ge and Sotiropoulos (2007), where
the momentum equation is first discretized at cell centers, then interpolated and solved at face centers in a staggered fashion. Cartesian velocity is subsequently reconstructed at cell centers by interpolating contravariant fluxes at the same location. With respect to a standard staggered formulation (e.g. in Cartesian coordinates), this procedure encompasses additional steps for interpolating the discretized momentum equation at face centers and for transforming the interpolated fluxes into the Cartesian velocity at cell centers (flux interpolation is required in either case). It should be noted that the overhead in computational cost
is minimal, as it only involves 1D interpolations along grid lines, for which a second-order central scheme is used. Another slightly different approach (Rosenfeld et al., 1992) is to discretize the momentum equation in a staggered manner. This avoids





interpolating the whole momentum right-hand side at cell faces, but it requires interpolating contravariant fluxes instead. In either case, methods based on partially transformed equations involve an additional interpolation step (as contravariant fluxes and Cartesian velocity are defined at different locations). This poses slightly tighter constraints on the time step value in order to

keep the method stable. For this reason, we opted for an implicit treatment of advection and viscous terms in Eq. 9. Specifically, we use the matrix-free Newton-Krylov solver implemented in PETSc (Balay et al., 2022a), where the iterative Krylov subspace generalized minimum residual (GMRES) method (Saad and Schultz, 1986) is used to solve the linear system associated with each inner iteration. (See Knoll and Keyes, 2004 for a comprehensive review and application of matrix-free Newton–Krylov methods.) In addition, such hybrid staggered/non-staggered formulation facilitates the application of boundary conditions,

which are prescribed on the Cartesian velocity using ghost cells.

Pressure-velocity coupling is provided using a second-order fractional step method similar to van Kan (1986), where velocity is first guessed by solving for the contravariant fluxes, which are then projected into a divergence-free space by means of a pressure correction $\phi$ obtained by solving a Poisson equation. Potential temperature is subsequently solved using the new velocity field, with an implicit treatment of the right-hand side. Time discretization uses a second-order implicit scheme for

both momentum and temperature equations. All derivatives are discretized using the second-order central scheme, while the advection term in Eq. 9 is discretized using a blend between central and QUICK (Leonard, 1979) schemes. The blending is such that QUICK is used in regions of almost uniform or slowly-varying velocity, avoiding the oscillations produced by the central scheme in such regions.

### 2.3 LES Modeling

To model the sub-grid stresses, TOSCA uses the dynamic Smagorinsky model (Lilly, 1992; Germano et al., 1991), with Lagrangian averaging of the model coefficient $C_s$ (Meneveau et al., 1996). The model has been recast in generalized curvilinear coordinates, similar to what was presented in Armenio and Piomelli (2000). The effect of unresolved scales in the momentum equation, after the filtering operation, appears in Cartesian coordinates through the term

$$\frac{\partial}{\partial x_j}\left(\overline{u_i u_j}\right) = \frac{\partial}{\partial x_j}\left(\bar{u}_i \bar{u}_j\right) + \frac{\partial}{\partial x_j}\left(\tau_{ij}^D\right),$$ (11)

where $\bar{\cdot}$ is the filtering operation defined as

$$\bar{\cdot} = \int \bar{\cdot}(\underline{x} - \underline{r}, t) G(|\underline{r}|) d\underline{r},$$ (12)

and $\tau_{ij}^D$ is the deviatoric part of the sub-grid stresses, as the isotropic part is absorbed in the pressure variable. In curvilinear coordinates, Eq. 11 reads

$$\frac{\partial}{\partial l_k}\left(\overline{V^k u_j}\right) = \frac{\partial}{\partial l_k}\left(\overline{V^k}\,\overline{u_j}\right) + \frac{\partial}{\partial l_k}\left(\sigma_j^k\right),$$ (13)

where

$$\sigma_j^k = \overline{V^k u_j} - \overline{V^k}\,\overline{u_j}$$ (14)





and assuming a linear eddy viscosity model,

$$\sigma_i^k = -2\nu_t S_j^k S_{ij} \tag{15}$$

$$\nu_t = C_s \Delta^2 S_{ij} S_{ij}, \tag{16}$$

where $S_j^k = 1/J \partial l_k / \partial x_j$ are the face area vectors, $S_{ij} = \frac{1}{2}(\partial u_i / \partial x_j + \partial u_j / \partial x_i)$ is the symmetric part of the velocity gradient tensor and $\Delta$ is the cubic root of the local cell volume. Using the idea of Germano et al. (1991), a second filter, denoted as $\widetilde{\cdot}$, can be applied which has $\widetilde{\Delta} = 3\Delta$ in TOSCA, leading to the tensor

$$T_j^k = \widetilde{\overline{V_k u_j}} - \widetilde{\overline{V_k}} \, \widetilde{\overline{u_j}} \tag{17}$$

that accounts for the effect of the unresolved plus the smallest resolved scales. The Germano tensor, i.e. the contribution to the resolved stresses from the largest unresolved motions, is defined in generalized curvilinear coordinates by subtracting the tilde-filtered Eq. 13 from Eq. 17

$$G_j^k = T_j^k - \widetilde{\sigma}_j^k = \widetilde{\overline{V_k u_j}} - \widetilde{\overline{V_k}} \, \widetilde{\overline{u_j}}. \tag{18}$$

Using Eq. 15 and 16 to express $\widetilde{\sigma}_j^k$ and $T_j^k$ reads

$$\widetilde{\sigma}_j^k = -2C_s \overline{\Delta}^2 \widetilde{|\overline{S}| S_j^k S_{ij}} \tag{19}$$

$$T_j^k = -2C_s \widetilde{\overline{\Delta}}^2 |\widetilde{\overline{S}}| \widetilde{\overline{S}}_j^k \widetilde{\overline{S}}_{ij}, \tag{20}$$

where in Eq. 20 the approximation $\widetilde{\overline{S_j^k S_{ij}}} = \widetilde{\overline{S}}_j^k \widetilde{\overline{S}}_{ij}$ has been used (good enough for smooth spatial face area vector variation, exact for uniform meshes). Inserting Eq. 19 and Eq. 20 into Eq. 18 leads to

$$G_j^k = -2C_s \widetilde{\overline{\Delta}}^2 |\widetilde{\overline{S}}| \widetilde{\overline{S}}_j^k \widetilde{\overline{S}}_{ij} + 2C_s \overline{\Delta}^2 \widetilde{|\overline{S}| S_j^k S_{ij}} = C_s M_j^k \tag{21}$$

where

$$M_j^k = 2\left[ \overline{\Delta}^2 \widetilde{|\overline{S}| S_j^k S_{ij}} - \widetilde{\overline{\Delta}}^2 |\widetilde{\overline{S}}| \widetilde{\overline{S}}_j^k \widetilde{\overline{S}}_{ij} \right]. \tag{22}$$

It is now possible to find $C_s$ in a least-squares sense as

$$C_s(\underline{x}, t) = \frac{M_j^k G_j^k}{M_m^n M_m^n}.$$

Note that the above relation is not invariant with respect to rotation of the reference frame, because it implicitly contains the face area vectors, hence tensors are no longer symmetric. Variables must then be transformed into physical space to find $C_s$ as

$$C_s(\underline{x}, t) = \frac{G_i^k M_i^q g_{kq}}{M_n^m M_n^l g_{ml}}, \tag{23}$$

where $g_{ij}$ is the covariant metric tensor. Since the $C_s$ coefficient oscillates in space, some sort of average is required. TOSCA follows the approach presented in Meneveau et al. (1996), where the numerator and denominator of Eq. 23 are averaged along





streamlines as

$$\langle G_i^k M_i^q g_{kq} \rangle = I_{GM} = \int_{-\infty}^{t} G_i^k(t') M_i^q(t') g_{kq} W(t-t') dt' \tag{24}$$

$$\langle M_n^m M_n^l g_{ml} \rangle = I_{MM} = \int_{-\infty}^{t} M_n^m(t') M_n^l(t') g_{ml} W(t-t') dt' \tag{25}$$

where $W(t) = 1/T_s \exp(-t/T)$ is a weighting function and $T_s$ is a time scale defined as

$$T_s = 1.5\Delta \left[ 8 G_i^k M_i^q g_{kq} M_n^m M_n^l g_{ml} \right]^{-1/8}. \tag{26}$$

The integrals of Eq. 24 and Eq. 25 can be evaluated as

$$I_{GM}^n(\mathbf{x}) = \epsilon(G_i^k M_i^q g_{kq}) + (1-\epsilon)(I_{GM}^{n-1}(\mathbf{x}-\mathbf{u}\Delta t)) \tag{27}$$

$$I_{MM}^n(\mathbf{x}) = \epsilon(M_n^m M_n^l g_{ml}) + (1-\epsilon)(I_{MM}^{n-1}(\mathbf{x}-\mathbf{u}\Delta t)), \tag{28}$$

where $\epsilon = (\Delta t/T_s)/(1 + \Delta t/T_s)$. We use tri-linear interpolation formulas to evaluate the integrals $I_{GM}$ and $I_{MM}$ at the $\mathbf{x}-\mathbf{u}\Delta t$ position, and all quantities are evaluated at cell centers, including contravariant fluxes, which are linearly interpolated from the faces.

Regarding potential temperature equation, sub-grid fluxes are evaluated following the approach of Moeng (1984), i.e. through the definition of a thermal eddy diffusivity $\kappa_t = \nu_t/Pr_t$, where $Pr_t$ is the turbulent Prandtl number, which depends on stability as

$$Pr = \frac{1}{1 + 2l/\Delta}, \tag{29}$$

$$l = \begin{cases} min\left(\frac{7.6\nu_t}{\Delta} \sqrt{\theta_0/|s|}, \Delta\right) & \text{if } s < 0 \\ \\ \Delta & \text{if } s \geq 0, \end{cases}$$

$$s = g_i \frac{\partial \theta}{\partial x_i}.$$

Note that, if the potential temperature gradient is locally stable, $Pr_t \to 1$, while for neutral or unstable cases $Pr_t = 1/3$. This reflects the decrease of the mixing length scale under stable conditions (Schumann, 1991).

## 2.4 Actuator Models

To represent wind turbines, different models have been implemented following Martínez Tossas et al. (2012), namely the actuator line (AL), actuator disk (AD), and uniform actuator disk (UAD) models. The idea behind actuator models is to represent the wind turbine as a distribution of points, each associated with a force (Sørensen and Shen, 2002; Sørensen et al., 2015; Porté-Agel et al., 2010; Jimenez et al., 2007, 2008). The sum of forces from all points must be equal to the total





wind turbine thrust for the UAD, while AL and AD models also account for blade rotation, hence tangential force. Once the Lagrangian force at each point has been calculated, it is distributed to the surrounding mesh cells through a projection function. In TOSCA, a classical isotropic Gaussian projection is used, namely

$$270 \quad g(x,y,z) = \frac{1}{\epsilon \pi^{3/2}} \exp\left( -\frac{(x-x_0)^2 + (y-y_0)^2 + (z-z_0)^2}{\epsilon^2} \right) \tag{30}$$

where $(x_0 \; y_0 \; z_0)$ is the position of the actuator point, and $\epsilon$ is a tunable parameter, corresponding to the standard deviation of the Gaussian projection function. Note that while the projection function should integrate to unity to preserve each point force, this is never exactly possible, and the projection distance is cut when 99% of the Gaussian volume has been taken into account. Moreover, the Gaussian width-to-grid size ratio should be larger than two in order to avoid large projection errors and numerical instabilities (Martínez Tossas et al., 2012).

The definition of the turbine point mesh, and the evaluation of the point force are different depending on the specific model. In the AL model, each rotor blade is represented by a line of points, which are physically rotated at each iteration, making it an unsteady model. In AD and UAD models, the number of points in the azimuthal direction is not equal to the number of blades, and it is usually set to a high value. For both AL and AD models, the point force is calculated exploiting the blade element theory (BEM, see Glauert, 1935). First, the radially varying velocity is estimated at each point, using information from the CFD mesh. This is known as velocity sampling, and different methods have been proposed (Churchfield et al., 2017). TOSCA samples the velocity at the actuator point, using nearest-neighbor interpolation from the closest mesh cell. Next, velocity magnitude and angle of attack are given as input to appropriate airfoil tables, which return lift and drag at the point location. Various airfoil tables are used along the blade radius because the airfoil type usually changes along the blade span, as does the operating Reynolds number. Lift and drag at each actuator point are then distributed to the surrounding CFD cells by convolution with the projection function. For the UAD model, the blade loading is uniform, and the force is calculated by dimensionalizing the turbine thrust coefficient with the freestream velocity and the portion of rotor area belonging to that point. In waked conditions, the concept of freestream velocity is not well defined. Hence, a common practice is to first average the wind velocity on the rotor disk, then use the momentum theory to infer the corresponding freestream velocity (Meyers and Meneveau, 2010). In our framework, since AD and AL models also account for blade rotation, they can be coupled with a rotor inertia and control system dynamics solver (pitch and angular velocity controllers), while nacelle yaw can be applied to any of the three models.

## 2.5 Controllers

This section reviews the current state of the art for velocity controllers in precursor simulations, and presents a novel technique, which we refer to as geostrophic damping, which allows control of the hub-height velocity while avoiding inertial oscillations generated by the Coriolis force. Moreover, a simple temperature controller is also presented that maintains a constant average potential temperature profile throughout the precursor simulation.





### 2.5.1 Velocity Controller

In the precursor simulation, the flow is usually driven by a uniform horizontal pressure gradient, which is related to the
geostrophic wind components by the geostrophic balance at equilibrium

$$\frac{1}{\rho_0}\frac{\partial p_\infty}{\partial x} = f_c V_G \qquad \frac{1}{\rho_0}\frac{\partial p_\infty}{\partial y} = -f_c U_G \tag{31}$$

where $f_c = 2\Omega_z$ is known as the Coriolis parameter. Using the above equations to prescribe the driving pressure gradient does
not give any control over velocity magnitude and direction at the wind turbine hub height. In fact, the latter will be a result of
the turbulent stresses inside the boundary layer, which are not known a-priori. However, being able to control these parameters
is convenient in wind farm simulations as it allows the operation point of the turbines to be easily prescribed. To this end,
Sescu and Meneveau (2014) and Allaerts and Meyers (2015) developed and tuned an algorithm that slowly rotates the flow in
the domain, allowing to control the wind direction at a specified height. Later, Stieren et al. (2021) used the same approach
to impose dynamic wind direction changes. Besides the driving pressure gradient, evaluated using Eq. 31, the additional cross
product $-\epsilon_{ijk}\omega_i u_j \widehat{x}_k$ is added to the momentum equation's right-hand side, where the angular frequency $\omega_i$ is calculated
based on the angle difference at the reference height (see Allaerts and Meyers, 2015 for details on this procedure). Such a
method, which we will refer to as the geostrophic controller, does not entirely solve the issue, as velocity magnitude at the hub
height is still unknown a-priori. Nevertheless, a different approach exists, available for example in SOWFA, which allows to
prescribe both magnitude and direction at a specified height $h_{ref}$. In particular, given a desired velocity $u_{ref,i}$, which should
be maintained at $h_{ref}$, an error vector can be defined as the difference between the reference wind and the velocity sampled
at the reference height, averaged over the homogeneous directions. At this point, a proportional-integral controller can be used
to evaluate the driving pressure gradient (i.e. the third term on the right-hand-side of Eq. 2) such that desired speed and angle
are maintained at $h_{ref}$. This approach will be referred to as the pressure controller. In TOSCA both controller methods are
implemented, and the driving pressure gradient in the second type of controller is evaluated as

$$\frac{1}{\rho_0}\frac{\partial p_\infty}{\partial x_i} = r\left(\alpha e_{P,i} + (1-\alpha)e_{I,i}^n\right) \tag{32}$$

$$e_{P,i} = \left(u_{ref,i} - \langle u_i(h_{ref})\rangle_{xy}\right)/\Delta t \tag{33}$$

$$e_{I,i}^n = (1-\Delta t/T)e_{I,i}^{n-1} + (\Delta t/T)e_{P,i} \tag{34}$$

where subscript $i$ refers to the $i^{\text{th}}$ component, $e_{P,i}$ is the proportional error, $e_{I,i}^n$ is the integral error evaluated at time step $n$,
$r$ is a relaxation factor, $\alpha$ is the proportional fraction of the controlling action, $T$ is the time filter for the integral error, $\Delta t$ is
the time step size and $\langle\cdot\rangle_{xy}$ denotes a spatial average along the homogeneous directions $x$ and $y$. In the present study, we set
$r = 0.7$, $\alpha = 0.8$ and $T = 2$ h.

On one hand, the pressure controller is more convenient for wind turbine simulations, as hub wind and direction can be
directly specified. However, unlike the geostrophic controller, it does not provide knowledge of the geostrophic wind a-priori,
making it impossible to initialize the flow such that Eq. 31 is satisfied. An inconsistency in the initial condition produces inertial
oscillations above the boundary layer, as the initial wind speed aloft differs from its equilibrium geostrophic value. This can be



easily verified by noting that the unsteady form of Eq. 31 (see for example Stull, 2016), namely

$$
\begin{cases}
\frac{\partial u}{\partial t} + f_c(V_G - v) = 0 \\
\frac{\partial v}{\partial t} - f_c(U_G - u) = 0
\end{cases}
\tag{35}
$$

represents an undamped linear oscillator with angular frequency $f_c$. In particular, if $v \neq V_G$ or $u \neq U_G$, at any point during the simulation, inertial oscillations will be produced. In some wind energy applications, for example, when studying the formation of atmospheric gravity waves above the boundary layer, the physics of the problem strongly depends on the magnitude of the geostrophic wind. In such cases, results would be negatively impacted by these inertial oscillations, whose amplitude depends on the initial condition.

Nevertheless, being able to exactly define the wind speed and direction at a specified height is a desirable property of the pressure controller. For this reason, we developed a new methodology that allows to remove these inertial oscillations, enabling the use of the pressure controller also in those cases where a steady state geostrophic wind is preferred. First, we note that the system of equations 35 can be damped by introducing an additional term as follows

$$
\begin{cases}
\frac{\partial u}{\partial t} + 2\alpha f_c(u - U_G) + f_c(V_G - v) = 0 \\
\frac{\partial v}{\partial t} + 2\alpha f_c(v - V_G) - f_c(U_G - u) = 0
\end{cases}
\tag{36}
$$

where the coefficient $\alpha$ determines if the system is over-damped ($\alpha > 1$), under-damped ($\alpha < 1$) or critically damped $\alpha = 1$. With some manipulation, Eq. 36 can be rewritten as

$$
\begin{cases}
\frac{\partial^2 u}{\partial t^2} + 2\alpha f_c \frac{\partial u}{\partial t} + 2\alpha f_c^2(v - V_G) + f_c^2(u - U_G) = 0 \\
\frac{\partial^2 v}{\partial t^2} + 2\alpha f_c \frac{\partial v}{\partial t} - 2\alpha f_c^2(u - U_G) + f_c^2(v - V_G) = 0
\end{cases}
\tag{37}
$$

These equations slightly differ from a conventional spring-mass-damper system in the additional coupling terms $2\alpha f_c^2(v - V_G)$ and $2\alpha f_c^2(u - U_G)$.

We observed that the presence of these terms enhances the damping action, halving the exponent of the decay rate, characterized by an e-folding time of $1/(2\alpha f_c)$. In order for the oscillation amplitude to reach less than 3% of the initial value, a damping time $T_{3\%} = ln(100/3)/(2\alpha f_c)$ is necessary. Note that, in order for the damping term to be evaluated, knowledge about the geostrophic wind components is still required. We deduce $U_G$ and $V_G$ from the driving pressure gradients imposed by the pressure controller (Eq. 32, 33, 34) by means of the definition of the geostrophic wind speed (i.e., the geostrophic balance given by Eq. 31). In addition, we filter the obtained geostrophic components using a filter constant of $0.2\pi/f_c$ (corresponding to one-tenth of the inertial oscillation period). Finally, we highlight that the damping action should start after the boundary layer has become fully developed, as the pressure gradient prescribed by the controller depends on the turbulent stresses if $h_{\text{ref}}$ is inside the BL. In our simulations, we start the damping action after almost one inertial period ($T_D \approx 2\pi/f_c$), and we maintain the damping active for at least a time equal to $T_{3\%}$. In order for this geostrophic damping method not to affect the velocity inside the ABL, we smoothly bring the damping to zero below a certain height by multiplying the damping terms with



the following function

$$f_d = \frac{1}{2}\left[1 + \tanh\left(\frac{7(h - H_d)}{\Delta_d}\right)\right] \tag{38}$$

where $H_d$ is the height where the damping has halved its strength. If the simulation models a capping inversion layer, we set $H_d = H$, where $H$ is the capping inversion center, and $\Delta_d = \Delta$, where $\Delta$ is set as the capping inversion width.

### 2.5.2 Temperature Controller

When running precursor CNBL simulations, the predicted ABL height, as well as the final value of potential temperature at the ground, depend on the mixing history experienced inside the boundary layer. This is in turn affected by the specific LES setup
and the type of discretization used. Moreover, the impact of such code details is made even more noticeable by the fact that these simulations usually run for a very long time (of the order of $2\pi/f_c$). For example, SP-Wind, which employs a pseudo-spectral discretization in the horizontal directions and an energy-conservative fourth-order advection scheme in the vertical, predicts less mixing than other pseudo-spectra codes, such as NCAR-LES or Wire-LES, which use for example a second-order central scheme in the vertical direction (this can be appreciated in Fig. 3). Besides, in finite-volume codes, like TOSCA or
SOWFA, an upwind-biased advection scheme is usually preferred, as it stabilizes the numerical method, but it does not allow to conserve mechanical energy. These considerations pose comparison issues among different codes, as their differences will have an impact for example on the final ABL height, inversion thickness and potential temperature jump and, in general, on the heating history of the boundary layer, ultimately affecting the successor solution, in which wind turbines are present. In particular, if a CNBL precursor is run with a certain initial potential temperature profile, this will evolve differently based on
both the adopted simulation framework and the length of the precursor run, determining a discrepancy in the initial condition of the wind farm simulation.

In the present work, we propose to apply a potential temperature controller in the precursor simulation, so that the successor can be exactly run with the intended temperature profile. Our objective is to make results from different codes more comparable, ensuring that the successor background potential temperature profile matches the precursor initial condition. In particular, we
apply the following height-dependent source term on the right-hand side of Eq. 3

$$s_\theta(h) = r\frac{\bar{\theta}(h) - \langle\theta(h)\rangle_{xy}}{\Delta t}, \tag{39}$$

where $\bar{\theta}(h)$ is the desired vertical potential temperature profile, taken as the initial value of $\langle\theta(h)\rangle_{xy}$, which is average of the potential temperature along the homogeneous directions at a given height. The parameter $r$ is a relaxation coefficient that we set to 0.7. Note that a very similar method was used by Allaerts et al. (2020, 2023) to drive LES simulations with realistic
mesoscale information from mesoscale models or observations.

### 2.6 Hybrid Off-line/Concurrent Precursor

Wind turbine wake recovery, and thus power production, are greatly influenced by background atmospheric turbulence. As a consequence, prescribing a physical turbulent inflow is necessary if real wind turbine operation is to be simulated. A commonly





used approach is the so-called precursor-successor method, where a first simulation of the sole ABL, without wind turbines,
is run until turbulence reaches steady state statistics. After this first phase, the latter is further progressed, and velocity and
potential temperature are saved on a plane parallel to the inlet boundary, at each iteration, forming the inflow database. At
this point, the simulation with the wind turbines (successor) is started and, at each time step, the inflow boundary condition is
interpolated from the saved slices at the two closest times in the database, so that precursor and successor time steps can take
different values. The above methodology implies no periodicity of the domain in the streamwise direction, and has proved to
work extremely well for isolated turbine simulations, cases where the ABL height is not perturbed by objects located below it,
or in the absence of thermal stratification. On the contrary, when thermal stratification is present, atmospheric gravity waves
can be triggered above the ABL, and a careful design of the simulation should prevent such waves from being reflected by the
physical boundaries. In particular, the LES set-up should be equipped with damping regions at the top, inlet and outlet, or if
streamwise periodic boundary conditions are used, only one damping region in the streamwise direction is then required. The
latter, also known as fringe region, must ensure that turbine wakes are removed to avoid being re-introduced into the domain
by periodic boundaries, and that a realistic turbulent inflow is reached at the fringe exit. To achieve this, the desired flow that is
used to compute the damping term should ideally contain time-resolved turbulent structures at every cell located in the fringe.
The precursor is then advanced in sync with the successor, in a domain larger or equal to the fringe region, so that velocity and
temperature fields are available, at each time step and spatial location, to compute damping sources.

In order to spin-up the wind farm simulation, we developed a three step procedure (see Fig. 1), where both off-line and
concurrent precursor methodologies are adopted. We believe that the latter method is necessary when dealing with wind farm-
induced gravity waves, as it allows to avoid wave reflections, while prescribing a realistic turbulent inflow at the same time.
Since the concurrent precursor domain should coincide with the successor—whose size is dictated by the gravity waves and
wind farm—both in the spanwise and vertical directions, it is usually oversized from the turbulence generation point of view.
This leads to a considerable amount of computational resources being consumed when starting up the unperturbed ABL. In
fact, domains of much smaller size are used in literature when the sole ABL is of interest, or when the inflow data is generated
using the off-line precursor technique.

In TOSCA, we exploit the flexibility of the finite-volume formulation by combining the two techniques. In particular, we
initialize the ABL on what we refer to as the off-line precursor domain, which can be arbitrarily defined both in the streamwise
and vertical directions. The only requirement is that the spanwise size of the successor is an exact multiple of the off-line
precursor domain. When turbulence has reached a statistically steady state, we save flow slices of velocity and potential
temperature from this domain into a so-called inflow database. After this first phase, the concurrent precursor and successor
simulations are started, and streamwise inflow-outflow boundary conditions are used in the former for one flow-through time.
Inflow slices from the inflow database are periodized in the spanwise direction, while extrapolation is performed in the vertical
direction. We note that it is extremely important that the flow above the inversion layer does not contain any periodic variations
in time, as this would be noticed in the successor, at streamwise intervals equal to the concurrent precursor domain length.
For this reason, we average the off-line precursor data at the ten highest cells, and slowly merge the instantaneous data to this
average across such an interval. This removes even the smallest periodic content in the flow above the boundary layer, which



is now characterized by a truly constant geostrophic wind. For instance, we weight the average and instantaneous velocities
using two hyperbolic weighting functions (Eq. 38 is used for both, but a minus sign after unity is applied for the instantaneous
velocity). $H_d$ is set to the height of the fifth cell center from the off-line precursor top boundary, while $\Delta_d$ is equal to the width
of the ten highest cells.

After the concurrent precursor has run for one flow-through time, the turbulent inflow has reached the outlet, and stream-
wise boundary conditions are switched to periodic. At this point, the simulation is self-sustained and we run precursor and
successor simultaneously for one successor flow-through time so that gravity waves and wind turbine wakes are formed. For
the simulation presented in Sec. 5, we ran the off-line precursor for $10^5$ s, the concurrent-precursor spin-up phase, where we
used inflow-outflow boundary conditions for 600 s, and the overall successor spin-up for 5000 s (note that this phase can start
in parallel with the previous one). Data were gathered from 105000 s to 120000 s.

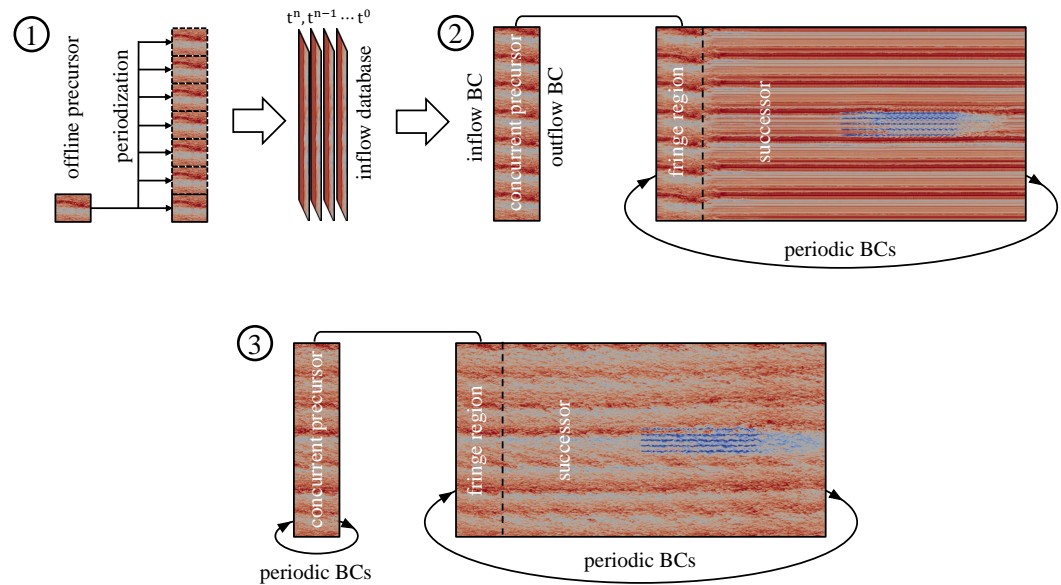

**Figure 1.** Sketch of the hybrid off-line/concurrent precursor method.

The drawback of such method is that a spanwise periodicity is introduced in the concurrent-precursor and successor domains.
If the Coriolis force is active, this can be broken everywhere except at hub height, where the flow is aligned with the x axis, by
setting the off-line precursor and the concurrent-precursor streamwise domain length to a different value. At the hub-height,
the larger turbulence structures might be locked in position if they span the whole domain length. Although they will eventually
disappear, they result in slow convergence of flow averages at the hub-height. This issue has been already observed in the past
for example by Munters et al. (2016), who proposed to use shifted periodic boundary conditions in the concurrent-precursor
domain.



Nevertheless, the proposed hybrid method, sketched in Fig. 1, is very convenient as it allows to reduce the overall computational cost of the ABL spin-up phase. In fact, the latter is run on a domain whose size is dictated by the real flow physics, rather than on quantities such as the wind farm dimensions or wavelength of atmospheric gravity waves, which are only of interest at later simulation stages.

## 3 Validation

In this section, we validate the developed solver using three different benchmark cases. In Sec. 3.1, we simulate an NREL 5MW Reference wind turbine, operating in a uniform inflow equal to 8 m/s, and compare our results to Martínez Tossas et al. (2012). In Sec. 3.2 we validate the ability of TOSCA to simulate conventionally neutral boundary layer (CNBL) evolution, comparing our results against data from different LES codes reported by Allaerts (2016). Finally, in Sec. 3.3, an infinite wind farm in a turbulent boundary layer without thermal stratification is compared to experimental and numerical data collected by Chamorro and Porté-Agel (2010) and Stevens et al. (2018), respectively.

### 3.1 Isolated Rotor in Uniform Inflow

In this validation case, we perform two simulations of the NREL 5MW reference wind turbine (Jonkman et al., 2009) using the ADM and the ALM techniques, with a uniform inflow velocity of 8 m/s. Periodic boundary conditions are applied at the upper, lower and spanwise boundaries. At the outlet, a zero normal gradient on velocity outflow is specified. The domain is 10 rotor diameters in all directions, with the turbine rotor placed in the geometric center of the domain. The mesh is graded in all directions from a resolution of 16.8 m next to all boundaries to 2.1 m near the wind turbine. In particular, this fine region, where the mesh is uniform, extends 1 diameter upstream of the turbine and 5 diameters downstream. In the other two directions, it extends beyond the edge of the rotor for 1 diameter.

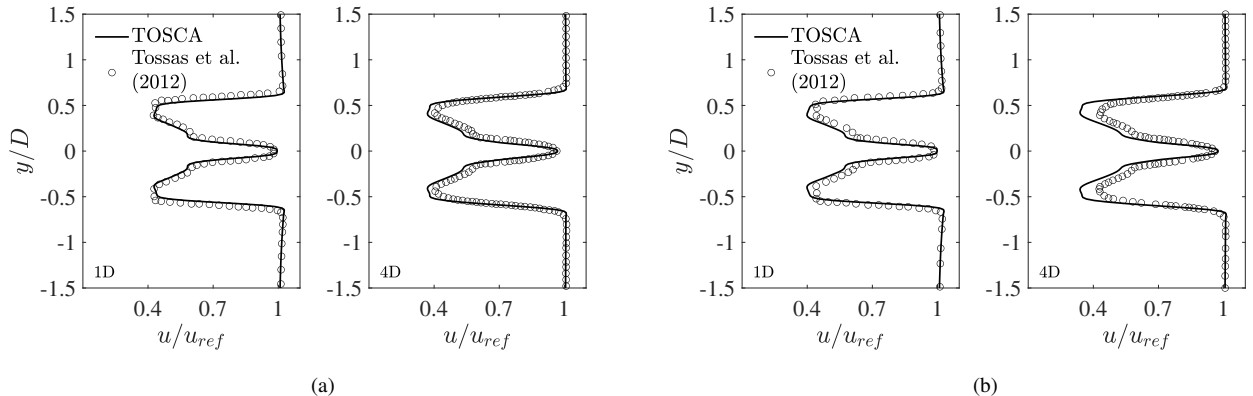

**Figure 2.** Normalized wind speed deficit 1D and 4D behind a wind turbine represented by (a) ADM and (b) ALM.



For this case, the standard Smagorinsky model was used, where we set the $C_s$ coefficient of Eq. 16 to 0.028224 (corresponding to the value of $c_s = \sqrt{C_s} = 0.168$ used in Martínez Tossas et al., 2012). The ALM has 63 points in the radial direction, while the ADM has 63 and 72 points in the radial and azimuthal direction. The rotational speed of the wind turbine is set to a constant value of 9.1552 rpm in all cases. The projection $\epsilon$ in Eq. 30 is set to 4.2 m. Both simulations are advanced for 300 s, after which data are averaged for the next 300 s. Fig. 2 shows normalized wind speed deficit at 1 and 4 downstream rotor diameters for ALM and ADM simulations performed with both TOSCA and by Martínez Tossas et al. (2012). An excellent match can be observed at 1 diameter for both models, while at 4 diameters TOSCA predicts sightly higher deficits, especially for the ALM. This difference is due to an earlier breakdown of the blade-tip vortices in the simulations of Martínez Tossas et al. (2012), which was performed with OpenFOAM. As OpenFOAM is an unstructured code, we believe that non-hexahedral elements arising from the three successive refinement regions produce some small oscillations in the velocity, which is seen by the simulation as added turbulence intensity, determining an earlier breakdown of the blade-tip vortices. This effect is not present in TOSCA, as the mesh is fully structured and smoothly graded from 16.8 to 2.1 m. In van der Laan et al. (2014), the same case is run without a turbulence model using EllipSys3D and SnS, and a higher maximum deficit than both TOSCA and Martínez Tossas et al. (2012) has been observed at 2.5 diameters. Such discrepancy between different codes in predicting turbine wake recovery is not observed when a precursor is used to prescribe the inflow, as wake mixing is guided by ABL turbulence instead of numerical oscillations. In Tab. 1 we report the aerodynamic power produced by the wind turbine as predicted by TOSCA with the two actuator models and that obtained by Martínez Tossas et al. (2012).

|          | ALM [MW] | ADM [MW] |
|----------|----------|----------|
| TOSCA    | 2.18     | 2.04     |
| OpenFOAM | 1.92     | 2.02     |

**Table 1.** Wind turbine power as predicted by TOSCA and OpenFOAM for the ADM and ALM model. OpenFOAM data correspond to Martínez Tossas et al. (2012) with a mesh resolution of 2.1 m and projection width equal to 4.2 m.

The ADM matches well with results from Martínez Tossas et al. (2012), while TOSCA's ALM predicts a slightly higher power. We attribute such differences to how the velocity is sampled at the actuator points. In TOSCA, nearest neighbor interpolation from the background mesh is adopted, which can result in small differences in the sampled wind, as the actuator point does not coincide with the closest cell center in general. Moreover, as pointed out by Churchfield et al. (2017), failing to sample the velocity at the geometric center of the projection function would determine self-induction at the sampling location. Using for example a linear interpolation at the actuator point would probably solve such a mismatch.

### 3.2 CNBL Evolution

In this section, we validate TOSCA's ability to perform CNBL simulations by running two cases of a neutral boundary layer, developing against a stable background stratification with lapse rate of 1 K/km and 10 K/km. In both validation cases, the geostrophic wind is $G = 10$ m/s, the Coriolis parameter is set to $f_c = 10^{-4}\ s^{-1}$, the surface roughness is $z_0 = 0.01$ m and the





reference temperature is 290 K. The numerical domain size is $3\text{ km} \times 3\text{ km} \times 2\text{ km}$, with $256^3$ grid points. This corresponds to a grid resolution of approximately $11.7\text{ m} \times 11.7\text{ m} \times 7.8\text{ m}$. Results are compared with results from SP-Wind (Allaerts, 2016), Wire-LES (Abkar and Porté-Agel, 2013), and NCAR-LES (Pedersen et al., 2014), which all use a pseudo-spectral horizontal

discretization.

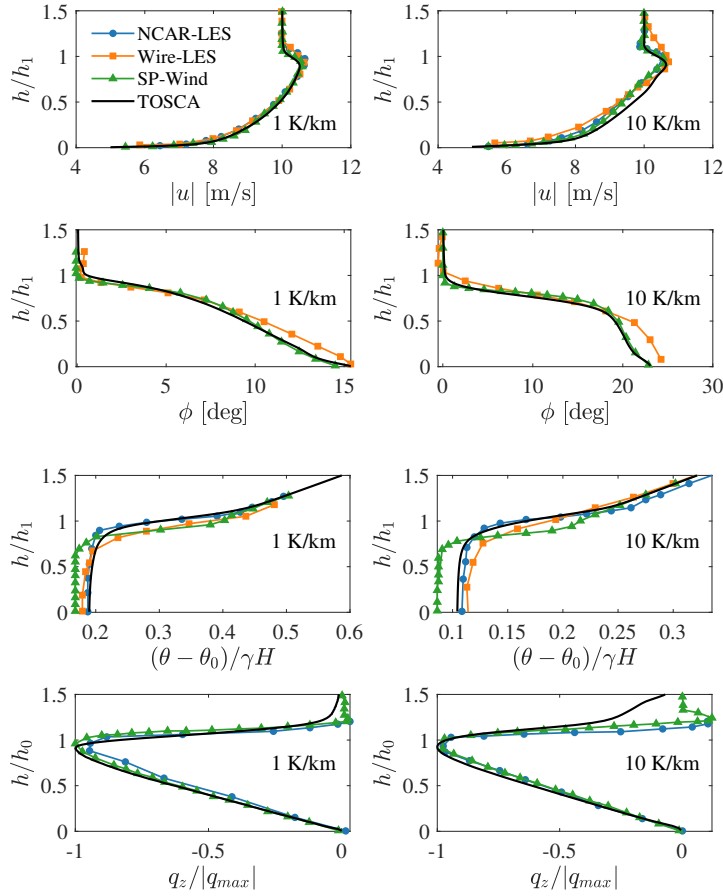

**Figure 3.** Vertical profiles averaged over the last simulation hour of, from top to bottom, velocity magnitude, wind angle, potential temperature and heat flux. The initial lapse rate is (left) 1 K/km and (right) 10 K/km.

Velocity and temperature fields are initialized with a constant and linear profile equal to the geostrophic velocity and background stratification respectively. Furthermore, sinusoidal perturbations are added to the velocity profile, below 100 m, with an amplitude of $0.1G$ and 12 periods in the $x$ and $y$ directions to trigger turbulent fluctuations. Simulations are advanced in time for 24 h, and results are averaged over the last hour. Periodic boundary conditions are applied in the horizontal directions,

while a slip boundary condition is applied at the upper boundary. At the ground, the wall shear stress is prescribed through classic Monin-Obukhov similarity laws (Monin and Obukhov, 1954; Paulson, 1970; Etling, 1996), following the approach of



Yang et al. (2017) to address the log-layer mismatch. The flow is driven using pressure gradients obtained from Eq. 31. We do not apply any velocity controller in these simulations, so that the wind direction inside the boundary layer is free to change, while the geostrophic wind remains aligned with the $x$-axis. Fig. 3 compares vertical profiles of velocity magnitude, horizontal wind direction, potential temperature and kinematic heat flux obtained from TOSCA, with profiles reported by Allaerts (2016); Abkar and Porté-Agel (2013) and Pedersen et al. (2014). Very good agreement is found in the horizontal wind direction and magnitude. Regarding temperature profiles, TOSCA is more aligned with NCAR-LES and Wire-LES results, while SP-Wind predicts a slightly lower inversion layer and potential temperature at the ground. This highlights how turbulent mixing is predicted differently by the four codes. Heat flux profiles agree well below the inversion layer, with the exception that TOSCA predicts a more diffused kinematic heat flux profile above the inversion layer for the 10 K/km case. We do not have a clear explanation for such behavior.

In Tab. 2 quantitative parameters of the resulting ABLs are reported for the different cases and LES codes. The reference temperature $\theta_0$, the capping inversion strength $\Delta\theta$ and the inversion width $\Delta h$ are evaluated by a least-squares fit of the resulting temperature profiles with the model proposed by Rampanelli and Zardi (2004). The ABL height is taken as the center of the capping inversion layer. Both vertical profiles of Fig. 3 and quantitative ABL parameters reported in Tab. 2 demonstrate that TOSCA is well aligned with results from Pedersen et al. (2014); Abkar and Porté-Agel (2013), thus capable of conducting CNBL simulations.

|  | $\gamma$ [K/km] | $\theta_0$ [K] | $\Delta\theta$ [K] | $\Delta h$ [m] | $H$ [m] | $u^*$ [m/s] | $q_{min}/10^{-4}$ [Km/s] |
|---|---|---|---|---|---|---|---|
| TOSCA | 1 | 290.38 | 0.50 | 213 | 783 | 0.34 | -7.0 |
| NCAR LES | 1 | 290.36 | 1.18 | 342 | 800 | 0.37 | -5.8 |
| Wire LES | 1 | 290.36 | 0.54 | 229 | 717 | 0.36 | – |
| SP-Wind | 1 | 290.34 | 0.41 | 148 | 687 | 0.34 | -4.2 |
| TOSCA | 10 | 292.08 | 2.85 | 160 | 429 | 0.34 | -22.5 |
| NCAR LES | 10 | 292.17 | 2.92 | 119 | 439 | 0.37 | -25.5 |
| Wire LES | 10 | 292.28 | 3.03 | 210 | 425 | 0.35 | – |
| SP-Wind | 10 | 291.72 | 2.08 | 97 | 356 | 0.34 | -13.8 |

**Table 2.** Quantitative ABL results from the two ABL simulations performed with different codes.

In addition, we note that simulations performed using SP-Wind consistently predict less mixing than other codes, which is confirmed by the lower absolute value of the minimum kinematic heat flux $q_{min}$. This could be due to the fourth-order energy conservative scheme, which is adopted in SP-Wind simulations, while other codes employ second or third-order non-conservative advection schemes.



## 3.3 Infinite Wind Farm in Neutral Conditions

In this section, we run the same infinite wind farm simulation that has been conducted in Stevens et al. (2018), corresponding to the wind tunnel experiments performed by Chamorro and Porté-Agel (2010). The scaled wind farm consists of 30 wind

turbines, arranged in an aligned configuration with 3 columns and 10 rows. Spanwise and streamwise spacings are set to $S_y = 4D$ and $S_x = 5D$ respectively, where $D = 0.15$ m is the turbine diameter. The wind farm is made periodic in the spanwise direction by placing turbine columns 1 and 3 at a distance of $S_y/2$ from the lateral boundaries, where periodic boundary conditions are applied. In Stevens et al. (2018), simulations are run with both the ALM and the non-rotating uniform ADM model, which we refer to as the uniform actuator disk model (UADM), and each wind farm row has a different $C_t$. For this

validation case, we did not attempt to use rotating actuator models (ADM or ALM), as $C_t$ coefficients applied in Stevens et al. (2018) at some rows are higher than the value of $C_{t,max}$ from their reported BEM calculations. Therefore, since it wouldn't have been possible to match their exact angular velocity for some of the rows, we decided to opt for the UADM, where turbine-specific thrust calculation at the $p-th$ disk element is solely based on the thrust coefficient $C_t$ and the freestream velocity $U_\infty$ as

$$\mathbf{f}_p = \frac{1}{2} U_\infty^2 dA_p C_t \widehat{\mathbf{e}}_t. \tag{40}$$

In the above expression, $dA_p$ is the disk area associated to the $p_{\text{th}}$ actuator disk point, $\widehat{\mathbf{e}}_t$ is a vector normal to the rotor disk, pointing in the upstream direction, and $U_\infty$ is evaluated from the average disk velocity $U_{disk}$ exploiting the momentum theory. For instance, $U_\infty = U_{disk}/(1-a)$ where $a$ is the induction factor, related to the thrust coefficient as $C_t = 4a(1-a)$. Note that Eq. 40 can be rewritten in an equivalent form by using $U_{disk}$ and the disk-based thrust coefficient $C_t' = C_t/(1-a)^2$ in place of $U_\infty$ and $C_t$. We use the latter formulation in the present case, as Stevens et al. (2018) reported the value of $C_t'$ at each wind

farm row.

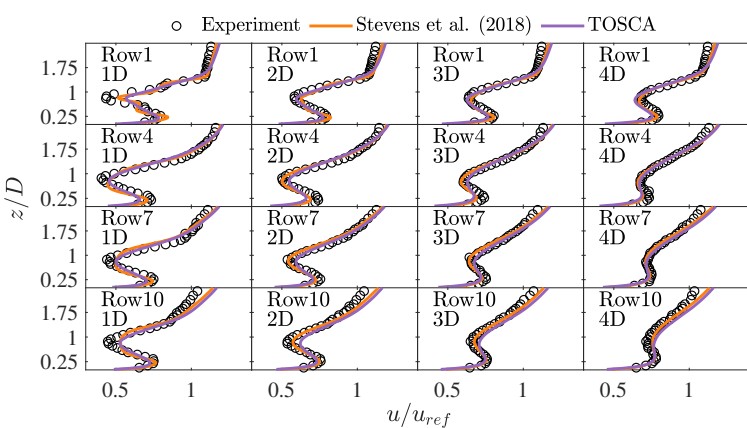

**Figure 4.** Velocity profiles made non-dimensional with the hub-height velocity for rows 1, 4, 7, and 10, averaged on the three columns. For each row, the wake evolution is reported at 1,2,3 and 4 diameters downstream.



The wind tunnel model used in the experiment by Chamorro and Porté-Agel (2010) is the GWS/EP-6030 turbine, it has a hub height of 0.125 m, an overhang of 0.03 m, a hub radius of 0.0075 m and a tower diameter of 0.01 m. Tower and nacelle have been modeled following the approach of Stevens et al. (2018), except for the projection function, which is given by Eq. 30. The value of $\epsilon$ has been set to 0.02452 for tower and nacelle, and to 0.03515625 for the rotor, in order to closely match their approach. The tower is represented by 50 actuator points, and is characterized by a drag coefficient of 0.68, while the nacelle consists of a single point where the force is calculated by dimensionalizing a drag coefficient of 4. Moreover, the rotor has been discretized using 20 radial points and 50 azimuthal points. To prescribe a turbulent inflow, Stevens et al. (2018) used the concurrent precursor technique, as their code is pseudo-spectral. Since TOSCA is a finite-volume code and inflow-outflow conditions can be applied, we opted for the computationally-cheaper off-line precursor technique described in Sec. 2.6. The precursor domain is 1.8 m × 1.8 m × 0.675 m, with 129 × 129 × 145 cells in each direction in order to match their cell size. The flow is driven by the pressure controller described in Sec. 2.5.1, with a desired flow velocity $u_{ref}$ of 3 m/s at the hub height. Potential temperature stratification is turned off so that the boundary layer height coincides with the domain size in the vertical direction $z$. We used periodic boundary conditions in the horizontal directions, while at the upper boundary a slip condition is applied. At the ground, we used the same similarity laws of Stevens et al. (2018), with an equivalent roughness height of 0.03 mm.

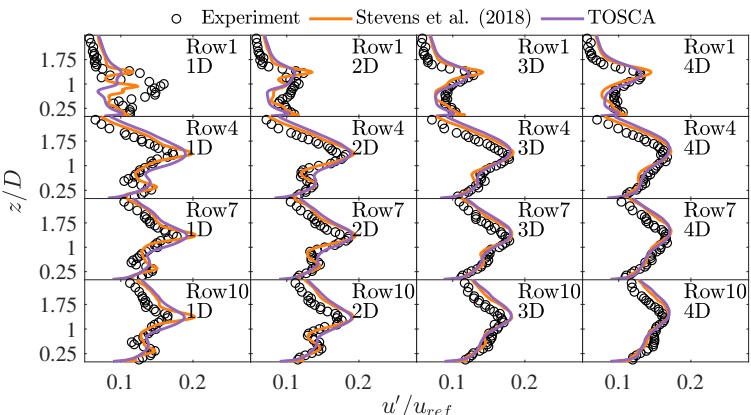

**Figure 5.** Profiles of streamwise fluctuations non-dimensionalized with hub-height velocity for rows 1, 4, 7, and 10, averaged on the three columns. For each row, the wake evolution is reported at 1,2,3 and 4 diameters downstream.

The precursor is run for 100 s (corresponding to $\approx$ 160 flow through times), after which we saved the inflow field at each time step for 300 s. In the successor, we apply the pre-calculated inflow and source terms from the precursor, linearly interpolating from the two closest available times. At the outlet, we use a zero normal gradient condition on the velocity. The remaining boundaries are treated in the same manner as the precursor. The successor domain is 8.25 m × 1.8 m × 0.675 m, with 588 × 129 × 145 cells in each direction. The first row of the wind farm is located 5 diameters from the inlet boundary, matching the



setup of Stevens et al. (2018). The successor is advanced in time for 300 s, and we start gathering data after one flow through time ($\approx 3$ s).

In Fig. 4 we show the velocity profiles, averaged among the wind farm columns, for rows 1, 4, 7, and 10, together with experimental data from Chamorro and Porté-Agel (2010) and numerical results from Stevens et al. (2018). As can be noticed, TOSCA matches very well with both numerical and experimental data. In the upper portion of the velocity profile, for increasing wind turbine row, both TOSCA and results from Stevens et al. (2018) predict higher velocities than the experiment. This effect is given by wind farm area blockage in the numerical domain, which causes the flow to accelerate close to the upper boundary in order to conserve mass. In the experimental data, this is not observed, as the height of the wind tunnel test section was 1.7 m.

In Fig. 5, profiles of $u'/u_{ref}$ are reported for the same location of Fig. 4. Given the velocity time history at a point, we first evaluate $u'u'$ by averaging the square of the fluctuation history, obtained as the difference between the velocity signal and its average. Then $u'$ is obtained as the square root of $u'u'$. Results show that TOSCA is well aligned with results from Stevens et al. (2018), both predicting higher fluctuations than experiments in the top-most downwind part of the wind farm for the reason mentioned above. These results demonstrate that TOSCA accurately predicts turbine-wake interactions inside a wind farm, both in the mean and in the fluctuations, making it suitable for the simulation of wind turbines immersed in a turbulent boundary layer.

## 4 CNBL Simulations with Different Controllers

In this section, we present CNBL results obtained using the different velocity and temperature controllers described in Sec. 2.5. In particular, we compare case S2 from Allaerts and Meyers (2017) against results obtained from TOSCA using both pressure and temperature controllers at the same time (case PT), and pressure and geostrophic controllers with no temperature forcing (case P and G, respectively). A summary of the different cases with the relative controllers is given in Tab. 3. The simulations employ periodic boundary conditions in the horizontal directions and a slip boundary condition at the upper boundary. Classic Monin-Obukhov similarity theory is enforced at the ground.

|  | Velocity | Temperature | Geo. Damping |
|---|---|---|---|
| G | geostrophic forcing + hub-wind angle | off | off |
| P | pressure forcing based on hub-wind | off | on |
| PT | pressure forcing based on hub-wind | mean temperature forcing | on |
| S2 | geostrophic forcing + hub-wind angle | off | off |

**Table 3.** Velocity and potential temperature controlling strategies for the cases presented in this section and for case S2 from Allaerts and Meyers (2017).





Following Allaerts and Meyers (2017), the domain size is 9.6 km × 4.8 km × 1.5 km in the streamwise, spanwise and vertical directions respectively, discretized using 320 × 320 × 300 cells in each direction. Case G is forced with a geostrophic wind speed of 12 m/s, matching the setup used by Allaerts and Meyers (2017) in case S2. Conversely, in P and PT cases the pressure controller aims to maintain a wind speed of 10.871 m/s at $h_{\text{ref}} = 100$ m, thus matching the hub height wind speed obtained from case G. The Coriolis parameter $f_c$ is set to $10^{-4}$. In all cases, potential temperature has been initialized using
the Rampanelli and Zardi (2004) model, the inputs of which are reported in Tab. 4.

| $\gamma$ [K/km] | $\theta_0$ [K] | $\Delta\theta$ [K] | $\Delta h$ [m] | $H$ [m] |
|---|---|---|---|---|
| 1 | 288.15 | 2.0 | 100 | 550 |

**Table 4.** Inputs for the Rampanelli and Zardi (2004) model used to initialize the CNBL simulations described in the present section. In the present work, $H$ identifies the capping inversion center, while it refers to the capping inversion base in Allaerts and Meyers (2017).

The model parameter $c$ which determines potential temperature smearing across the capping inversion is set to 0.33. For the velocity, we use a uniform log-law to prescribe the initial condition, namely

$$\begin{cases} u(z) = \frac{u^*}{\kappa} \ln\left(\frac{z}{z_0}\right) & z < H \\ u(z) = \frac{u^*}{\kappa} \ln\left(\frac{H}{z_0}\right) & z \geq H \end{cases} \tag{41}$$

In case G, where geostrophic damping is not applied, care must be paid in prescribing an initial geostrophic wind consistent
with geostrophic forcing in order not to trigger inertial oscillations above the capping inversion.

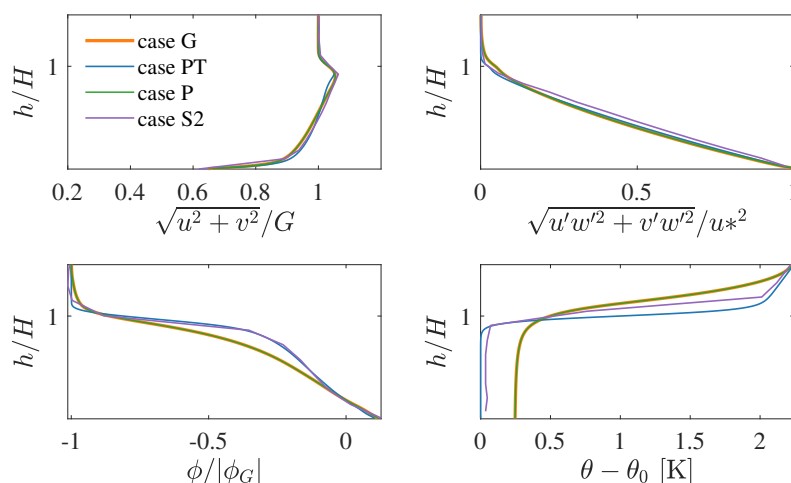

**Figure 6.** Comparison of results extracted from cases G, P, and PT against data from Allaerts and Meyers (2017). Flow statistics from cases G, P, and PT are averaged from 92800 s to 100000 s, while in Allaerts and Meyers (2017) (case S2) data are averaged from 54000 s to 72000 s.



For this reason, $u^*$ is set to $\kappa G/\ln(H/z_0)$ in case G, while it is calculated as $\kappa u_{\mathrm{ref}}/\ln(h_{\mathrm{ref}}/z_0)$ for cases P and PT. Note that, when the pressure controller is used, inertial oscillations can't be avoided since geostrophic wind is not known a-priori. All three simulations are carried out for $10^5$ s ($\approx 27.8$ h), while Allaerts and Meyers (2017) run case S2 for 20 h, gathering statistics from 54000 s to 72000 s (over the last five hours of simulated time). Geostrophic damping in cases P and PT starts at

$T_D = 6 \cdot 10^4$ s (this value is close to the oscillation period of the geostrophic wind), and it will be later shown the wind angle controller of case G stabilizes the wind angle at around $6.5 \cdot 10^4$ s. Hence, we average flow statistics from cases G, P and PT from 92800 s to 100000 s, while in Allaerts and Meyers (2017) results are averaged from 54000 s to 72000 s. In Fig. 6, we report vertical profiles of velocity magnitude, direction, shear stress and potential temperature obtained from the four different case.

As shown before in Sec. 3.2, TOSCA predicts more mixing than Allaerts and Meyers (2017) computed with SP-Wind, which results in a higher inversion height for a given set of ABL parameters. This can be observed by comparing cases G and S2, which feature the same wind-angle controller. In addition, this leads to an increased surface temperature and a different wind veer profile in results obtained from TOSCA. We note that such differences are accentuated by the fact that statistics from SP-Wind are collected at an earlier time. The difference in mixing also affects the average hub-height velocity, which differs

by 0.33 m/s between the two codes. For cases P and PT such parameter is an input, and it has been set according to results from case G. In Tab. 5, output quantities extracted from the four different simulations are reported, averaging flow statistics in the above-mentioned time intervals. The capping inversion center $H$, ground temperature $\theta_0$, inversion strength $\Delta\theta$ and inversion width $\Delta h$ are calculated by fitting the Rampanelli and Zardi (2004) model in a least-squares sense.

|     | $u_{ref}$ | $G$   | $\theta_0$ [K] | $\Delta\theta$ [K] | $\Delta h$ [m] | $H$ [m] | $u^*$ [m/s] | $q_{min}/10^{-4}$ [Km/s] | $\phi_G$ [deg] |
|-----|-----------|-------|----------------|--------------------|----------------|---------|-------------|--------------------------|----------------|
| G   | 10.871    | 12.00 | 288.33         | 1.95               | 612            | 113.5   | 0.323       | -1.04                    | -12.23         |
| P   | 10.871    | 11.97 | 288.33         | 1.95               | 612            | 113.5   | 0.323       | -1.04                    | -12.23         |
| PT  | 10.871    | 11.69 | 288.15         | 2.0                | 550            | 84.5    | 0.323       | -1.33                    | -11.48         |
| S2  | 11.200    | 12.00 | 288.19         | 1.99               | 585            | 93.8    | 0.315       | –                        | –              |

**Table 5.** ABL parameters obtained by fitting the Rampanelli and Zardi (2004) model for the CNBL cases presented in this section, together with resulting friction velocity, minimum heat flux and geostrophic wind angle. Data from cases G, P, and PT are averaged from 92800 s to 100000 s, while in Allaerts and Meyers (2017) (case S2) data are averaged from 54000 s to 72000 s.

Fig. 6, together with quantitative data reported in Tab. 5, demonstrate how the pressure controller with geostrophic damping

(case P) almost exactly matches results obtained using the geostrophic controller (case G), predicting a geostrophic wind that only differs by 0.25% with respect to $G = 12$ m/s. Fig. 6 also highlights how sensitive the ABL is to its heating history, since case PT - where the average $\theta$ profile is kept constant - predicts a lower geostrophic wind than cases G and P. In fact, it can be noticed from Fig. 7b how the inversion height is kept constant in case PT, while it grows in time during simulations P and G. For this reason, in the latter cases the ABL will experience a slightly higher amount of dissipation, which results in a small increase

in the geostrophic wind if compared to case PT. Therefore, in simulations P, G, and S2, the boundary layer is developing against



a potential temperature profile that is slowly evolving, in turn affecting the mean velocity profile. This mechanism, which is of course physical, does not reproduce what happens in real life, where the boundary layer stability evolves following the time scale of the diurnal cycle instead. As a consequence, since such temperature drift is physical but arises from an idealization, we believe that a better—and more reproducible—way of studying wind farms under specific atmospheric conditions would

be to fix the average potential temperature profile. On top of that, we also suggest driving the ABL with a pressure controller, which allows specifying the hub-height velocity, as the issue related to geostrophic inertial oscillations can be addressed using geostrophic damping. This would lead to a better agreement on wind farm power predictions and on the actual inflow conditions used in successor simulations. Results from different codes would thereby be more aligned with each other, and precursor simulations that did not run for the same amount of time could still be compared.

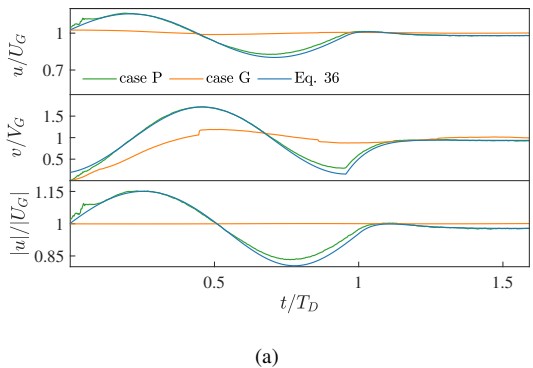

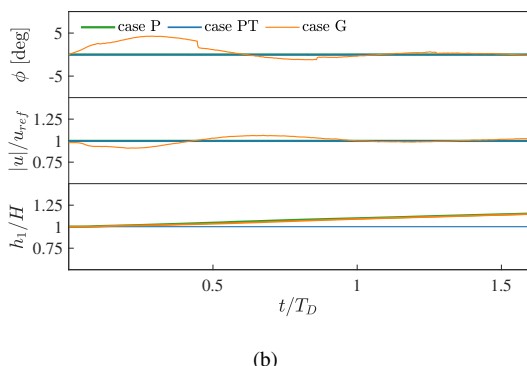

(a)                                         (b)

**Figure 7.** (a) Time evolution of average geostrophic wind (streamwise, spanwise components and magnitude from top to bottom) from cases P and G, where pressure and geostrophic controllers are used respectively. Predictions using Eq. 36 are also shown; (b) Time evolution of hub-height wind angle, hub-height velocity magnitude, and capping inversion center from cases P, G (no potential temperature control), and PT (with potential temperature controller described in Sec. 2.5.2). Time is non-dimensionalized with the start time of the geostrophic damping action $T_D$.

Fig. 7a shows the evolution of the geostrophic wind (components and magnitude), calculated as the spatial average in the homogeneous directions and at those cells where $h > H + \Delta$, produced by cases P and G. It can be seen how the developed damping technique is able to stop inertial oscillations after a time $T_D + T_{3\%}$, reaching a geostrophic wind that only differs slightly from the simulation where the geostrophic controller has been applied ($T_{3\%} = 17500$, see Sec. 2.5.1 for definition). Moreover, in Fig. 7b we report wind angle and velocity magnitude horizontally averaged at the reference height, together with

the height of the inversion center over time, evaluated by fitting the Rampanelli and Zardi (2004) model at each time step. It is evident that the pressure controller exactly maintains the wind at the desired speed and direction. Interestingly, it can be also noticed that the geostrophic controller produces small oscillations in the hub-height wind speed. These are inertial oscillations as well, but they are naturally damped by turbulence as they happen inside the boundary layer. Finally, looking at the evolution of the inversion layer height in Fig. 7b and at the final potential temperature profile in Fig. 6, it is clear that controlling the mean





potential temperature prevents the boundary layer from growing indefinitely, preserving the initial capping inversion height and the initial value of potential temperature at the ground.

## 5 Finite Wind Farm with Thermal Effects

In this section, we present results from the simulation of a finite-size wind farm consisting of 100 NREL 5MW wind turbines, aligned in 20 rows and 5 columns, with streamwise and spanwise spacing of 5 and 4.76 rotor diameters respectively. We include

thermal stratification to assess the effects of gravity waves blockage for a lapse rate of 1 K/km, a capping inversion centered at 500 m with a strength of 7.312 K. Given the large scale of the gravity waves, the numerical domain is set to $40 \times 21 \times 28$ km in the streamwise, spanwise and vertical direction respectively. All directions are graded to reach a mesh resolution of $30 \times 12.5 \times 10$ m around the wind turbines. The hybrid off-line/concurrent precursor technique described in Sec. 2.6 has been used to spin-up turbulence in the precursor, providing a realistic CNBL inflow for the successor simulation. This technique

is combined with a Rayleigh damping layer and the advection damping technique (Lanzilao and Meyers, 2022a) to ensure low reflectivity of gravity waves from the top boundary and the fringe region exit. Further details on the successor/precursor meshes and simulations, CNBL parameters and tuning of fringe, Rayleigh, and advection damping region coefficients are given in Appendix A.

Fig. 8 shows hub-height instantaneous velocity and pressure contours around the wind farm. The gravity wave footprint

inside the ABL can be clearly noticed in the pressure field, together with the small-scale pressure increase in front of each rotor. This effect is superimposed on the much larger pressure variation due to atmospheric gravity waves, which take place from the farm entrance to the exit.

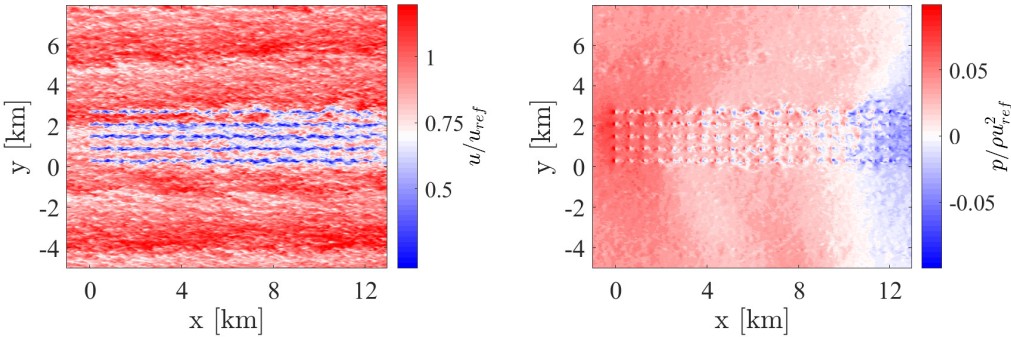

**Figure 8.** Contours of instantaneous velocity (left) and pressure (right) at the wind turbine hub-height.

Regarding the instantaneous velocity field, streamwise streaks generated by elongated turbulence structures can be appreciated. The large size of these structures is related to the high value of the prescribed equivalent roughness height $z_0$, and to the

fact that periodic boundary conditions artificially increase their length when they span the entire domain in the streamwise direction. Although these streaks do not alter the simulation results, they can make statistics convergence extremely slower. Such





an issue can be alleviated by using the so-called shifted periodic boundary conditions in the concurrent precursor simulation, where a spanwise offset is applied in the streamwise periodicity to artificially break the locking in position of such structures (Munters et al., 2016).

At any given location, we define the perturbation value of a quantity as the difference between its successor time average and the precursor time average, evaluated at the same height. Fig. 9 shows horizontal contours of pressure and temperature perturbations at the hub and inversion heights respectively. An interesting aspect is that, due to the presence of the Coriolis force, the direction of propagation of interfacial waves in the inversion layer is not aligned with the wind farm streamwise symmetry axis.

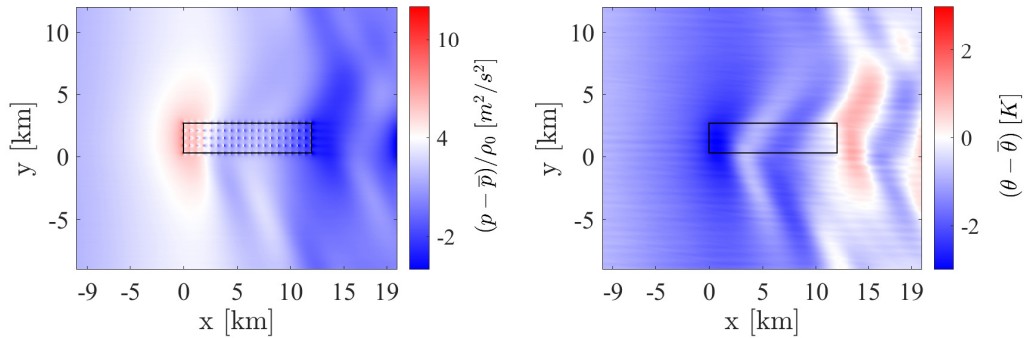

**Figure 9.** Perturbation pressure inside the ABL (left). Perturbation potential temperature at the capping inversion height (right).

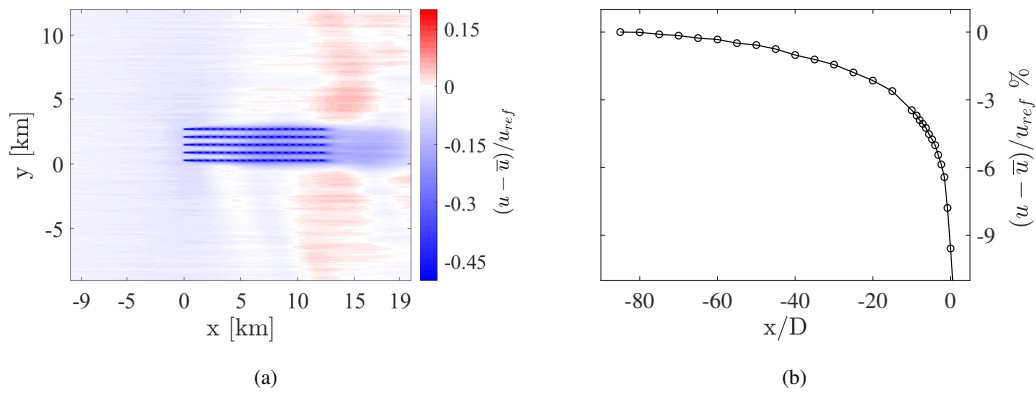

**Figure 10.** (a) Hub-height perturbation velocity; (b) hub-height perturbation velocity upstream the first wind farm row. Each upstream location is averaged along the spanwise direction within the wind farm envelope.

For instance, the two trains of waves generated by the positive and negative inversion layer displacements, at the wind farm entrance and exit respectively, have a spanwise offset, resulting in a much more complex interaction. Moreover, spurious wave interactions with their periodic images can also be noticed, but the spanwise size of the domain ensures that they happen far





from and downstream of the wind farm. Nevertheless, we are developing a lateral fringe region, which is aimed at removing this effect, where the instantaneous desired flow is reconstructed from the concurrent precursor, allowing for a smaller spanwise domain size. At this time, we do not believe wave interactions from periodic images alter the gravity wave pattern in the region near and upstream of the wind farm, which is of primary interest in order to assess wind farm blockage.

Fig. 10a shows the average hub-height perturbation velocity field, from which the effect of gravity waves on velocity can be assessed. In particular, positive perturbations are observed where negative pressure gradients are experienced and vice versa. Data from Fig. 10a, averaged along the spanwise direction and within the region enclosed by the wind farm spanwise limits, is shown quantitatively in Fig. 10b. In particular, we record velocity reductions as high as 5.80%, 2.15%, and 1.02% at 2.5, 20, and 40 diameters upstream of the first turbine row. For instance, Wu and Porté-Agel (2017) observed velocity reductions of 1.2% and 10% at 2.5 diameters upstream of the first row of an infinitely wide wind farm, for a boundary layer initialized with linear lapse rates of 1 and 5 K/km respectively.

Overall, these wave patterns agree well with linear theory (Allaerts and Meyers, 2019), and demonstrate that the developed framework and methodology allow conducting finite-size wind farm simulations, capturing gravity waves effects unaltered by spurious wave reflections from the fringe region or interaction from periodic images.

## 6 Conclusions

In the present paper, we introduced TOSCA, a new open-source LES framework for the simulation of large wind farms interacting with thermally stratified boundary layers. We validated TOSCA's wind turbine models, its ability to simulate the evolution of conventionally neutral boundary layers and to accurately predict the flow around infinite wind farms in neutral conditions. We presented a new controlling methodology for ABL precursors that allows to prescribe a desired wind speed at a reference height - located inside the boundary layer - while at the same time avoiding velocity oscillations produced by the Coriolis force in the geostrophic region above the inversion height. This approach, if combined with a potential temperature controller, allows to obtain CNBL inflow profiles that only differ slightly in the geostrophic wind between different codes, but which are characterized by the same potential temperature profile and hub-height wind speed. Conversely, using geostrophic forcing makes the hub-height velocity dependent on the amount of numerical dissipation of the adopted code, while the final temperature profile depends on both numerical dissipation and precursor simulated time. Using the proposed methodology instead would ultimately enable better agreement on wind farm power estimates using LES. We also described a new methodology for simulating finite-size wind farms under atmospheric gravity wave effects. In particular, we introduced the hybrid off-line precursor/concurrent-precursor method, where the off-line technique is used on a small domain, in order to spin-up ABL turbulence, while the concurrent method is adopted for the turbine simulation. In fact, we found that the concurrent precursor, combined with a fringe region, are crucial elements to avoid spurious gravity wave reflections while providing a realistic turbulent inflow at the same time. The off-line precursor data is used to start-up the flow field in the concurrent-precursor by means of spanwise periodization. The concurrent-precursor domain is usually bigger than required, as its size is determined by





the successor domain that runs concurrently. Hence, being able to reach steady state turbulent statistics on a smaller domain is
indeed convenient, as it makes finite wind farm simulations less computationally intensive.

        Finally, we demonstrated that TOSCA is able to capture wind farm gravity wave interactions and large scale blockage effects.
Specifically, for the CNBL simulated herein, we measured a velocity reduction of 5.80% at 2.5 diameters upstream the first
row.

In the future, we will implement shifted-periodic boundary conditions to obtain field statistics which are less dependent on
the spanwise location, and we will address the heat flux mismatch above the inversion layer.



## Appendix A: Finite Wind Farm Set-up

In this section, we describe in detail the setup of the finite wind farm case presented in Sec. 5. To avoid wave reflections from inflow-outflow boundaries, we adopt periodic boundary conditions and the concurrent precursor technique. This also provides
a suitable turbulent inflow, eliminating the wind farm wake re-advected at the inlet by the periodic boundaries. To avoid wave reflections from the upper boundary, we use a Rayleigh damping layer, while lateral boundaries are periodic. Spanwise periodicity implies that gravity waves induced by the wind farm will interact with their periodic images, requiring the domain to be sufficiently large for these interactions to happen far from and downstream of the wind turbines. Moreover, we use the advection damping technique developed by Lanzilao and Meyers (2022a) to ensure that interactions between fringe-generated
and physical waves are not advected downstream, but remain trapped inside the advection damping region.

| $z_s$ [km] | $z_e$ [km] | $\Delta z$ [m] | $N$ [-] | $f$ [-] |
|---|---|---|---|---|
| 0 | 0.4 | 10 | 40 | 1 |
| 0.4 | 0.5 | 10-4.85 | 14 | 0.94591 |
| 0.5 | 0.6 | 4.59-10 | 15 | 1.05125 |
| 0.6 | 1 | 10 | 40 | 1 |
| 1 | 3 | 10-100 | 51 | 1.04698 |
| 3 | 17 | 100 | 140 | 1 |
| 17 | 28 | 100-500 | 44 | 1.03818 |

(a) Vertical discretization parameters.

| $x_s$ [km] | $x_e$ [km] | $\Delta x$ [m] | $N$ [-] | $f$ [-] | $y_s$ [km] | $y_e$ [km] | $\Delta y$ [m] | $N$ [-] | $f$ [-] |
|---|---|---|---|---|---|---|---|---|---|
| -20 | -15.005 | 15 | 333 | 1 | -9 | -1.5 | 20 | 375 | 1 |
| -15.005 | -13 | 15-30 | 94 | 1.00748 | -1.5 | -0.5 | 20-12.5 | 62 | 0.99269 |
| -13 | 18.02 | 30 | 1035 | 1 | -0.5 | 3.5 | 12.5 | 320 | 1 |
| 18.02 | 19.97 | 30-15 | 90 | 0.9923 | 3.5 | 4.5 | 12.5-20 | 62 | 1.00805 |
| 19.97 | 20 | 15 | 2 | 1 | 4.5 | 12 | 20 | 375 | 1 |

(b) Streamwise (left) and spanwise (right) discretization parameters.

**Table A1.** Mesh information for the finite wind farm case.

The size of the successor domain is 40 km × 21 km × 28 km in the streamwise, spanwise and vertical direction respectively, discretized with 1554 × 1194 × 345 cells. All directions are graded to reach a mesh resolution of 30 m × 12.5 m × 10 m around the wind farm .





The wind farm has a rectangular planform, with 20 rows and 5 columns. The first row is located at $x = 0$, and extends
from 300 m to 2700 m. This determines a lateral spacing of 600 m (4.76 D), while streamwise spacing is set to 630 m (5
D). Wind turbines are equipped with angular velocity and pitch controllers described in Jonkman et al. (2009). A very simple
yaw controller is also added, which rotates each wind turbine independently with a uniform speed of 0.5 deg/s when flow
misalignment exceeds 1 deg. Flow angle is calculated by filtering the wind velocity at a sampling point located 1 D upstream
of the rotor center, using a time constant of 600 s. Turbines are modeled using the ADM, while tower and nacelle are not
accounted for. The projection width $\epsilon$ is set to 18.75 m.

The concurrent precursor mesh coincides with the portion of the successor domain which is located inside the fringe region.
In particular, it is 5 km × 21 km × 28 km. The mesh resolution in the streamwise direction is 15 m, while in the spanwise and
vertical directions, it is the same as the successor.

In order to save computational resources, we do not run the whole precursor simulation on the concurrent precursor mesh,
which size is determined by the wind farm and gravity wave parameters. Instead, we perform the spin-up phase on a 6 km ×
3 km × 1 km domain, characterized by a resolution of 15 m × 15 m in the streamwise and spanwise directions. The vertical
direction is discretized in the same manner as the successor in order to increase the resolution inside the capping inversion
layer. This spin-up phase is carried out for $10^5$ s, after which an inflow database is collected. The generated inflow database
is then used to start-up the solution in the concurrent precursor and successor domains. This technique, which we refer to as
the hybrid off-line/concurrent precursor, is explained in Sec. 2.6. In the successor, after a spin-up of 5000 s, corresponding to
slightly more than one flow-through time, data are gathered for 15000 s.

The off-line precursor simulation uses the pressure and temperature controllers described in Sec. 2.5, while in both the
concurrent precursor and successor simulations, velocity is controlled using a constant source term, obtained by averaging
the off-line precursor source from 100000 s to 120000 s. The temperature controller is retained in the concurrent-precursor
simulation, but it is switched off in the successor so that the inversion height is free to be perturbed by the wind farm.

CNBL parameters used for the off-line precursor are summarized in Tab. A2. They are calculated based on the sensitivity
analysis performed in Allaerts and Meyers (2019). In particular, our objective is to choose a set of non-dimensional parameters
such that the capping inversion layer is strongly perturbed by the wind farm. This results in a capping inversion Froude number
of $Fr = 0.94$ and an internal wave parameter of $P_N = 3.02$. These non-dimensional groups are related to the physics and
magnitude of interfacial waves inside the inversion layer and internal gravity waves above the ABL, respectively.

| $u_{ref}$ [m/s] | $h_{ref}$ [m] | $\theta_0$ [K] | $\Delta\theta$ [K] | $\Delta h$ [m] | $\gamma$ [K/km] | $H$ [m] | $f_c$ [1/s] | $z_0$ [m] |
|---|---|---|---|---|---|---|---|---|
| 9.0 | 90 | 300 | 7.312 | 100 | 1 | 500 | $9.6057 \cdot 10^{-5}$ | 0.05 |

**Table A2.** ABL parameters used for the finite wind farm simulation presented in this section.

In Fig. A1 we show vertical profiles of wind speed magnitude, inflow angle, non-dimensional shear stress, and potential
temperature, averaged over the last 15000 s from the concurrent precursor domain. It can be noticed how the pressure controller
accurately maintains the desired wind speed and direction at $h_{ref}$, and how the temperature controller removes the ground





temperature shift observed in the previous sections by keeping the average profile constant in time. As a consequence, inversion height and strength are maintained equal to their initial values (Tab. A2), while the resulting friction velocity corresponds to 0.432 m/s.

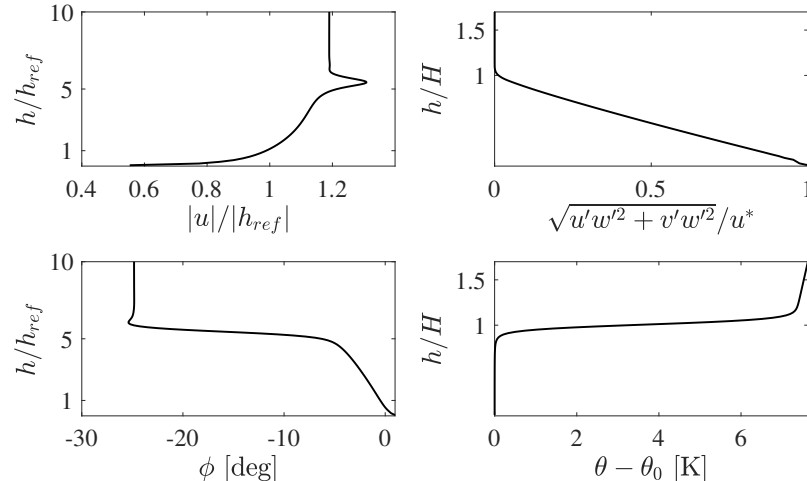

**Figure A1.** Precursor inflow data, averaged from the concurrent precursor domain from 105000 s onward. We only show a subsection of the domain in the vertical direction corresponding to $10 \cdot h_{ref}$. All profiles are uniform above, except from the potential temperature profile which exhibits a linear increase equal to the lapse rate $\gamma$.

Regarding the successor case, we followed the approach of Lanzilao and Meyers (2022a) to chose the damping layer and fringe coefficients. In particular, after a reflectivity study that employed a computationally cheap canopy model (not shown 755 here) we found that a Rayleigh damping coefficient of $\nu_{RDL} = 0.05$ and a fringe damping coefficient of $\nu_{FR} = 0.03$ yielded minimal gravity waves reflectivity. The dominant vertical wavelength of the gravity waves is estimated as $\lambda_z = 2\pi G/N \approx 11.8$ km (Allaerts and Meyers, 2017), where $N$ is the Brunt-Vaisala frequency ($N = 5.72 \cdot 10^{-3}\ s^{-1}$ based on parameters listed in Tab. A2), and $G = 10.815$ m/s. We ensure that at least one $\lambda_z$ is contained in the Rayleigh damping layer by setting its width to 12 km. Regarding the advection damping technique developed by Lanzilao and Meyers (2022a), we observed that their 760 guidelines in how to choose the length of the advection damping region did not apply to our case, which is characterized by a very strong inversion layer and a shallow boundary layer. In fact, we believe that a key parameter that needs to be tuned in order to avoid spurious gravity wave interactions is the length of the region where advection damping is applied after the fringe. This holds in particular for sub-critical ($Fr < 1$) cases, where waves inside the capping inversion can propagate against the flow. Here, perturbations would be propagated upstream from the wind farm to the fringe exit, being suddenly forced to 765 obey the precursor inflow inside the fringe region. Such a sharp change in the boundary layer displacement at the fringe exit induces spurious gravity waves which remain trapped at their streamwise location if horizontal advection of vertical velocity is turned off. Nevertheless, these waves would interact with physical waves from inside the domain, resulting in more spurious interactions. As a consequence, it is crucial to ensure that all spurious interactions generated by this mechanism are fully con-



tained within the advection damping region and are not advected downstream.


| $x_s$ [km] | $x_e$ [km] | $\Delta_s$ [km] | $\Delta_e$ [km] |
|---|---|---|---|
| −20 | −15 | 1 | 1 |

(a) Fringe region parameters.

| $x_s$ [km] | $x_e$ [km] | $\Delta_s$ [km] | $\Delta_e$ [km] |
|---|---|---|---|
| −18 | −11 | 1 | 1 |

(b) Advection damping region parameters.

**Table A3.** Fringe and advection damping region information.

We used the same damping functions as Lanzilao and Meyers (2022a), and in Tab. A3 their parameters are reported for our finite wind farm simulation.

**Appendix B: TOSCA Parallel Scaling**

In this section, we show TOSCA's strong and weak parallel performance by running CNBL simulations with an increasing
number of Niagara nodes for three different mesh sizes. The simulation setup corresponds to the off-line precursor described in Appendix A. The different meshes are evaluated by systematically doubling the number of elements in each direction, starting from $300 \times 300 \times 100$ cells. As a consequence, they consist of 9M, 72M, and 576M elements in total. Tab. B1 reports the number of nodes for each run, which only consisted of two hours of wall-clock time.

| | CNBL 9M | CNBL 72M | CNBL 576M |
|---|---|---|---|
| | 5 | 10 | 100 |
| | 2 | 10 | 200 |
| Number of Niagara Nodes | 4 | 20 | 400 |
| | 8 | 40 | 800 |
| | 16 | 80 | - |

**Table B1.** Scaling tests performed on Niagara Compute Canada cluster, each node consists of 40 CPUs.

Tests have been performed on Compute Canada's Niagara cluster, which consists of 2024 nodes, each with 40 Intel "Skylake"
cores at 2.4 GHz or 40 Intel "Cascade Lake" cores at 2.5 GHz. Node interconnection consists of an EDR Infiniband network, organized in a "Dragonfly+" topology with five dragonfly wings. Fig. B1 shows the time per iteration as the node count increases for each of the CNBL meshes. TOSCA's strong scaling performance remains close to linear until roughly 25k cells per core are reached, which we identify as a reasonable trade-off between efficiency and speed. TOSCA was also successfully run at-scale on the entire Niagara cluster to simulate a finite wind farm on a mesh exceeding 1 billion elements, proving
TOSCA's suitability for massively parallel computations.





We also highlight that the time per iteration does not reflect the actual speed at which the simulation advances in time, as the time step size depends on the numerical method. In implicit methods like the Newton-Krylov solvers employed by TOSCA, the computational cost of each time step depends on the time-step size, whereas these quantities are unrelated in explicit methods. Nevertheless, implicit methods are able to advance in time with a Courant-Friedrichs-Lewy (CFL) number greater than one 790 (we used 0.9 for these analyses), while explicit methods are usually limited to a value close to 0.5.

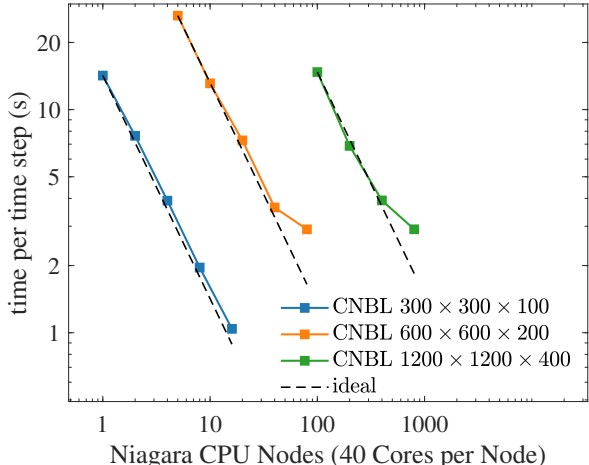

**Figure B1.** TOSCA strong scaling performance on Compute Canada's Niagara cluster. All simulations share the same setup and only differ in the number of mesh elements.

Regarding actuator models, their solution and I/O operations are also parallelized in TOSCA. Specifically, we define a sphere of cells, for each wind turbine, that contains all cells that the rotor can possibly intersect when yawing. Processors owning mesh cells belonging to the sphere are then grouped into turbine-specific sub-communicators, which are used to solve wind turbines simultaneously. Hence, provided that a sufficiently high core count is used, wind turbine update time in TOSCA is independent 795 of the number of wind turbines in the simulation, and each communicator can write turbine data to file simultaneously. For the finite wind farm simulation presented in Sec. 5 (100 wind turbines), the turbine update time was less than 0.1 s. Individual turbine update time depends on cell size and on actuator point-processor ownership search (which processor in the communicator controls which actuator point). The latter is only triggered by a change in yaw for the ADM and uniform ADM models, while it has to be performed at every iteration for the ALM, as actuator points are physically rotating.



*Code availability.*  TOSCA is available at https://osf.io/q4vaf/, DOI 10.17605/OSF.IO/Q4VAF

*Author contributions.*  Conceptualization, S.S, D.A., J.B.; methodology, S.S.; software, S.S, A.A.; validation, S.S.; formal analysis, S.S.; investigation, S.S.; computational resources, J.B.; data curation, S.S.; writing–original draft preparation, S.S.; writing–review and editing, J.B., D.A.; visualization, S.S.; supervision, J.B., D.A.; project administration, J.B.; funding acquisition, J.B.. All authors have read and agreed to the published version of the manuscript.

*Competing interests.*  No competing interests are present.

*Acknowledgements.*  The present study is supported by UL Renewables and the Natural Science and Engineering Research Council of Canada (NSERC) through Alliance grant no. 556326. Computational resources provided by the Digital Research Alliance of Canada (www.alliancecan.ca) and Advanced Research Computing at the University of British Columbia (www.arc.ubc.ca) are gratefully acknowledged.





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
