# Peer review of "TOSCA - An Open-Source Finite-Volume LES Environment for Wind Farm Flows"

_Wind Energy Science, 2023_

## Author Response (AR1)

**Replies to First Reviewer**

**We thank the reviewer for their excellent comments. Our response, denoted as AUT, is shown below. Modified text of the paper is shown in *italic* between quotes, while the reviewer's comments are denoted as REV.**

**REV**: I have read the manuscript with great interest. The manuscript and figures have been carefully prepared. The study introduces a new simulation code to the community and new simulation strategies that can help study the effect of gravity waves and wind farm blockage. The results for the wind farm in section 5 are not directly compared to related literature references. Given that the manuscript already discusses so many aspects, that is logical [the corresponding section in the introduction needs improvement; see below]. Statements on the benefit of the new control methods should be made more carefully [as now these statements are very bold] as indicated below. The ideas behind the presented controls are very interesting and the manuscript clearly outlines why this needs to be considered. Overall, this is an excellent and very important study for the community and certainly qualifies for publication in Wind Energy Science. Below I prepared a list of questions/suggestions/recommendations. I hope this helps the authors to improve the manuscript.

**REV**: comment 1, line 15: "The proposed methodology overcomes these issues by ensuring that the
same hub wind and thermal stratification are used in successor simulations, regardless of the adopted code or precursor run time." --> This should be formulated more carefully; see also below. Also, with the proposed methodology, there can be differences between codes.

**AUT**: rephrased to "*The proposed methodology overcomes these issues by ensuring that inflow conditions produced from different codes feature the same hub wind and thermal stratification, regardless of the adopted precursor run time*". Although the shape of the velocity profile, the wall shear stress, the minimum heat flux or the geostrophic wind can differ slightly between codes, hub wind and potential temperature profile are exactly the same, as they are controlled variables. We rephrased the sentence so that it specifies that these variables are the same in the inflow condition, while they can differ slightly in the successor due to the interaction between fringe region and gravity waves, as mentioned later in the paper. We also highlight that such methodology is more focused on the wind farm rather than the CNBL comparability. In fact, ensuring the same hub wind yields very similar wind turbine power, at least at the first row, while the same temperature profile yields the same gravity waves physics, which has been demonstrated to influence the wind farm operation.

**REV**: comment 2, line 30: references in this section must be checked for accuracy and relevance. Some of these studies consider wind farm blockage, not just turbine-induced blockage, as suggested. Gribben and Hawkes does not seem to be a peer-reviewed publication.

**AUT**: we removed Segalini and Dahlberg 2019, as it is an experimental campaign that studies wind farm blockage as a function of the wind farm layout. This study may not be relevant for the present paper. Moreover, we moved those studies that introduce linear superposition of individual effects by rephrasing the sentence slightly. Regarding Gribben and Hawkes, the reviewer is right to say that it is not a peer reviewed article, but the Rankine half body is now an industry standard in firms such as UL Renewables and RWE, so we believe the study is relevant. We hope that the reviewer will agree on this aspect.

**REV**: comment 3, line 33: no reference is given.

**AUT**: if the reviewer refers to the superposition of individual effects, now the reference has been added. For the following sentence: "*However, recent studies suggest that this could underestimate --- if not totally misrepresent --- wind farm-level blockage, which is heavily influenced by the mutual interaction between the wind farm and the density-stratified atmospheric boundary layer (ABL)*" references are given at the end of the sentence.

**REV**: comment 4, line 35: same as line 30. Make sure the correct references are cited in the correct place.

**AUT**: the references Smith, 2010; Wu and Porté-Agel, 2017; Allaerts and Meyers, 2017, 2018, 2019; Centurelli et al., 2021 all point to the conclusion outlined above, i.e. that wind farm-level blockage could be underestimated by applying superposition of individual effects alone. All these studies suggest that wind farm – ABL interaction has a leading role in the blockage produced by large wind farms.

**REV**:comment 5, line 40: Anderson, 2009; Openwind; are not scientific publications. Within the context of this paper, you may not need so many references here.

**AUT**: these references have been removed.

**REV**: comment 6, line 69: the selected references are not the common SOFWA references, in particular Matiz-Chicacausa and Lopez, 2018 and Zhang et al., 2022.

**AUT**: these two references have been removed and replaced with two more cited references (Churchfield 2012a and 2012b).

**REV**: comment 7, line 74: better to replace the reference to one of the new grand challenges papers.

**AUT**: rephrased using "Scientific challenges to characterizing the wind resource in the marine atmospheric boundary layer", Shaw et al. 2022, Wind Energy Science.

**REV**: comment 8, line 82/83:  the results indicate some artificial effects are still observed. The statement "avoids artificial wave reflections" is too strong. Effects in the top of the domain by design are limited, but some effects are still visible in the remainder of the domain.

**AUT**:  substituted "*avoids*" with "*limits*", which is indeed more correct. Regarding some other effects pointed out by the reviewer, we provide some more explanation below:

1. If the reviewer refers to the wave images from the periodic spanwise sides of the domain, we note that these are not reflections but interactions with waves that are physical – in the sense that they are a physical result of the actual simulation setup – as they are produced by the wind farm images mirrored w.r.t these boundaries. To remove these waves completely, a larger domain would be sufficient.
2. Conversely, wave reflections are introduced by the fringe region and by the Rayleigh damping region. These are not physical as energy cannot be radiated back to the wind farm in reality. Such spurious effects will be always present to a certain extent, as the Rayleigh damping and fringe region are proportional controllers, i.e. an error has to exist for the damping to be actually different from zero. In practice, there is no way to avoid reflections and some wave energy will always be reflected inside the domain. Moreover, the fringe region itself triggers gravity waves as it displaces the BL back to its original state, and also this effect is unavoidable. To make sure that these do not interact with the physical ones we

use the advection damping region, which we found in later studies to be crucial, especially when a strong capping layer is present. By killing the horizontal advection of vertical velocity, these spurious perturbations are not transported, thus they cannot interact.

**REV**: comment 9, line 98: does this mean TOSCA only works with "openMPI" or would it also work with other MPI libraries? I assume the latter.

**AUT**: yes, although TOSCA has only been tested with the OpenMPI implementation of the MPI standard.

**REV**: comment 10, line 107: the references "(Loken et al., 2010; Ponce et al., 2019)" are not about the Niagara system and should be removed.

**AUT**: we refer to the following web page https://docs.scinet.utoronto.ca/index.php/Acknowledging_SciNet, where these are stated as the publication to be used to acknowledge Niagara and the SciNet datacenter.

**REV**: comment 11, in section 2.4: the manuscript compares with Martínez Tossas et al. (2012). However, an updated version of this work is provided at https://doi.org/10.1002/we.1747. Please
update the reference accordingly. Also, please check whether the results have been updated because some numbers have been changed compared to the 2012 version.

**AUT**: we thank the reviewer for having brought this aspect to our attention. Velocity plots seem exactly the same, while power vs grid spacing plots have been changed. In particular, turbine powers from OpenFOAM are higher now and they compare better with TOSCA. Also, the reference has been changed to Martinez Tossas 2015.

**REV**: comment 12, line 378: the manuscript compares with Martínez Tossas et al. (2012). However, an updated version of this work is provided at https://doi.org/10.1002/we.1747. Please update the reference accordingly. Also, please check whether the results have been updated because some numbers have been changed compared to the 2012 version.

**AUT**: the sentence has been rephrased according to the reviewer comment to "*This could be beneficial for example when making comparisons between different codes, [...]*".

**REV**: comment 13, figure 1: concerning the introduced inflow generation method. The only aspect discussed is the spanwise periodicity, for which it is claimed that it can be removed completely by the Munters 2016 method. However, if the domain is as small as indicated in Figure 1 there could be additional effects as the large-scale flow features in the boundary layer are not fully resolved (see, for example, the work by Javier Jiménez and coworkers). So there would be limitations in size.

**AUT**: yes, there are limitations in size for CNBLs, but our argument is that, for large wind farms, the limitations on the domain imposed by the gravity waves and by the farm itself are much bigger than the ones that are required to resolve a CNBL. When using the concurrent precursor technique, precursor and successor should have the same spanwise and vertical size, so performing the spin-up on a precursor as big as the successor would be a waste of resources if a smaller domain – which still resolves the CNBL – could be used. Our method decouples the two domains, allowing to perform the spin-up with the domain size that is suitable to resolve the CNBL, and the successor phase on a different domain that suits the requirements of the wind farm – atmospheric interactions.

**REV**: comment 14, line 441-444: : the formulation, as is, suggests atmospheric gravity waves are not real flow physics, which is not what the authors want to convey.

**AUT**: the spin-up phase and, in general, the precursor phase, do not contain gravity waves as there is no perturbation to the CNBL (there are small gravity waves, generated by the turbulent structures that hit the inversion layer, but they are negligible). For this reason the requirement on the domain size is much more computationally feasible at this stage. In any case, the paragraph has been rephrased to make this concept more clear, namely "Nevertheless, the proposed hybrid method [...] is very convenient as it allows to reduce the overall computational cost of the ABL spin-up phase,where wind farm-induced gravity waves are not yet present. In fact, this initial phase is run on a domain whose size is dictated by the current flow physics, rather than on quantities that will only become relevant at later simulation stages".

**REV**: comment 15, section 3.1: compare to https://doi.org/10.1002/we.1747 instead of the 2012 version.

**AUT**: the reference and data have been modified according to the new version.

**REV**: comment 16, line 462: what is the azimuthal direction here?

**AUT**: this is the discretization along the direction that is tangential to the disk perimeter. This is an angular discretization, as its actual size changes with radius, that is why sometimes is referred to as the azimuthal discretization.

**REV**: comment 17, Table 1: double-check the numbers in the table.

**AUT**: updated to Martinez-Tossas 2015 as suggested by the reviewer.

**REV**: comment 18, line 482: "*while TOSCA's ALM predicts a slightly higher power*". --> The difference is nearly 15%, certainly more than "slightly higher". One would expect a closer agreement since this concerns a uniform inflow case. Perhaps also compare to more recent simulation studies of that case performed with different codes, see e.g. https://doi.org/10.1063/1.5004710, https://doi.org/10.1002/we.2714.

**AUT**: both data from TOSCA and OpenFOAM have been changed. The latter has been replaced with data from Martinez-Tossas 2015 paper, which show a higher power for the ALM. TOSCA data have been initially calculated without excluding the initial transitory by mistake. Now data have been averaged so that the initial transient of the flow and turbine controller is avoided, as stated in the paper. Data are indeed closer (6% difference w.r.t. the average between the four powers for the ALM).

**REV**: comment 19, line 519: the reference is incorrect, as Chamorro and Porté-Agel (2010) only consider the single turbine case.

**AUT**: reference has been changed to Chamorro and Porté-Agel (2011) throughout the paper, this was a typo.

**REV**: comment 20, line 586: what exactly is the model parameter c (perhaps I missed the definition)?

**AUT**: this parameter is a very important one within the Rampanelli and Zardi model (defined after Eq. 13 in their paper), as it essentially controls how sharp the capping inversion layer is. As one can imagine, very different results can be obtained by solely changing c. For this reason, we believe it is important to specify the choice of such parameter when detailing the initial condition of the simulation. In our study, we set it to 0.33 as Rampanelli and Zardi suggest 1/3 as a suitable value for CNBLs.

**REV**: comment 21, figure 6d: is the S2 case supposed to look like the case PT? This is unclear from the text. The temperature profile for case PT looks like it is intended.

**AUT**: we rephrased the paragraph to make it a bit clearer: "This results in a higher inversion height for a given set of ABL parameters, and can be observed by comparing cases G and S2, which feature the same wind-angle controller, but which differ in the obtained profile of potential temperature. This leads to an increased surface temperature predicted by TOSCA and a different wind veer profile between the two codes. Although we note that such differences are accentuated by the fact that statistics from SP-Wind are collected at an earlier time, i.e. the CNBL has grown by a lower extent, case S2 seems to be more aligned to case PT, where the average potential temperature profile is kept constant by the controller. The difference in mixing between the two codes also affects the average hub-height velocity, which differs by 0.33 m/s between case G and S2".

This means that case S2 should be exactly the same as case G if SP-Wind and TOSCA would be the same, as these two cases feature the same controller. Conversely, case P and PT feature the pressure controller with geostrophic damping, which is slightly different. One aspect of notice, also mentioned in the text, is that statistics from S2 have been collected at an earlier time, hence the CNBL had less time to mix (lower ground temperature, lower inversion height etc). The fact that case S2 and PT agree simply means that SP-Wind predicts less mixing than TOSCA, as in SP-Wind the potential temperature profile only moves slightly w.r.t to case PT, where it is forced to be constant in the spatial average, equal to its initial condition. This even more highlights the need to use a temperature controller when running precursor simulations as, by the time that turbulence has developed and the CNBL has formed, the potential temperature profile has shifted from its initial shape.

To sum up, approaches G and P are almost exactly equivalent if applied to the same code base, while the use of the temperature controller allows to compare different code bases. If one doesn't want to control temperature but desires to quantify the amount of mixing between different codes, the comparison has to be made by starting and ending the averaging at the same time. If this doesn't happen, potential temperature evolution could be confused with more or less mixing in the code.

We hope this clarifies the reviewer comment/doubt.

**REV**: comment 22, figure 6: one can see that the different controls have some effect on the flow.

**AUT**: the effect is mainly due to the temperature controller, which affects the heating history (cases P or G vs case PT). Conversely, the different velocity controllers (P or G) can be used interchangeably. Controller G is preferred when the geostrophic wind is to be controlled, while controller P + geostrophic damping is better suited to control the wind inside the boundary layer.

**REV**: comment 23, around line 620 : keeping the temperature profile fixed is also an idealization. I agree there are reasons to do so, but keeping the temperature profile fixed also changes boundary layer dynamics as the way buoyancy is represented in simulation changes a bit

**AUT**: we do not believe that fixing the temperature profile is more neither less physical than not doing so. It is only more reproducible as it rules out the dependency of its evolution on the adopted code, numerical discretization and turbulence closure both in the momentum as well as potential temperature equations. This applies to Boussinesq based solvers; we didn't investigate the effect of fixing potential temperature e.g. to anelastic formulations.

**REV**: comment 24, line 623: "*Results from different codes would thereby be more aligned with each other*" ==> This is not shown and may depend on the quantity of interest.

**AUT**: rephrased to "This would lead to a better agreement on wind farm power predictions and on the actual inflow conditions used in successor simulations, allowing to compare precursor simulations that did not run for the same amount of time."

**REF**: comment 25, figure 7: one can see some oscillation also remains towards the end of the simulation, which is why statements that all oscillations are completely removed should be adjusted.

**AUT**: referring to Fig. 7, if the reviewer refers to oscillations in the G case, this does not feature geostrophic damping for the reason mentioned in the paper (i.e. the geostrophic wind is known a-priori and it is possible to provide a consistent initial condition), hence oscillations are present, especially within the boundary layer where the initial condition cannot be consistent (these are small as they are limited by turbulence which acts as a damping term). Conversely, case P does not show any noticeable oscillations after the damping time mentioned in the paper.

**REV**: comment 26, line 627: "*differs slightly*" --> please define.

**AUT**: Rephrased to "*differs by 0.25%*". See Tab. 5, geostrophic wind (G) incases G and P is 12.00 m/s (prescribed) and 11.97 (calculated) m/s respectively.

**REV**: comment 27, line 656: "*Although these streaks do not alter the simulation results*" --> probably depends on what quantities are considered. The periodicity of the flow in a spanwise direction is visible in Figure 8

**AUT**: rephrased to "*If averages are gathered for a sufficient amount of time, these streaks do not alter the simulation results from the wind farm performance point of view. Nevertheless, as statistics convergence can become extremely slow, this issue can be alleviated by using the so-called shifted periodic boundary condition [...]*". We believe that the periodicity in the streaks does not alter any average results if averages are taken for a sufficient amount of time. We also carried out our simulations further forward in time to ensure such statement. Another way to remove the streaks is to increase the spanwise size of the fringe region, so that they do not remain locked in position between the inlet and outlet of the concurrent precursor. As mentioned in the paper, we also think that such phenomenon is enhanced by the choice of equivalent roughness height, which is higher than other studies.

**REV**: comment 28, figure 9: can you comment on the origin of the oscillations shown in Figure 9b?

**AUT**: these gravity wave patterns are free of spurious (i.e. non-physical) waves. The waves that seem reflected are instead waves interactions produced by the fact that the side boundaries are periodic. Hence, to the eyes of our LES set-up, it is as if many domains were staked laterally, and wave trains that are exiting from one side are re-entering from the other side. However, it is

important to highlight that these are wave transmissions, not reflections, across the periodic boundary, not a numerical artifact such as gravity waves reflecting due to a boundary condition (which would happen for example with a zero-gradient BC). Moreover, there isn't any attempt in current state of the art wind farm LES in avoiding such lateral interactions, as they do not affect wind farm operation if they happen far and downstream with respect to the wind farm. That is why we chose a domain which is fairly large in the spanwise direction, allowing waves to be close to the outlet when they reach the domain sides. By doing so, their images will re-enter the domain already close to the outlet and far downstream the wind farm

**REV**: comment 29, figure 10: what do the symbols in Figure 10b represent?

**AUT**: they are sample probes, for this reason they are finer where we expect the blockage to have a high gradient, transitioning from global to local induction. We removed the symbols as they are misleading.

**REV**: comment 30, line 676: as different cases are considered, these numbers cannot be compared one-to-one.

**AUT**: We removed the reference to Wu and Porté-Agel, which were provided for reference. Another aspect to notice is that their results might be affected by high wave reflectivity, as the vertical domain size, set to 2.4 km in their simulations, was insufficient to vertically resolve the gravity waves, characterized by an estimated vertical wavelength of 10 km and 4.8 km for the weak and strong lapse rates respectively.

**REV**: comment 31, line 679: "*Overall, these wave patterns agree well with linear theory (Allaerts and Meyers, 2019)*" --> This is not shown in the manuscript. This statement should either be removed or made more specific.

**AUT**: this was a qualitative comparison that can be assessed e.g. from Allaerts & Meyers 2019, but we removed the whole sentence for simplicity, as the case shown in Allaerts & Meyers 2019 is run with different CNBL parameters and could be confusing.

**REV**: comment 32, figure A1: the low-level jet looks quite sharp. Is there any reason for that?

**AUT**: it may be a consequence of the chosen CNBL input parameters. In this case, the vertical resolution has been refined to 4.6 m within the inversion layer to capture the Ellison scale.

**REV**: comment 33, line 757: "Brunt-Vaisala frequency" add accents.

**AUT**: accents added.

**Replies to Second Reviewer**

**We would like to thank the reviewer for the time dedicated to revising the paper. We proceed with answering and clarifying, where possible, the proposed comments.**

**Our response, denoted as AUT, is shown below. Modified text of the paper is shown between quotes, while the reviewer's comments are denoted as REV.**

**REV**: The manuscript provides a detailed overview of a new LES model for wind energy applications based on finite volume discretization. The manuscript also includes presentation of model tests for simulation of conventionally neutral boundary layer (CNBL) as well as simulations of an isolated wind turbine rotor and a wind plant. The simulation of the CNBL is compared with previous studies that utilized three different LES models. The model performance in reproducing CNBL is comparable to that reported in previous studies. The simulation of an isolated wind turbine rotor is comparable to that reported by Stevens et al. (2018).

**AUT**: We would like to clarify that the isolated wind turbine is not compared against Stevens et al. (2018), but rather against Martinez Tossas 2015 (it was compared against the 2012 preprint in the manuscript, but that has been already changed and the data updated, according to comments from the first reviewer). Furthermore, there are two more studies in the paper, not mentioned by the reviewer, which are in our opinion relevant. First, we compare an infinitely wide wind farm without thermal stratification against numerical results from Stevens et al. (2018) and experimental results from Chamorro and Porté-Agel (2011), showing that TOSCA agrees well with both previous numerical models and against wind tunnel measurements. Secondly, we simulate a finite wind farm under CNBL (neutral stratification within the BL and stable aloft, with an inversion layer), quantifying blockage for this specific scenario, providing valuable details on the setup of this kind of simulations, that are useful in order to avoid contaminating the results by unphysical wave reflections. In particular, we provide considerations on the domain size, damping regions and propose a novel precursor methodology that allows to resolve gravity waves at a lower computational cost. We also note that—to the best of our knowledge—prior to our study, resolving gravity waves in an LES of a wind farm has only been achieved by the group of Meyers, which uses the non-open source SP-Wind code. For this reason, we believe that showcasing this capability of TOSCA while at the same time providing the source code used to achieve it, are two aspects that strengthen the novelty and importance of the publication.

**REV**: The authors present several flow field controllers that enable specifying stationary flow conditions. They introduced a novel hybrid off-line precursor/concurrent-precursor method for generation of turbulent inflow for non-periodic streamwise flow conditions. In addition to the LES model itself this is the novel component of the current study.

**AUT**: We are glad that the reviewer found this aspect of the paper useful. Nonetheless, we believe that in addition to the methodological advances, our LES results showing gravity waves triggered by a finite wind farm are also novel both due to the scarcity of similar studies in literature and because the code used to produce them is made available to other researchers.

**REV**: The presentation of the new open-source LES model and the off-line precursor/concurrent-precursor methods are focused on model development and that brings two questions:

First, the focus on the manuscript is on model development and not a well formulated scientific question, suggesting that this manuscript is better suited for a model-focused journal, like

Geoscientific Model Development, rather than Wind Energy Science which expects exploration of a scientific question.

Second, the motivation for the development of a new model is to provide an open-source platform, and that is a noble goal. However, considering that there are numerous other open-source models for wind energy applications like SOWFA, PALM, and Exawind/AMRwind (as mentioned in the manuscript), it is not clear what is clear distinguishing characteristic of the TOSCA model other than it is a finite-volume model. It is stated that SOWFA is not "sufficiently parallel-efficient," but no evidence is given for that statement. No comparison in performance between TOSCA and other models is provided.

Third, the authors ignore a significant body of work focused on creating realistic simulations of wind turbine and wind plant flows including comparisons with observations (e.g., Mirocha et al. 2014, Aitken et al. 2014, Marjanovic et al. 2017, Arthur et al. 2020, Sanchez Gomez et al. 2023). Rather than include comparisons with observations, the authors instead opt for creating different flow controls to achieve stationary conditions, without inertial oscillations or wind veer, essentially creating idealized conditions that are not commonly observed in the atmosphere. Better justification of new idealized scenarios would be required.

Finally, there are a number of imprecise or inaccurate statements made that are listed below, under Specific Remarks.

Taking all the above into account, Wind Energy Science journal may not be appropriate journal for publication of the work primarily focused on model development and idealized simulations. Perhaps Geoscientific Model Development journal would be more appropriate, but only if the numerous issues regarding inaccurate statements about the state of the science addressed and modeling choices better justified.

**AUT**: The reviewer's opinion that the paper is primarily focused on model development is fair. Nonetheless, we note that Wind Energy Science has published numerous studies that primarily report on model development, such as Martinez-Tossas et al. 2021; Bradstock and Schlez 2020; Blondel, F. and Cathelain 2020; to name a few.

The research question that motivated the development of TOSCA is detailed in the introduction. Specifically, as wind farms are getting larger and closer to each other, there is a need of better understanding blockage effects and cluster wakes. This is a complicated task which cannot be accomplished using only observation data. As a consequence, the need for high-fidelity models is always present, as these are run in a more controlled environment and data are more easily available at the desired locations and in the desired time window. However, the available high-fidelity models are not uniformly capable of capturing the physical phenomena involved in wind farm blockage, slowing scientific advance. We argue that our paper adds to the scientific inquiry of wind farm blockage by (i) contributing an open-source tool that is specifically tailored for investigating these phenomena, and (ii) suggesting a consistent method for initializing wind farm simulations.

Regarding other open-source codes, we agree with the reviewer that at least three more platforms are available. Regarding SOWFA, we mention that this is not well-suited for large wind farm simulations as it is based on OpenFOAM, a general-purpose code with limited performance at scale. This is due to its intense I/O, which results in the generation of a massive number of files when running on thousands of cores (one folder per processor with several field files). We are also aware that OpenFOAM features a "collated" I/O option, but this definitely is too slow for large size problems, as it features multiple MPI_AllReduce calls for every field writing. We do not want to

include a scaling performance comparison in our paper, as a fair comparison would require the same HPC platform and case. Besides, the NREL itself acknowledged such limitations as it stopped developing SOWFA and it is now concentrating on the ExaWind project. While ExaWind may become a leading tool in the future, its development is still at an initial phase. It does not feature an IBM method and, as well as PALM, lacks a concurrent precursor method, which we found to be essential if one wants to at the same time prescribe a realistic turbulent inflow and damp gravity wave reflections, as mentioned in the paper. TOSCA is at the moment the only finite-volume open-source code that allows both inlet/outlet (off-line) precursor field mapping and concurrent-precursor technologies. Making it to our knowledge the preferred open-source tool when the objective is to capture gravity waves around a large wind farm.

Nevertheless, to make this concept clearer, we rephrased to "In addition, although SOWFA has been used in several research studies in the last decade […], it is not sufficiently parallel-efficient when running with thousands of cores, as the number of produced files increases drastically with processor count. While some of these shortcomings have been addressed and solved in the NREL Exawind project […] the latter is not yet at a production stage. Moreover, all SOWFA, PALM and Exawind platforms do not feature the concurrent precursor technique, making TOSCA the only finite-volume open-source code to possess such capability."

Regarding the third comment of the referee, we would like to point out that our CNBL simulations do feature wind veer, as the Coriolis force is considered in the governing equations (not in the infinite wind farm and isolated wind turbine case, but those cases are matching other validation data which didn't model wind veer). Secondly, the velocity and temperature controllers are models to represent existing phenomena such as large-scale horizontal pressure gradients and subsidence respectively. Our choice to simulate a non-evolving flow (mainly in terms of BL height) is explained in the paper, and we believe that this is not more idealized than running a precursor for big amount of time without temperature control, as in reality CNBLs are only observed for short lapses of time, especially in transition between night and day. Without temperature controller, by the time turbulence has developed and reached a statistical steady state, the BL height has increased and the inlet conditions for the wind farm in terms of potential temperature structure are not the ones initially prescribed in the precursor, and should be recomputed.

We agree with the reviewer that the most physically-realistic simulation platform would be the one that is coupled with mesoscale models (such as WRF), but this is out of scope for the two objectives that have driven the initial development of TOSCA, i.e. blockage and cluster wakes. Nevertheless, it is not excluded that we will add such feature in the future, when interested in simulating wind farms in real-life conditions also from the mesoscale point of view. On the same note, we would also like to highlight that the numerical implementation that is necessary to drive LES simulation with realistic mesoscale information is essentially the same that we propose, with the only difference that the average reference state changes in time (Allaerts at al. 2020, Allaerts at al. 2023). Despite this, the message that we want to convey is that it may be useful, when studying the ABL response to the wind farm in terms of gravity waves and wakes, to impose a certain wind speed and a very precise average thermal stratification (imposing a given roughness height is trivial), as these are the parameters upon which such response mainly depends. Having certain parameters such as the hub-height wind speed, BL height and inversion strength varying between codes, given the same initial condition, introduces uncertainties which we argue could be eliminated by adopting the proposed controlling methodology. Finally, within the paper, we mainly talk about realistic inflow conditions, or realistic turbulent inflow, which refer to prescribing an inflow that is realistically turbulent as it results from a prior simulation.

Regarding the reviewer's conclusive comment, we believe that our CNBL simulations are as idealized as the majority of CNBL precursors available in literature. They feature a realistic velocity and temperature profile, as shown in the paper, with the addition that the proposed controlling techniques allow to have direct control on the variables of interest. We also believe that trying to study blockage effects or wake effects in varying mesoscale pressure gradients, BL height or thermal stratification would provide an inconclusive picture of the underling physics, as too many variables would be changing at the same time.

REV: Abstract, line 9 – It is stated that "…prescribe a desired average hub-height wind speed while avoiding inertial oscillations above the atmospheric boundary layer (ABL) caused by the Coriolis force, a known problem in wind farm LES studies." However, inertial oscillations are not necessarily a problem: they are a component of a realistic atmospheric boundary layer flow in near-neutral and stably stratified conditions (e.g., Baas et al. 2011, QJRMS). Spurious inertial oscillations that are part of model-spin-up are a problem, but realistic inertial oscillations should be included in simulations that seek to match the real world.

AUT: We agree with the reviewer that inertial oscillations of the atmospheric boundary layer are physically realistic and important. Nevertheless, when one desires to impose a given hub-height wind speed instead of a geostrophic wind, the inertial oscillations that are triggered are spurious, as they arise because of a mismatch between the initial condition and the equilibrium solution (which is initially unknown as explained in the paper). As a consequence, these oscillations associated with the initial spin-up of the model should be removed, as they would not be part of the solution if, for example, one forced the BL using what is referred to as the geostrophic controller in the paper. Moreover, again in the context of gravity waves, their vertical wavelength depends on the geostrophic wind magnitude. As a consequence, when trying to characterize their impact it is evident that having a non-stationary free atmosphere wind (an effect that is in the equations but it is spurious in this case) would probably lead to wrong conclusions if for example averages are not performed within one oscillation period (which is rather long and unfeasible computationally for a whole wind farm simulation).

REV: Abstract, line 14 – It is stated that "The proposed methodology overcomes these issues by ensuring that the same hub wind and thermal stratification are used in successor simulations, regardless of the adopted code or precursor run time." Based on this statement it would seem that the goal of this work is to make all codes produce the same results regardless of the numerics. Observations do not seem to be relevant. It is not clear that this is a scientifically valid goal.

AUT: Our goal is to propose a method that would facilitate comparisons across codes by ensuring that wind farms are simulated under consistent initial conditions in terms of thermal stratification, wind speed, and direction. We believe that this will help systematize computational investigations of large-scale wind farms. Moreover, our statement does not convey that such approach should be always used (for example it is not of interest when trying to couple LES and mesoscale models) but we deem it useful when trying to simulate the wind farm behavior under a specified thermal stratification and wind speed and direction, which many researchers do.

REV: Abstract, line 17 – It is stated that "periodic wind farm are also presented to assess TOSCA's ability to model large-scale wind farm flows accurately and with high parallel efficiency." Here, it is not clear how is accuracy or parallel efficiency assessed – no comparison to observations or assessment of computational performance is presented.

AUT: We asses the accuracy of the model by comparing our results against existing and well validated numerical codes. Moreover, the periodic wind farm validation case is also compared

against wind tunnel measurements. Finally, in the appendix we included TOSCA's strong scaling test results that indeed prove that the code can efficiently run at scale (576M cells with 32.000 cores).

**REV**: Line 27 – Lundquist et al. 2019 (Nature Energy) presented simulations of wind plant losses due to wind plant wakes.

**AUT**: Citation added, thanks.

**REV**: Line 35 – It should be pointed out that Smith 2010 presented a 2D linearized analysis and that Wu and Porte-Agel 2017, Allaerts and Meyers 2017, 2018, 2019 used incompressible codes with Boussinesq approximation, and Allaerts and Meyers use 25 km vertical domain size, which is not appropriate for incompressible simulations (c.f., Lilly 1996 Atmos. Res., Wood and Bushby 2016, JFM). By citing papers based only on incompressible simulations with an inappropriate domain, the authors present an inaccurate picture of the state of the science.

**AUT**: The references mentioned by the reviewer and provided in the paper are indeed pointing to the very conclusion that we try to convey, i.e. that thermal stratification has an important effect on wind farm blockage. From the date the manuscript was submitted, more papers appeared in this regard: Smith 2023: The wind farm pressure field, and Lanzilao and Meyers 2023: A parametric large-eddy simulation study of wind-farm blockage and gravity waves in conventionally neutral boundary layers.

Regarding the results from Smith, we highlight that they employ linear gravity wave theory, which also features the Boussinesq approximation. Several books (Nappo 2012, Lin 2007) or articles (Allaerts and Meyers 2019, Smith 2002, Smith 2006, Smith 2010) use linear theory to model free atmosphere gravity waves, where the latter are produced by vertical disturbances in the flow generated by terrain features or wind farms.

In in the context of the present work, one should pay attention that the vertical length scale of interest for the validity of the Boussinesq approximation is not the vertical domain size, but rather the scale height of the atmosphere, typically around 10 km. Regarding the references pointed out by the reviewer, Lilly 1996 demonstrates that the Boussinesq approximation does not strictly conserve mass, a general property of such approximation. Conversely, Wood and Bushby 2016 perform an analysis that addresses the predictive difference in the onset of convective instability between Boussinesq and compressible models. As such instabilities eventually produce a turbulent convective flow in the entire domain (they also cite the Rayleigh-Benard convection) in their case the whole domain length is of interest. In our study, we model a simple CNBL without moist, deep or shallow convection or cloud formation and precipitation. Here, stability is neutral up to the inversion height, where it is capped by a stable inversion layer. This prevents the BL from reaching the top of the domain, and it is followed by a stable lapse rate aloft, where the flow is laminar, referred to as the free atmosphere. Atmospheric gravity waves would develop in the free atmosphere, inducing disturbances in the flow variables that can be studied using the Boussinesq approximation, as done in almost every LES study published so far with e.g. the ETH code, SP-Wind, SOWFA, AMRWind, PALM and TOSCA. In particular, in order for the latter to hold, the following hypotheses should be verified (Lin 2007): (1) the vertical dimension of the fluid motion is much less than any scale height and (2) the motion-induced fluctuations in density and pressure do not exceed, in order of magnitude, the total static variations of these quantities. These hypotheses are verified, as wind farm gravity waves only slightly perturb the pre-existing equilibrium, inducing extremely small motions. These waves can become unstable and break into clear air turbulence, but

this is likely to happen only with variable lapse rates, when the local Richardson number becomes less than its critical value (Klemp and Lilly 1978). We use a constant linear lapse rate aloft.

Conversely, we believe that the vertical size of the domain is instead dictated by the vertical wavelength of gravity waves. In fact, it has been shown by several studies (Klemp and Lilly 1978, Allaerts and Meyers 2017, 2018 and Lanzilao and Meyers 2022) that this should be split between a physical domain region and a damping region, which damps out gravity waves before they reach the top boundary and reflect, contaminating the solution. This is achieved by a damping region that is greater than the dominant wavelength, calculated as 2piG/N, where G is the geostrophic wind and N the Brunt-Vaisala frequency (again assuming linear lapse rate). As this is usually of the order of kilometers (around 12 in our study), and it increases with decreasing lapse rate, it appears clear how vertical domain sizes of 25/30 km are easily achieved in literature (at least two wavelengths, one for the damping region and one for the physical region). If this is not satisfied, results may be contaminated by gravity waves reflections produced at the top boundary, or simply unable to resolve gravity waves due to the insufficient space. For example, in Wu and Porte-Agel 2017, the vertical domain size is set to 2.4 km. This is insufficient to vertically resolve the gravity waves, characterized by estimated vertical wavelengths of 10 km and 4.8 km for the weak and strong lapse rate that they analyzed respectively.

**REV**: Line 39 –Again, this literature review is inaccurate. The statement "wind energy predictions are made using low-cost but fast, often analytical, reduced-order models (Katic et al., 1987; Ainslie, 1988; Larsen, 1988; Anderson, 2009; Openwind; Bastankhah and Port.-Agel, 2014; Niayifar and Port.-Agel, 2016)," is incorrect. The Ainslie model predicts wind flow in wake of a turbine, Katic is for cluster efficiency, and Larsen is wake model, Anderson model is wake model, while Openwind is a commercial tool for wind plant design (and its name is deceptive, because it is not open), so these papers do not refer to wind energy prediction (i.e. wind energy resource assessment of forecasting). The sentence should be rewritten and properly focus either on wind turbine wake prediction or different references should be provided.

**AUT**: Rephrased to "In industry, annual energy captures are made using low-cost but fast, often analytical, reduced-order wake models" and updated references (removed Openwind and changed 1987 Katic and Jensen to 1983 Jensen, which more clearly refers to the top-hat wake model).

**REV**: Line 41 – The statement "While they [the models listed above] have been used effectively for hundreds of wind energy projects, the majority of these models currently struggle in accurately reproducing wind farm blockage and farm-farm wake interactions" is not supported by any references. Are these models really used for hundreds of wind energy projects? Perhaps Openwind, but others are research and not design tools. Vague and/or inaccurate comments should be avoided.

**AUT**: Wake models such as the Jensen or Bastankhah wake models are indeed industry standard in large companies such as RWE or Orsted. Other models are also very popular such as the EV model by Ainslie, later reformulated analytically by Larsen with some approximations (preferred in Openwind, which lately started to propose TurbOPark as its core model). We do not believe that such statement is inaccurate information. Nevertheless, we added the reference from Nygaard 2022, which clearly shows that the park model (i.e. the Jensen model) underestimates wake effects and additional physical phenomena have to be accounted for, such as a variable wake expansion coefficient to better capture the wind farm wake evolution. Regarding blockage effects, it is easy to show that those models completely ignore it, as they do not predict any velocity deficit upstream the wind turbine. Moreover, the work by Centurelli et al (2021) clearly shows that current analytical blockage parametrizations do not agree with LES, but instead underpredict blockage in some cases.

**REV**: Line 66 – The statement "do not implement the concurrent-precursor method, making it difficult to properly capture gravity waves effects and inlet/outlet reflections" is confusing, it is not clear what is meant with "properly capture… inlet/outlet reflections." Such reflections should be avoided not captured. Furthermore, such problems are not encountered when using compressible codes like the Weather Research and Forecasting (WRF) model (c.f. Mirocha et al. 2014 JRSE, Aitken et al. 2014 JRSE, Vanderwende et al. 2016 JAMES, Marjanovic et al. 2017 JRSE, Arthur et al. 2020 , Dubrowa et al. 2020 Wind En., Kale et al. 2022 Renew. En., Sanchez Gomez et al. 2023 WES, Bui et al. 2023 EGUsphere).

**AUT**:  That was a typo, thanks for pointing it out. Rephrased to "making it difficult to properly capture gravity waves effects at the same time avoiding inlet/outlet reflections". Regarding the references pointed out by the reviewer, it is not clear whether gravity waves could even be captured in those studies. Mirocha et al 2014 and Aitken et al. 2014 could not observe any gravity waves as the physical domain height (1 km) was too small to actually resolve them, so they would be damped immediately by the Rayleigh damping layer (0.7 km). In the paper of Vanderwende et al. 2016 it is difficult to say if gravity waves are present in the domain, as no large-scale pictures or data is shown. Marjanovic et al. 2017 is not relevant for the present study, as they simulated a precipitating event over the Balkans and their domain was extremely large (whole Europe and part of Asia). Arthur et al. 2020 and Bui et al. 2023 might have picked conditions where gravity waves were not strongly excited, upwind propagating or simply died out before reaching the domain boundaries, it is difficult to say as the authors do not mention anything about it. We prefer to avoid speculating in this sense. Kale et al. 2022 simulated isolated wind turbines, so the wind farm – induced boundary layer displacement, which is the triggering mechanism for gravity waves, is minimal and gravity waves are probably absent. In Sanchez Gomez et al. 2023, as for Wu and Porte-Agel 2017, the vertical domain size is not sufficient to resolve gravity waves. Moreover, the mentioned Froude number is a meaningful parameter for surface waves only, which only propagate within the inversion layer. Internal waves always propagate upstream and upwards, but in some cases they can reach the top Rayleigh damping before the inlet, so that they are damped before reflecting (Gabriele Centurelli on PALM, personal communication). Given the very small vertical domain size, this seems to be the case for Sanchez Gomez et al. 2023.

As a general remark, it is difficult to talk about the behavior of gravity waves in WRF without exact knowledge of how the domain nesting technique is implemented, on how the data is transferred from one domain to the next, or given the extremely high variability of the flow properties in such simulations featuring dynamic evolution of the background flow. We believe that WRF would face the same issues in a simulation where the background stratification is steady and the simulation is carried out for a sufficient amount of time. In fact, as pointed out by Klemp and Lilly 1978, wave reflectivity increases over time for a given steady background state. It may be that energy accumulation due to wave reflection might represent a smaller issue for those WRF simulations where the background flow is evolving. These issues are worth investigating, but we feel fall outside the scope of the present paper.

**REV**: Line 78 – It is stated that "TOSCA is designed to enable LES simulations of large finite wind farms, operating in realistic atmospheric conditions," however, for simulations of wind plants in realistic atmospheric conditions it is necessary to couple mesoscale and microscale simulations (c.f. Haupt et al. 2023, BAMS). This requirement should be acknowledged.

**AUT**: Rephrased to "TOSCA is designed to enable LES simulations of large finite wind farms under realistic turbulence inflow and thermal stratification". In the reminder of the paper, we always refer to realistic turbulent inflow conditions. See above for our response regarding the mesoscale to microscale coupling.

**REV**: Line 79 – It is not clear what is meant by "mid-fidelity actuator model."

**AUT**: Rephrased to "Wind turbines can be modeled using the actuator line […] and the actuator disk […] models".

**REV**: Line 86 - In simulations with the compressible WRF model by Sanchez Gomes et al. 2023, WES, there are not gravity waves.

**AUT**: See above. In particular, too small domain combined with Rayleigh damping layer could completely eliminate their presence.

**REV**: Line 88 – Horizontal grid stretching can be detrimental to resolving well boundary layer turbulence.

**AUT**: We agree with the reviewer regarding this aspect. The favorable agreement with the presented validation studies suggests that the current level of grid stretching is not too detrimental.

**REV**: Line 92 – Previously it was mentioned that CNBL will be simulated and not stably stratified ABL.

**AUT**: The CNBL is an idealized type of boundary layer characterized by neutral stratification below a strong inversion layer and a stable lapse rate aloft. This can be imposed as an initial condition (Sec. 4 and 5) or reached starting from an initial condition with fully stable stratification and uniform non turbulent wind. As turbulence develops, the temperature profile will depict a region where the mixing generates a neutral stratification, an inversion layer that caps such region and the initially prescribed stable lapse rate aloft (Sec. 3.2). At line 92 we do not mention that the ABL is stably stratified (which would require a heat flux), but rather that it is thermally stratified, conveying that a dynamic evolution is present. This is explained in detail in Sec 3.2.

**REV**: Line 98 – The sentence starting with "TOSCA is written…" reiterates what was already mentioned in the Introduction in line 75.

**AUT**: Eliminated.

**REV**: Line 146 – 155 - This is standard textbook curvilinear coordinate transformation that can be put in an appendix or a reference could be provided.

**AUT**: Moved to appendix.

**REV**: Line 230 – Here, 'k' should be under the filter overbar.

**AUT**: Corrected for better readability also in the following lines.

**REV**: Line 231 - Filtered value of a product is not equal to product of filtered values regardless of discretization, but only approximately equal.

**AUT**: If the mesh is uniform, the filtering operation on the face area vectors has no effect. Hence the equality holds exactly. If it is stretched, this is an approximation as stated.

**REV**: Equation 19 –line 237 – This is a standard dynamic model derivation and as such can be put in an appendix or a reference can be provided.

**AUT**: Moved to appendix.

**REV**: Line 264 – It is not clear what is meant by "uniform actuator disk (UAD) models." This is not standard terminology.

**AUT**: Rephrased to: "To represent wind turbines, different models have been implemented, namely the actuator line (AL), actuator disk (AD), and uniform actuator disk (UAD) models. Following Tossas Leonardi (2015), the first two models require detailed blade information (i.e. airfoils, twist and chord), while the UAD only requires turbine thrust coefficient and general rotor information such as diameter and hub height.

**REV**: Line 265 – Reference to actuator disk and actuator line models are provided, while they share elements it would be better to describe and reference them separately.

**AUT**: Referring to our actuator disk and actuator line models, their implementation is identical. The only difference consists in the fact that points physically rotate in the AL model, while they are stationary in the AD model. Secondly, in the point force calculation the AL uses c*dr as local area element (where c is the local chord and dr the line segment), while the AD uses c*dr*Nblades/Ntheta, where Nblades is the number of blades and Ntheta is the number of points in the azimuthal (tangential direction). This approach is exactly as in SOWFA. In the UAD, the force at each actuator point is only axial and it is evaluated by using the conventional turbine thrust formula, scaled by dA/Adisk, where dA is the actuator element area and Adisk is the turbine disk area. This is explained in the paper, and plenty of references are given. Our implementation follows what is done in previous literature studies, so we believe that describing the models in detail is not necessary.

**REV**: Line 267 - Not all ADs account for blade rotation.

**AUT**: The reviewer is correct. We rephrased the first paragraph so that it is clearer we are talking about the actuator models implemented in TOSCA. The paragraph now reads "To represent wind turbines, different models have been implemented. In TOSCA, they are referred to as the actuator line (AL), actuator disk (AD), and uniform actuator disk (UAD) models. Following […], the first two models require detailed blade information (i.e. airfoils, twist and chord), while the UAD only requires turbine thrust coefficient and general rotor information such as diameter and hub height […]. The idea behind actuator models is to represent the wind turbine as a distribution of points, each associated with a Lagrangian force. For the UAD model, the sum of forces from all points must be equal to the total wind turbine thrust, while the AL and AD models in TOSCA additionally include rotor torque, as they also model blade rotation."

**REV**: Line 286 – It is not clear what is meant by "blade loading is uniform:" is this uniform thrust (and torque) along the blade?

**AUT**: As mentioned earlier in the paper, the UAD model is different from the AL and AD models in which the blade info is accounted for. In the UAD model, only thrust coefficient is considered, such that the thrust is uniform along the blade and torque is null. If one would plot the radial distribution of blade loading, it would be constant, as the sampled velocity is the same for all points.

**REV**: Line 368 – References for NCAR-LES and Wire-LES should be provided.
**AUT**: References added. Thank you.

**REV**: Line 403 – A long review of precursor methodology is given but no reference is provided.

**AUT**: References added.

**REF**: Line 407 – The question is if the plant induced gravity waves are a numerical artifact or they are real.

**AUT**: We believe the observed gravity waves are consistent with what is published on the characteristics of gravity waves in literature.  Their amplitude is dependent on the disturbance and on stratification, and their scale (in terms of wavelength) is extremely large and also depends on stratification. All this has theoretical and mathematical support (Gill 1982, Nappo 2012, Lin 2007, Smith 1980, 2007, 2010, Allaerts and Meyers 2018, 2019, Lanzilao and Meyers 2022a,b,2023).

We included an addendum that clarifies such concept through a simple but insightful model.

**REV**: Line 434 – The simulations were conducted for 120000s, or more than 33 hours, it is very unrealistic for the atmosphere to be steady for 33 hours.

**AUT**: The spin up phase itself requires more time than what a CNBL would be observed in real life. The use of the proposed controllers allows to run such idealized flow (in terms of its temporal evolution) in truly statistically steady state conditions, while e.g. not fixing the average temperature profile would produce a physical, but unreal because too slow, evolution of the ABL height. This whole concept is explained in Sec 2.5.2.

**REV**: Line 442 – It is not clear what is meant by "real flow physics."

**AUT**: Rephrased to "Nevertheless, the proposed hybrid method, sketched in Fig. […], is very convenient as it allows to reduce the overall computational cost of the ABL spin-up phase, where wind farm-induced gravity waves are not yet present. In fact, this initial phase is run on a domain whose size is dictated by the current flow physics, rather than on quantities that will only become relevant at later simulation stages."

**REV**: Line 448 – It would be better to reference the journal publication instead of the conference preprint, i.e., Martinez et al. 2018.

**AUT**: Revised also according to the first reviewer comment.

**REV**: Line 458 – It would be appropriate to provide a cross-section of the grid so that one can see the shape of the grid.

**AUT**: We do not believe it would add much information. Details are provided to exactly reproduce the mesh for anyone that is interested.

**REV**: Line 460 – Smagorinsky coefficient for standard Smagorinsky model is between 0.18 and 0.23 (cf. Lilly 1966 NCAR Tech. Note).

**AUT**: Our objective is to compare with Martinez Tossas (2015), so we set up the simulation in such a way that the two are exactly comparable.

**REV**: Line 462 – It is not clear what is meant by 72 points in azimuthal direction.

**AUT**: This is the discretization along the direction that is tangential to the disk perimeter. This is an angular discretization, as its actual size changes with radius, that is why sometimes is referred to as the azimuthal direction, for example in the SOWFA source code.

**REV**: Line 478 – It is not clear why is nearest neighbor approach used, it should be easy to implement interpolation to the geometric center of the Gaussian function.

**AUT**: Yes, this will be added in the future. With the revised data from Martinez Tossas (2015) data are more closer with TOSCA, so it does not seem that the interpolation makes a big difference.

**REV**: Line 487 – Why are sinusoidal perturbations added, why not random?

**AUT**: Random perturbations do not satisfy continuity and would be killed by the Poisson iteration. We follow the same approach used in SOWFA, where the introduced oscillations are sinusoidal and divergence free.

**REV**: Line 488 – These are surprisingly large wavelength perturbations- can the authors show why is that the best option?

**AUT**: We do not understand the comment of the reviewer in relation to line 488.

**REV**: Line 541 – The tower is just represented by drag - not actual actuator which is supposed to represent the rotor.

**AUT**: Yes this is the case. It is exactly the same approach, with the only difference that lift coefficient is zero. It is exactly the same as the rotor points that are located at the blade root for AL models, they have usually a lift coefficient of zero.

**REV**: Table 3 – It should be mentioned how S2 case differs from G case.

**AUT**: It is, see line 575: "In particular, we compare case S2 from Allaerts and Meyers (2017) against results obtained from TOSCA using both pressure and temperature controllers at the same time (case PT), and pressure and geostrophic controllers with no temperature forcing (case P and G, respectively)."

**REV**: Line 594 – 72000s is 20 hours. Again, is this realistic to expect stationarity for this long?

**AUT**: See previous comment and Sec 2.5.1 and Sec 2.5.2, this concept is explained.

**REV**: Line 612 – It is stated that "predicts lower geostrophic wind…" Isn't that just due to mass (i.e., more precisely volume) conservation?

**AUT**: This is in our opinion a consequence. Dissipation produces more mixing, and the inversion height raises. Then, due to mass (volume) conservation, geostrophic wind increases a bit.

**REV**: Line 614 – This is not due to "higher amount of dissipation" but due to mass (i.e., volume) conservation.

**AUT**: See previous comment.

**REV**: Line 639 – The streamwise spacing is smaller than used in any recently developed wind plant. The authors should explain why such unrealistic spacing is used.

**AUT**: We agree that the spacing is small, but not excessively in our opinion. Smaller spacing (down to 3.3D can be found in the Lillgrund wind farm for some wind directions)

**REV**: Line 641 – Is the 7.312 K a step change, or over some height?

**AUT**: Added "The temperature profile is initialized according to RampanelliZardi (2004)". In Appendix A, where the whole set up is described.

**REV**: Line 646 – Low reflectivity may not be enough – reflected waves can be amplified.

**AUT**: We agree, but it depends when the reflectivity is calculated. Reflectivity should be always calculated on the latest available samples, as reflections usually build up during the simulation.

**REV**: Line 654 – The subgrid model plays a role in this, too, c.f. Mirocha et al. 2010 MWR, and Kirkil et al. 2012 MWR.

**AUT**: Yes, even though we did not observe such streakiness with lower values of equivalent roughness height.

**REV**: Line 671 – While the wind farm blockage may be amplified by gravity waves, it still exists in the absence of waves (Sanchez Gomez et al. 2023 WES)

**AUT**: We respectfully argue that reviewer's comment is only partially true, that in the absence of gravity waves, wind farm blockage reduces to the superposition of the pressure increase created by individual turbines. If thermal stratification is modelled, the domain should be designed such that gravity waves can be captured, which is not the case in Sanchez Gomez et al. 2023 WES. In the absence of stratification, streamlines will be displaced more, as they do not encounter any resistance from buoyancy. As a result, induced pressure gradients – and consequently blockage - will be smaller (favorable at the farm exit and unfavorable at the wind farm entrance). This is because mass imbalance produced by velocity decrease is mainly compensated by streamline upward displacement. Pressure response is absent, as neutral stratification does not generate any density anomaly when streamlines are displaced. All this can be proved mathematically, as shown in the attached addendum. As a result, we believe that, without any stratification, blockage reduces to the superposition of individual turbine pressure increase, which has nothing to do with stratification. When stable stratification is present in the free atmosphere, large scale pressure gradients will be always produced to some extent, as they are a consequence of overtopping density anomalies generated by streamline displacement.

In conclusion, we do state that there is an effect that is not related to gravity waves in the sentence at line 649 "The gravity wave footprint inside the ABL can be clearly noticed in the pressure field, together with the small-scale pressure increase in front of each rotor", but the dynamics is much more complex than simply asserting that blockage can be observed without waves.

**REV**: Line 677 – What is the relevance of "an infinitely wide wind farm" for real wind plants?

**AUT**: We removed such comparison. It is misleading, as Wu and Porte Agel 2017 used a vertical domain size that didn't allow to resolve gravity waves. On the other side, we believe that infinitely

wide wind farms are useful, as they allow to simplify the physics while still providing insights, enabling model developments that could then be further extended to fully general cases.

**REV**: Line 693 – It is stated that "We also described a new methodology for simulating finite-size wind farm…" Strictly speaking, spanwise infinite wind plant is not finite-size.

**AUT**: Maybe the reviewer did not notice that a spanwise-finite wind farm is simulated in Sec 5, and the proposed methodology refers to this type of simulations (although is also applicable to infinitely wide wind farms). There are also Figures that show the simulated wind farm in relation to the domain (Fig. 9 and 10).

**REV**: Line 699 – The sentence "The concurrent-precursor domain is usually bigger than required, as its size is determined by the successor domain that runs concurrently" is not clear, why is the domain bigger than required?

**AUT**: We have added an explanation at line 443 (in the respective section) "Nevertheless, the proposed hybrid method, sketched in Fig. […], is very convenient as it allows to reduce the overall computational cost of the ABL spin-up phase, where wind farm-induced gravity waves are not yet present. In fact, this initial phase is run on a domain whose size is dictated by the current flow physics, rather than on quantities that will only become relevant at later simulation stages."

**REV**: Line 702 – It is stated that "… TOSCA is able to capture wind farm gravity wave interactions and large scale blockage effects," however, it is not clear that if the gravity waves are realistic – no comparison with observations is provided, so the term "capture" cannot be justified. This work just reproduced some of the previous numerical results that have not been verified in comparison to observations.

**AUT**: We agree that our study lacks comparisons to observations, as it is difficult to observe gravity waves in real life given their small amplitudes compared with the boundary-layer turbulence and their massive spatial scale, and we are not aware of any project that was dedicated, with suitable equipment, to these kind of observations. As such, we concede that the word "capture" may be better replaced with "simulate".

Regarding the physicality of the simulated gravity waves, we argue that their existence is supported by the cited literature. If streamlines in a stably stratified flow like the free atmosphere are perturbed upwards by an object lying inside the BL, waves will be triggered. Their amplitude is dependent on the disturbance and on stratification, and their scale (in terms of wavelength) is extremely large and depends on stratification. All this has theoretical and mathematical support (Gill 1982, Nappo 2012, Lin 2007, Smith 1980, 2007, 2010, Allaerts and Meyers 2018, 2019, Lanzilao and Meyers 2022a,b,2023). Please refer to the addendum for the mathematical explanation.

**University of British Columbia**

**UBCO-UL NSERC Alliance Grant "Reduced-Order Models of Wind Farm Induction and Far-Field Wake Recovery"**

**Addendum to Reviewer 2**

Exec. S. Stipa - August 8, 2023

**1 Wind Farm and CNBL Stability**

In the present addendum, we propose a simple analytical model to explain the origin of global wind farm blockage. The model uses a perturbation analysis applied to the depth-averaged linearized Navier-Stokes equations, similarly to what previously done by Allaerts and Meyers (2019); Smith (1980, 2007, 2010). In particular, we assume an infinitely wide wind farm in the spanwise direction, so that quantities can only change along the streamwise direction, i.e. $\partial/\partial y = 0$. Furthermore, the background flow, is assumed to have null spanwise component $V = 0$ (this means that Coriolis forcing is neglected). The structure of the potential temperature profile is proper of a conventionally neutral boundary layer (CNBL). This is characterized by a neutral stratification until a strong inversion layer, located at $H$ is reached. For simplicity, this is modeled as a discontinuity in potential temperature, characterized by a jump $\Delta\theta$ located at $H$, followed by a stable lapse rate $\gamma$ aloft. The bulk velocity within the boundary layer and the geostrophic wind are referred to as $U$ and $U_g$ respectively. Similarly to Allaerts and Meyers (2019), the region below $H$ is divided into two layers, namely the wind farm layer, characterized by a height $H_1$, and the upper layer, of depth $H - H_1$. The depth of the wind farm layer is chosen as twice the hub height, i.e. $H_1 = 2h_{\mathrm{hub}}$. Finally, we assume that wind farm and upper layer are characterized by the same background velocity $U$, but at the same time admit different perturbation velocities $u_1$ and $u_2$. This leads to consistent simplification in the equations proposed by Allaerts and Meyers (2019), while still provides insight regarding the ongoing physics.

The three layer model equations developed by Allaerts and Meyers (2019), with the above simplifications becomes

$$\begin{cases} U\dfrac{\partial u_1}{\partial x} + \dfrac{1}{\rho}\dfrac{\partial p}{\partial x} = -\dfrac{C}{H_1}u_1 - \dfrac{f_x}{H_1} \\[2mm] U\dfrac{\partial \eta_1}{\partial x} + H_1\dfrac{\partial u_1}{\partial x} = 0 \end{cases} \tag{1}$$

and

$$\begin{cases} U\dfrac{\partial u_2}{\partial x} + \dfrac{1}{\rho}\dfrac{\partial p}{\partial x} = 0 \\[2mm] U\dfrac{\partial \eta_2}{\partial x} + H_2\dfrac{\partial u_2}{\partial x} = 0, \end{cases} \tag{2}$$

where $C = 2u^{*2}/U$. These equations can be rearranged by defining the new variable $\eta = \eta_1 + \eta_2$, i.e. the total vertical displacement of the pliant surface initially located at $H$. This, at steady state, coincides with the flow streamline through $H$ far upstream, and can be thought as the inversion layer displacement. By doing so, the system of equations becomes

$$\begin{cases} U\dfrac{\partial u_1}{\partial x} + \dfrac{1}{\rho}\dfrac{\partial p}{\partial x} = -\dfrac{C}{H_1}u_1 - \dfrac{f_x}{H_1} \\[2mm] U\dfrac{\partial u_2}{\partial x} + \dfrac{1}{\rho}\dfrac{\partial p}{\partial x} = 0 \\[2mm] U\dfrac{\partial \eta}{\partial x} + H_1\dfrac{\partial u_1}{\partial x} + H_2\dfrac{\partial u_2}{\partial x} = 0, \end{cases} \tag{3}$$

To complete the system, we add an extra equation that relates the vertical inversion displacement to the pressure anomaly that will be felt inside the boundary layer due to the increase or decrease in weight of the air column overtopping a given $x$ location. This can be expressed in Fourier space (Lin, 2007; Nappo, 2012) as

$$\frac{1}{\rho}\hat{p} = \Phi\hat{\eta} \tag{4}$$

where $\Phi$ accounts for pressure anomalies generated by both the inversion layer displacement (surface waves) and the resulting motion aloft (internal waves). We refer to Allaerts and Meyers (2019); Smith (2010) for the definition of such function. For the context of the present study it suffices to know that $\Phi$ depends on $U_g$, $\Delta\theta$ and $\gamma$. Specifically, when $\Delta\theta$ and $\gamma$ are null, i.e. no stratification, $\Phi \to 0$. This means that whatever is the vertical streamline displacement at $H$, this will generate no pressure response. Conversely, in very strong stratification any vertical motion is instantly suppressed by buoyancy, and $\Phi \to \infty$ (rigid lid case). Then, Eq. 4 becomes meaningless, and pressure disturbance is not linked to stability anymore.

In general, except from these two limiting cases, the resulting pressure produced by the system of equations 3 and 4 is the one that reconciles momentum conservation and anomalies of overtopping density through a determined vertical displacement of the pliant surface at $H$.

Considering a wind farm of length $L_f$, we calculate the wind farm force per unit length as

$$f_l = \frac{f_t}{L_f}\left(\frac{L_f \cdot 1}{S_x S_y}\right) \tag{5}$$

where the quantity between brackets is the turbine density per unit area and $f_t$ is the turbine thrust force, evaluated as

$$f_t = -\frac{1}{8}\pi U^2 C_T D^2 \tag{6}$$

The force $f_x$ is applied using the smoothed box function proposed by Inoue et al. (2014) for their fringe region.

[Figure]

(a) subcritical                (b) supercritical

(c) rigid lid                (d) fully neutral

**Figure 1.** Perturbation velocities, vertical displacement of the pliant surface located at $H$ and perturbation pressure for (a) subcritical case, (b) supercritical case, (c) rigid lid case and (d) fully neutral case. Vertical dashed lines indicate the wind farm start and end coordinates.

By transforming the system 3 in Fourier space, perturbation velocities, $\eta$ and $p$ can be calculated for different cases. In the remainder, we show results obtained with $H = 500$ m, $u^* = 0.45$ m/s, $h_{\text{hub}} = 90$ m, $U = 10$ m/s, $U_g = 12$ m/s (the latter enters in the computation of $Phi$, together with $\Delta\theta$ and $\gamma$ and the reference temperature $\theta_0$, which is set to 300 K), $C_T = 0.7$, $S_x = S_y = 10D$ and $D = 126$ m. The parameters that we wish to vary are $\Delta\theta$ and $\gamma$. In particular, we set them to very high values ($\Phi \to \infty$) to simulate the rigid lid case (RL), to zero ($\Phi \to 0$) for the fully neutral case (N), to 8 K and 1 K/km respectively for a sub-critical case in which interface waves within the inversion layer propagate upstream (SBC), and to 4 K and 1 K/km respectively for a super-critical case in which interface waves within the inversion layer cannot propagate upstream (SPC). Note that such property of interface waves depends on inversion strength and not lapse rate, as will be shown later.

Resulting streamwise profile of velocity perturbations in the wind farm and upper layer, $u_1$ and $u_2$, vertical displacement of the pliant surface $\eta$ and perturbation pressure $p$ are shown in Fig. 1 for each of the above cases.

In the subcritical case, interface waves in the inversion layer can travel upstream, as shown by the greater value of all perturbations in such region if compared to all other scenarios. Moreover, resonant lee waves are triggered by the wind farm (Allaerts and Meyers, 2019), and the decay of perturbations downstream is faster than what observed for the supercritical case, as also reported by Lanzilao and Meyers (2023). Regarding the limiting scenarios, the rigid-lid case shows that, if the inversion layer is kept in place, a decrease in velocity in the wind farm layer produced by the momentum extraction operated by the wind farm leads to an increase in velocity in the upper layer in order to conserve mass. This is counterbalanced by a negative pressure disturbance inside the farm so that momentum can be conserved in upper layer. In real life, if the lid is so shallow that two layers cannot exist, velocity is constant due to mass conservation, and wind farm momentum is totally balanced by the pressure gradient. In the fully neutral case, a pressure gradient is not needed, as mass imbalance produced by velocity decrease is entirely compensated by a streamline expansion. In real life, a small pressure gradient is still observed, produced locally around each with turbine according to momentum theory. This is neglected in the present model, as vertical pressure gradient is assumed to be zero within the boundary layer.

As can be seen, wind farm interaction with the stratified atmosphere cannot be fully understood without considering the atmosphere response to the applied wind farm thrust. These simple but yet insightful model, together with plenty previous research work, proves that wind farms do trigger gravity waves by initiating a momentum imbalance, if the free atmosphere is stratified. Additionally, surface waves are also produced by the presence of the inversion layer.

As a concluding remark, wind farm simulations — for example LES — which do not feature a sufficient vertical resolution of the free atmosphere, inevitably fail to capture the full evolution of gravity waves aloft, as there is not enough domain to actually resolve them. As a consequence, the simulation fails to correctly address the pressure response and pliant surface displacement, as these are contaminated by the poor resolution. Finally, even providing a sufficient domain height is not enough, as waves can reflect on the top or on the sides of the domain, again contaminating the solution. For this reason, we believe that any effort dedicated to better understand the modalities under which such reflections arise, and to define guidelines for the set up of such massive simulations, should be welcomed by the research community.

**References**

Allaerts, D. and Meyers, J.: Sensitivity and feedback of wind-farm-induced gravity waves, Journal of Fluid Mechanics, 862, 990–1028, https://doi.org/10.1017/jfm.2018.969, 2019.

Inoue, M., Matheou, G., and Teixeira, J.: LES of a Spatially Developing Atmospheric Boundary Layer: Application of a Fringe Method for the Stratocumulus to Shallow Cumulus Cloud Transition, Monthly Weather Review, 142, 3418 – 3424, https://doi.org/https://doi.org/10.1175/MWR-D-13-00400.1, 2014.

Lanzilao, L. and Meyers, J.: A parametric large-eddy simulation study of wind-farm blockage and gravity waves in conventionally neutral boundary layers, 2023.

Lin, Y.-L.: Mesoscale Dynamics, Cambridge University Press, 2007.

Nappo, C. J., ed.: Copyright, vol. 102 of *International Geophysics*, Academic Press, https://doi.org/https://doi.org/10.1016/B978-0-12-385223-6.00014-8, 2012.

Smith, R. B.: Linear theory of stratified hydrostatic flow past an isolated mountain, Tellus, 32, 348–364, https://doi.org/10.3402/tellusa.v32i4.10590, 1980.

Smith, R. B.: Interacting Mountain Waves and Boundary Layers, Journal of the Atmospheric Sciences, 64, 594 – 607, https://doi.org/https://doi.org/10.1175/JAS3836.1, 2007.

Smith, R. B.: Gravity wave effects on wind farm efficiency, Wind Energy, 13, 449–458, https://doi.org/https://doi.org/10.1002/we.366, 2010.

---

## Author Response (AR2)

**University of British Columbia**

**UBCO-UL NSERC Alliance Grant "Reduced-Order Models of Wind Farm Induction and Far-Field Wake Recovery"**

**Response to Reviewer 1**

Exec. S. Stipa - November 14, 2023

We would like to thank the reviewer for the time dedicated to revising the paper. We proceed with answering and clarifying, where possible, the proposed comments.

Our response, denoted in black, is shown below. Modified text of the paper is shown between quotes in *italic*, while the reviewers' comments are denoted in blue. Please refer to the track changes section at the end of this document for a detailed overview of the changes made to the manuscript.

I very much appreciate the very extensive responses that have been provided. As previously indicated, I strongly support the publication of the manuscript. Overall, this is an excellent and very important study for the community and certainly qualifies for publication in Wind Energy Science. Nevertheless, I feel that some of the answers should be better reflected in the updated manuscript, as indicated below.

While I agree that some of the suggested controllers are useful, as explained in the manuscript, I believe the statement that these are "better" (line 641) is too strong. While the temperature profile controller is useful in the context of reproducibility, it does affect physics as the way buoyancy is handled is changed. This aspect should be better reflected in the used formulation. In the abstract and conclusions, the reason for reproducibility should be better reflected instead of claiming it is simply better. As an aside, as the manuscript shows that results from different codes can vary from each other for different reasons, the proposed technique may not even assure full reproducibility between different codes.

The comment has been introduced in the manuscript. In particular abstract, result section and conclusions have been modified.

I thank the authors for updating Table 1. While the difference between TOSCA and OpenFOAM is much smaller than in the previous version, a difference of 6% is still significant, considering that a uniform inflow of cases is considered. The explanation below in Table 1 seems rather speculative, and I believe it should be replaced by a statement that the reason for the difference is unknown. Unless the authors could prove that their statement line 501 is indeed correct.

Updated accordingly.

Figure 7: There is no equation 36.

Thank you very much for pointing it out. The reference to Eq. 11 in Fig. 7 has now been updated after moving the LES section in the Appendix.

Figures 9 and 10 have a high-frequency oscillation in the spanwise direction. –> In the manuscript, no clarification has been made on this point. Some of the statements provided in answer to the referee could be elaborated, in particular, why these waves already seem to be present in the inflow, so these waves also travel upstream. Is the statement "free of spurious (i.e. non-physical) waves" based on a calculation, or it this interpretation of the simulation result?

An explanation has been added in the paper after figures mentioned by the reviewer. However, the phenomenon was already explained at the end of Sec. 2.4. In addition, $15,000$ s may not be sufficient to obtain perfectly clean average fields, and the artefact of inflow turbulence may still be observed. As a reference, we report below a comparison between two hub-height averages obtained from two similar cases where one has been run for $40,000$ s without a spanwise shift in the periodized inflow turbulence, while the second has been run for $20,000$ s with a spanwise shift of the inflow condition. As can be seen, while high-frequency oscillations can be detected in the first case, the second simulation depicts almost a perfectly uniform average velocity despite running for half the time. We do not believe this artefact poses any problem apart from yielding visualizations of lower quality.

[Figure]

**Figure 1.** (a) Hub-height wind speed for the simulation that ran for $40,000$ s without spanwise inflow shift; (b) Hub-height wind speed for the simulation that ran for $20,000$ s with spanwise inflow shift.

Line 677 " If averages are gathered for a sufficient amount of time, these streaks do not alter the simulation results, from the wind farm performance point of view" –> The statement is too strong as turbulent characteristics of streaks would affect coherence between different downstream rows. The present statement mostly refers to power production. Please formulate this more carefully.

Rephrased according to the reviewer comment.

Why is case S2 in Figure 6d not zero for $h/H \leq 0.8$ as is the case for case P?

The case S2, described in Allaerts and Meyers (2017), does not feature potential temperature control. As a consequence, the average $\theta$ inside the boundary layer increases over time due to the fact that turbulent mixing slowly "eats up" the free atmosphere. This is explained in the paper and it is exactly the reason why we suggest that potential temperature control should be used in these type of simulations. Case P, which also does not feature potential temperature control, is actually superimposed to case G on the plot and, for both, the higher numerical dissipation of TOSCA leads to faster increase of the average potential temperature inside the CNBL (these profiles are also took at a later time, as explained in the paper, so the difference is magnified). Finally, case PT is the only one that can reach a truly statistically steady state, as the mean potential temperature profile is maintained constant with the proposed controller. The reviewer is also referred to Fig. 3, where final potential temperature profiles – after their evolution – predicted by different codes are displayed. This test allows, among other things, to understand how dissipative a code is based on how much the ground temperature has increased over time. SP-Wind (the same code that

produced the S2 profile in Fig. 6) is much less dissipative than NCAR-LES and Wire-LES, despite them also being pseudo-spectral codes. Conversely, TOSCA's dissipation is comparable with the latter two even if it employs a finite-volume discretization.

**References**

Allaerts, D. and Meyers, J.: Boundary-layer development and gravity waves in conventionally neutral wind farms, Journal of Fluid Mechanics, 814, 95–130, https://doi.org/10.1017/jfm.2017.11, 2017.

**University of British Columbia**

**UBCO-UL NSERC Alliance Grant "Reduced-Order Models of Wind Farm Induction and Far-Field Wake Recovery"**

**Response to Reviewer 2**

Exec. S. Stipa - November 14, 2023

We would like to thank the reviewer for the time dedicated to revising the paper. We proceed with answering and clarifying, where possible, the proposed comments. Our response, denoted in black, is shown below. Modified text of the paper is shown between quotes in *italic*, while the reviewers' comments are denoted in blue. Please refer to the track changes section at the end of this document for a detailed overview of the changes made to the manuscript.

The authors have provided lengthy responses to my comments and suggestions in addition to an addendum presenting a simple analytical model to explain the origin of global wind farm blockage. However, the question is not about wind plant blockage, but about the reality of gravity wave effects induced by a wind plant. Gravity waves may be induced, but the question is if their effects as pronounced as in simulations that utilize incompressible solvers with Boussinesq approximation. The same question can be extended to the simple analytical mode presented in the addendum.

At the end of the addendum the authors state: "As a concluding remark, wind farm simulations—for example LES—which do not feature a sufficient vertical resolution of the free atmosphere, inevitably fail to capture the full evolution of gravity waves aloft, as there is not enough domain to actually resolve them." The sufficient height to resolve gravity waves implies that gravity waves are of significant vertical wavelength, i.e., a wavelength that is some large fraction of the depth of the troposphere. The question is if Boussinesq approximation can accurately capture such gravity waves.

We acknowledge that the reviewer remains skeptical about the use of an incompressible code employing the Boussinesq approximation to capture wind farm-induced gravity waves. We will try to allay this skepticism in the present answer.

The Boussinesq (1903) approximation was re-examined by Spiegel and Veronis (1960) for thermal convection problems. In particular, given a generic thermodynamic variable $f$, the authors introduce the following nomenclature

$$f(x,y,z,t) = f_m + f_0(z) + f'(x,y,z,t) \tag{1}$$

where $f_m$ is the constant space-average of $f$, $f_0$ is the variation in the absence of motion and $f'$ are the motion-induced fluctuations. Applying this nomenclature to the potential temperature field $\theta$ of a conventionally neutral atmosphere where a generic wind farm is immersed, we identify $\theta_m$ as the mean value of $\theta$ at the ground, $\theta_0(z)$ as the horizontally averaged vertical excess in temperature w.r.t. $\theta_m$, and $\theta'(x,y,z)$ as the spatial fluctuation in the mean temperature produced by the wind turbines. In the context of the finite wind farm simulations presented in the paper, $\theta_0(z)$ coincides with the horizontally-averaged potential temperature profile in the precursor simulation, while $\theta'(x,y,z)$ is the field obtained by removing $\theta_m$ and $\theta_0(z)$ from $\theta$ in the successor simulation after it has reached a statistically steady state. In terms of the time-averaged results, the precursor simulation does not induce any variation in $\theta_0(z)$ because the mean velocity field has a null vertical component. For this reason, the wind farm is the only cause of the perturbation field $\theta'(x,y,z)$, and equals zero if no turbines are present. (Here, our reasoning excludes the turbulent fluctuations, as we only focus our attention on steady-state spatial perturbations induced by gravity waves.)

We proceed to assessing that the Boussinesq approximation can be used with incompressible codes to study wind farm generated gravity waves, and even terrain-generated waves, which induce larger perturbations. To do so, we create a map on a $G$ vs $\gamma$ space, where $G$ is the geostrophic wind speed and $\gamma$ is the free-atmosphere lapse rate. In particular, for each point on the map, we evaluate the criteria introduced by Spiegel and Veronis (1960) and see if they hold. As more stringent criteria were outlined by Miesen et al. (1988), we will consider the Boussinesq approximation to be valid only if all criteria are verified at the

same time. Specifically, Spiegel and Veronis (1960) require that

$$\frac{\Delta\theta_0}{\theta_m} << 1 \tag{2}$$

where $\Delta\theta_0$ is the maximum variation in $\theta_0(z)$ across the domain height. Moreover, for non-linear problems where the flow departs from the initial condition (e.g., convection problems), Spiegel and Veronis (1960) require that

$$\alpha << \frac{\Delta\theta_0}{\theta_m} << 1 \rightarrow \frac{\alpha\theta_m}{\Delta\theta_0} << 1 \tag{3}$$

where $\alpha = \theta'/\theta_m$ is a measure of the wave amplitude. Eq. 3 states that motion-induced fluctuations cannot exceed, in order of magnitude, the static variations. In this context, we consider $\theta'(x, y, z)$ to be the maximum observed value of $|\theta(x, y, z) - \theta_m - \theta_0(z)|$, so that all perturbations are taken with the positive sign. Specifically referring to gravity wave problems, Miesen et al. (1988) further adds that

$$\frac{\lambda_z}{H} << 1 \tag{4}$$

where $\lambda_z = 2\pi G/\sqrt{g\gamma/\theta_m}$ is the vertical wavelength of gravity waves and $H = \theta_m\gamma^{-1}$ is the scale height of the problem (Miesen et al., 1988). Moreover, the same authors add a further requirement that the gravity waves' amplitude should satisfy for the approximation to hold, namely

$$\alpha << \frac{\lambda_z}{H} \rightarrow \frac{\alpha H}{\lambda_z} << 1. \tag{5}$$

As the value of $\alpha$ depends on the solution, for each $(G, \gamma)$ state we evaluate $\theta'$ using a simple 2D case corresponding to the uniform flow over a Gaussian hill that has a height equal to 250 m and half width equal to 500 m. The $\theta'$ corresponding to such a scenario is easily obtained using linear theory, and is well documented in Nappo (2012). The domain height is chosen as $2\lambda_z$, i.e. the minimum required to resolve gravity waves within numerical simulations, as one wavelength is allocated to the physical portion of the domain, while the second wavelength corresponds to the Rayleigh damping layer. The use of linear theory to estimate the maximum value of $\theta'$ is the only feasible option within the time constraints of the paper review process. We acknowledge that the linear theory itself employs the Boussinesq approximation, and so the solution may not be representative of the real physics when the validity bounds expressed by Eqs. 2 to 5 are approached. Nevertheless, when the left hand sides of Eqs. 2 to 5 are much less than unity, the Boussinesq approximation is justified and results from linear theory should be trustworthy.

Fig. 1 shows the left hand side of Eqs. 2 to 5 on a $G, \gamma$ space that is representative of atmospheric conditions for wind farm simulations. First, it can be noticed that conditions involving the perturbation amplitude $\alpha$ are critical at very low geostrophic wind speeds (low wavelengths), and mildly depend on the lapse rate, as both $\theta'$ and $\lambda_z$ decrease as $\gamma$ increases. Conversely, the left hand side expressed by Eqs. 2 and 4 increases with both $G$ and $\gamma$, with Spiegel and Veronis (1960) being more restrictive than Miesen et al. (1988).

[Figure]

**Figure 1.** Value of the left hand side of Eqs. 2 to 5 on the $G, \gamma$ space. (a) maximum value; (b) individual contributions associated with each condition. Dashed line indicates contour level 0.3.

As a consequence, the combined diagram shows that prohibited regions are towards high values of both $G$ and $\gamma$, or toward high values of $\gamma$ at low geostrophic speeds. Fig. 2 shows the gravity waves patterns for $G = 1, 10, 20$ m/s and $\gamma = 10$ K/km. While both the vertical and horizontal wavelengths increase with increasing geostrophic wind ($\lambda_z \approx 350, 3500$ and $7000$, respectively), the maximum perturbation amplitude is more constant, close to $\approx 2$ K. According to Fig. 1, the solution corresponding to $G = 1$ m/s lies outside of the validity bounds for the Boussinesq approximation, as the left-hand side of Eq. 5 is greater than unity. The case corresponding to $G = 20$ m/s could still be treated using the Boussinesq approximation, as the maximum value among the left hand sides of Eqs. 2 to 5 is around 0.4. Finally, the case corresponding to $G = 10$ m/s is well inside the validity bounds.

[Figure]

**Figure 2.** Gravity wave patterns for (a) $G = 1$ m/s; (b) $G = 10$ m/s; (c) $G = 20$ m/s; $\gamma = 10$ K/km in all cases. Horizontal coordinate normalized with the half width of the hill, $b$. Vertical coordinate normalized with $\lambda_z$.

Focusing now on the finite wind farm simulation presented in the paper, $G = 10.815$ m/s, $\gamma = 1$ K/km ($\lambda_z \approx 11.8$ km). As can be noticed, this lies in a region of Fig. 1a where the Boussinesq approximation is fully applicable, as the maximum among the left hand sides of Eqs. 2 to 5 is less than 0.1. We additionally note that the Boussinesq validity plots reported above are likely to be conservative if applied to wind

farms instead of terrain features. In Fig. 3, we report a comparison between the potential temperature perturbations observed in the finite wind farm simulation presented in the paper and around the hill.

[Figure]

**Figure 3.** Gravity wave patterns with $G = 10.815$ m/s and $\gamma = 1$ K/km obtained for (a) the hill using linear wave theory and (b) the wind farm using LES. The horizontal dashed line in the Fig. (b) indicates the height of the Rayleigh damping region, i.e. waves are damped moving above this line (the analytical solution does not requires vertical damping).

Only focusing on the free atmosphere, the maximum perturbation values of potential temperature are $\approx 0.09$ K for the wind farm and $\approx 0.25$ K (both negative) for the hill used in the previous part of this analysis (the wind farm perturbs the free atmosphere almost three times less). Finally, one last aspect to mention is that the wind farm simulation also features a capping inversion layer. Here, higher fluctuations in potential temperature are observed ($\approx 3$ K) but, since the inversion layer is extremely thin (100 m in our study), these do not represent a threat in violating the Boussinesq approximation.

In conclusion, we argue from the above analysis that gravity waves can be studied using incompressible codes following the Boussinesq approximation. Indeed, most of the $G$, $\gamma$ variable space can be covered by this approximation. Limiting conditions are represented by very low geostrophic winds (roughly below the cut-in wind speed for modern large wind turbines) and high geostrophic winds combined with a strong free atmosphere lapse rate.

As a last remark, the authors of this paper are aware that attempts of comparing WRF-LES (Weather Research and Forecasting Model in LES mode) results against incompressible codes such as TOSCA or SOWFA are underway at the University of Indiana (Matt Churchfield, Hrishikesh Sivanandan, Mehtab Khan, personal communication). In particular, WRF-LES has been tested on a hill case setup similar to the one presented in this answer, and similar gravity waves patterns were observed (preliminary results of this work have been recently presented as a poster at the NAWEA 2023 Conference in Boulder, Colorado). However, as WRF does not feature anti-reflection measures apart from the vertical Rayleigh damping layer, wave patterns were distorted by likely high wave reflectivity. Additionally, for those WRF studies that use nested grids, we argue that their use could impede gravity waves to propagate from inner to outer grids. Although this might not always occur, it should be considered that the relaxation regions used to drive the boundary conditions of inner grids from outer ones might act as a damping layers, distorting or damping these waves.

The authors have followed my suggestion and moved some of the background material about Numerical Procedure in Appendix A. They also made necessary corrections related to some of the minor comments. However, in a number of cases they did not consider my comments carefully and did not make any modifications to the manuscript. In some cases they misinterpreted my comments. For example, in my review I stated "Wind Energy Science journal may not be appropriate journal for publication of the work primarily focused on model development and idealized simulations." The key word is "may." Considering that the manuscript presents a new model and its validation, I do think that the work is worth publishing, the question is whether Wind Energy Science it the right journal and that decision rests with the editor.

On the other hand, the manuscript can be published "only if the numerous issues regarding inaccurate statements about the state of the science addressed and modeling choices better justified." The revised manuscript still does not meet this threshold.

The papers cited as justification that the manuscript devoted mainly to model development should be published in Wind Energy Science are all focused on wind turbine wake models and not on the large-eddy simulation models (i.e. simulation systems).

We agree that the decision about whether or not Wind Energy Science is the right journal for publishing the present work rests with the editor. We argue that TOSCA is more than merely a large eddy simulation model, but is a bespoke software specifically developed for wind energy applications, and which we believe will be useful for stimulating research within the wind research community. Moreover, it contains several novel features aimed at addressing research questions that have garnered significant attention in the past few years. We are making TOSCA available for the benefit of stimulating further research and scientific discussion within the community served by Wind Energy Science. Some of these beneficial features include

- A new open-source high-fidelity framework tailored to simulate large wind farms with realistic inflow conditions, i.e. with detailed turbulence resolution. The code can potentially simulate large-scale variation in hub-height wind magnitude and potential temperature structure as well. Advanced actuator models with pitch, yaw and angular velocity controller allow to realistically capture the wind farm response to the incoming wind and its interaction with the atmospheric flow.

- A new driving pressure gradient that allows to remove inertial oscillations that are generated from inconsistent initial conditions. There is no way, as explained in the paper, to choose a consistent set of initial conditions if the wind is controlled at the hub height. Hence, these oscillations, although physical in their nature, are just a spurious effect in this class of LES simulations.

- A new hybrid off-line/concurrent precursor method.

- A simulation of a finite-size wind farm (finite both in the spanwise and streamwise direction) under truly statistically steady-state CNBL. The CNBL profile includes wind veer and thermal stratification, while the wind turbine models include angular velocity, pitch and yaw controllers.

- An assessment of large scale blockage effects for a specific CNBL case, showcasing the capabilities of the developed open-source code.

We would submit that the level of dialogue that has been stimulated with this reviewer is evidence that the scientific debate surrounding the relevance of gravity waves to wind farms and the proper methodology for studying them remains important to the field. Tools that contribute to this debate – in spite of their necessary limitations – will be relevant to Wind Energy Science. Moreover, we counter the reviewer's argument that "idealized simulations" are less capable in advancing our understanding of physical processes occurring within and around wind farms. For example, wake models are fast, allowing a broader exploration of the variable space at the price of lower accuracy. LES, depending on the specific case, may offer a detailed picture that is capable of isolating the dominant physical processes. Weather models allow to model a broader physics covering a bigger spatial scale, sacrificing high resolution at the wind turbine scale. Each of these models has its strengths and weaknesses, but advancements in all, we believe, are worth publishing in this journal, especially if tightly related to the wind energy field, which we believe our paper to be.

The authors insist on keeping the statement that SOWFA is not "sufficiently parallel-efficient," should not be made without providing evidence that TOSCA is comparatively more "parallel-efficient," however, no comparison in performance between TOSCA and other models is provided. Comparison of the scaling plot presented in Appendix D, with the scaling plot presented in Min et al. (2022, arXiv:2210.00904v1), Figure

5, shows that on 4 Summit nodes (44 cores) CPU-based NekRS with needs about 7 s per time step for $512^3$ grid cell simulation and AMR-Wind needs about 5 s per time step, while TOSCA needs more than 20 s per time step for 600 x 600 x 200 grid cell simulation (1.8 times smaller number of cells) on 5 40-core nodes. Based on this comparison it would seem that TOSCA is more than factor of 3 to 4 slower than the NecRS and AMR-Wind. In their reply they state that "We do not want to include a scaling performance comparison in our paper, as a fair comparison would require the same HPC platform and case." If that is the case then statements, like the one quoted above, "sufficiently parallel-efficient," that is included in the revised manuscript cannot be supported without direct comparison.

The statement regarding SOWFA not being sufficiently parallel-efficient has been modified in this second revision of the paper. We instead state that OpenFOAM-6, which SOWFA is based on, is "*...a general-purpose set of libraries that are not specifically designed to efficiently run at scale.*"

Furthermore, we believe that the reviewer is conflating scalability and solver performance. Parallel efficiency is not a measure of absolute solver speed, but rather the relative gain in speed given a certain increase in the number of processors used for the calculation (strong scalability) or the relative loss of performance produced by an increase in job size with a constant number of cells per processor (weak scalability). As it is demonstrated in Appendix D, TOSCA is parallel-efficient, i.e. its performance does not degrade – up to a certain limit, like any other solver – when the number of used processors or the problem size increase.

Regarding differences in solver speed, these can attributed to differences in the numerical procedures. SOWFA is based on OpenFOAM, which is an unstructured solver. This inherently requires to perform more operations to form the discretized system of equations (mainly to access mesh information through the connectivity data structure), and requires complex processor communication at processor interfaces. Hence, while SOWFA certainly is a fast and flexible solver in general, it is inherently less efficient when running at scale, i.e. with thousands of processors, than codes specifically deigned to run on HPC architectures. In curvilinear codes like TOSCA or rectilinear codes like AMR Wind, mesh elements and quantities of interest are directly accessed indexing 3D arrays, so there is no need for connectivity, nor for send/receive operations that involve more than a few processors. Additionally, if wind energy applications are considered, unstructured meshes are not the optimal choice as the majority of the domain is usually represented by the atmosphere. This means that a structured grid provides a faster and more accurate solution (hence the NREL choice to use AMR Wind as their atmospheric solver). Moreover, as explained in the paper, in order to write output fields, OpenFOAM/SOWFA generate a number of directories equal to the number of processors. When dealing with thousands of cores, the number of produced files increases rapidly, and may quickly saturate the maximum file counts allowed on HPC systems.

Regarding AMR Wind and NekRS, the reviewer correctly mentioned that on an absolute level, TOSCA may be slower. However, the reviewer should consider that TOSCA uses an implicit matrix free momentum solution. This yields a higher time per iteration than explicit methods, but allows larger time steps with CFL values that can exceed 2. Hence, it is still not clear how much and if TOSCA is slower than AMR Wind and NekRS. Specific comparisons on the same system solving the same case would be required in this sense. Another aspect to mention is that AMR Wind and NekRS are specifically designed for speed and parallel scalability and, regarding AMR Wind, it is still not clear how flexible this code is. Moreover, TOSCA has features that the two solvers do not have, namely the concurrent precursor method, which is a unique feature among existing finite volume codes, and the immersed boundary method to model complex terrains. Despite being already present in the open-source repository and working both in dynamic (for fully resolved turbine cases) and static (terrains or objects) modes, the latter has not been included in the present paper but, as mentioned, will be the focus of a follow up study.

The authors state that "Our choice to simulate a non-evolving flow (mainly in terms of BL height) is explained in the paper, and we believe that this is not more idealized than running a precursor for big

amount of time without temperature control, as in reality CNBLs are only observed for short lapses of time, especially in transition between night and day." Perhaps this is a question of semantic, about what is meant by idealized. However, on the spectrum of idealized to realistic the simulations presented in the manuscript are leaning heavily to idealized. In the review of the original manuscript, I provided a number of references related to numerical simulations of wind turbines and wind plants under realistic atmospheric conditions. The issue is that the simulations of presented in the manuscript are idealized simulations and this should be clearly stated. I fully agree with the authors argument that "adopting the proposed controlling methodology" is beneficial for model comparison, however, this is still very different from simulating realistic atmospheric inflow conditions.

Fair enough. In this second revision, we modified the word "realistic" to "time-resolved".

The authors state that "prior to our study, resolving gravity waves in an LES of a wind farm has only been achieved by the group of Meyers." However, the question is what is meant by resolving gravity waves, are they resolved correctly – their wave length and magnitude, are the gravity waves real or a numerical artifact? The authors insistence in application of Boussinesq approximation to the domain that extends in vertical direction to more than 10 km is in conflict with assumptions approximation is based on. For example, Wood and Bushby (2016): "Specifically, the Boussinesq equations (e.g., Spiegel & Veronis 1960) are valid only for small perturbations to the thermodynamic variables in systems with small vertical length scales (in particular, the domain height must be much smaller than all of the thermodynamic scale heights)." Furthermore, Lilly (1996) states: "The principal defect of both systems is their inability to conveniently allow for arbitrary mean thermodynamic profiles. They both assume a reference state with neutral static stability. A non-neutral state can be imposed, but that represents a perturbation on the reference state which produces increasing error with layer depth." The authors use as a reference from which density perturbations are computed, in Equation 4 "$\theta_0$ is a reference potential temperature, chosen as the ground temperature," i.e., essentially a neutral reference state. Considering that this is the reference state, that the domain extends to 28 km in vertical, and that imposed stratification is 1 K/km this means that the perturbation through the troposphere and stratosphere the perturbation exceeds 10% of its reference value. Miesen et al. (1988, Physica Scripta) provides an analysis of the limits to application of the Boussinesq approximation for gravity waves in the atmosphere.

Again, the question is: can the magnitude and wavelength of gravity waves induced by a wind plant can be accurately captured with an incompressible solved based on Boussinesq approximation? In their response to my original comment about the use of Boussinesq approximation the authors provided a list of papers using this approximation. They state: "Regarding the results from Smith, we highlight that they employ linear gravity wave theory, which also features the Boussinesq approximation. Several books (Nappo 2012, Lin 2007) or articles (Allaerts and Meyers 2019, Smith 2002, Smith 2006, Smith 2010) use linear theory to model free atmosphere gravity waves, where the latter are produced by vertical disturbances in the flow generated by terrain features or wind farms." Furthermore, the authors state that: "In particular, in order for the latter to hold [Boussinesq approximation], the following hypotheses should be verified (Lin 2007): (1) the vertical dimension of the fluid motion is much less than any scale height and (2) the motion-induced fluctuations in density and pressure do not exceed, in order of magnitude, the total static variations of these quantities. These hypotheses are verified, as wind farm gravity waves only slightly perturb the pre-existing equilibrium, inducing extremely small motions." It is not clear where these hypotheses are verified, i.e. what is the ratio of vertical wave length to characteristic height, also since the perturbation in density is proportional to potential temperature perturbation and the reference temperature is ground temperature the "density" (i.e. temperature) perturbation is approximately 10% of the reference state. In the view of the analysis presented by Miesen et al. (1988) the question is if the Boussinesq approximation can be applied to this problem. The authors should show that the approximation is within the limits of the assumptions that the approximation is based on, or outside of its range of applicability. In particular in their simulations the ratio of density (i.e. temperature perturbation) to reference density (i.e. temperature) is 0.1 as indicated above and it is directly proportional to the relative amplitude of the gravity wave. At the same time the

ratio of vertical wave length to the characteristic height (in their reply the authors stat that "the scale height of the atmosphere [is], typically around 10 km) is likely larger than 0.1. However, the analysis by Miesen et al. (1988) shows that, for Boussinesq approximation to be applicable to analyses of atmospheric gravity waves, the relative magnitude of the gravity wave must be much smaller than the ratio of vertical wave length and characteristic height which in turns must be much less than one. In their reply the authors indicate that the vertical wavelength of gravity waves in their simulation is about 12 km "(around 12 in our study)" which means that the ratio of the vertical wave length to the characteristic length is larger than one.

Although this issue has already been addressed in our first answer, we apply the analysis proposed by Spiegel and Veronis (1960) specifically to our simulations to verify the appropriateness of the Boussinesq approximation by checking that both $\Delta\theta_0/\theta_m << 1$ and $\alpha << \Delta\theta_0/\theta_m$. The first inequality can be easily verified considering that the lapse rate is 1 K/km, and that the domain is 28 km height, yielding $\Delta\theta_0/\theta_m = 28/300 \approx 0.1 << 1$. The second inequality, considering that $\alpha = \theta'/\theta_m = 0.09/300$, where $\theta'$ are the maximum magnitude of the motion-induced fluctuations shown in Fig. 3, yields $3 \cdot 10^{-4} << \Delta\theta_0/\theta_m \approx 0.1$, which is also verified. Regarding the analysis proposed by Miesen et al. (1988), this requires $\lambda_z/H << 1$ and $\alpha << \lambda_z/H$, where $\lambda_z$ is the gravity wave vertical wavelength (definition is given in the first answer), $\approx 12$ km in our study, and $H$ is the problem scale height. In our previous answer we used the atmosphere scale height, which is around 10 km, to show that the motion-induced fluctuations in boundary layer displacement $\eta$ (on the order of 10 m near the capping inversion layer) are at least three orders of magnitude smaller, satisfying the condition that the motion happens in a layer whose thickness is much smaller than the atmospheric scale height. This is in fact another way of looking at the second hypothesis of Spiegel and Veronis (1960). Alternatively, in light of the analysis presented by Miesen et al. (1988), the problem scale height should be used, which is physically and numerically meaningful for the specific problem being simulated and input parameters chosen, i.e. $H = \theta_m/\gamma \approx 300$ km (this scale height is the one used for formal analysis by Miesen). Hence, $\lambda_z/H \approx 0.04 << 1$, and $\alpha = 3 \cdot 10^{-4} << \lambda_z/H \approx 0.04$ are both verified.

Finally, in relation to capturing gravity waves the authors state: "We believe that WRF would face the same issues in a simulation where the background stratification is steady and the simulation is carried out for a sufficient amount of time. In fact, as pointed out by Klemp and Lilly 1978, wave reflectivity increases over time for a given steady background state. It may be that energy accumulation due to wave reflection might represent a smaller issue for those WRF simulations where the background flow is evolving. These issues are worth investigating, but we feel fall outside the scope of the present paper." I agree that these issues fall outside the scope in terms of analysis, but they do not fall out of scope when the statements are made about the impacts that may not be realistic. Authors' statement points potentially to the essence of the question related to how accurately gravity waves are captured in simulations with incompressible solver and Boussinesq approximation. As pointed out in the review of the original manuscript the simulations with stationary conditions that extends for 20 hours is an idealized simulation and in authors' own words "It may be that energy accumulation due to wave reflection might represent a smaller issue for those WRF simulations where the background flow is evolving" while in their simulation it may result in unrealistic gravity waves.

While wind farm induced gravity waves are not a numerical artifact, as has been addressed in our first answer, it still remains to be seen what is their effect under evolving atmospheric conditions, mainly in terms of wind profile and potential temperature structure. We are currently leveraging TOSCA to simulate a full diurnal cycle using mean temperature and velocity forcing from ERA5 reanalysis data. The objective is to understand if and how results obtained from a temporal average of an evolving flow differ from results obtained by running a statistically steady state simulation forced by the same profiles averaged during the diurnal cycle. We re-emphasize that, although extremely important, such answer is not straightforward and requires dedicated analyses that fall outside of the present paper.

Regarding the WRF simulations mentioned by the authors which did not produce any gravity waves

(Sanchez Gomez et al., 2023) we note, as explained in our previous response, that these simulations are not properly setup to resolve gravity waves, as the domain height is less than $2\lambda_z$. The authors claim that they still observe global blockage without gravity waves. In fact, three forms of blockage are present around a wind farm:

1. Local blockage produced by individual turbine induction: this can be observed without modeling thermal stratification.

2. Global blockage produced by the flow confinement under the capping inversion layer, assuming that this is not perturbed (rigid lid approximation): this can be observed without modeling thermal stratification, i.e. placing the top boundary at the inversion height or with thermal stratification. In the latter case, if the domain is such that gravity waves are not resolved, the actual perturbation of the inversion layer will be incorrect, as it depends solely on gravity waves. Nevertheless, the large scale blockage might not be, in some cases, too far from the actual blockage observed when modeling gravity waves. Its perturbation though, i.e. beneficial/detrimental effects inside the wind farm, will be inaccurate.

3. Gravity wave blockage: this is produced by the vertical perturbation of the inversion layer w.r.t its equilibrium/freestream height. They are related to the physics of gravity waves aloft and interface waves within the inversion layer. The effect of this waves can be detrimental upstream and more or less beneficial inside the farm, hence it should be modeled.

If a simulation is conducted such that that it cannot capture gravity waves, only the first two contributions to the blockage will be observed, while the third becomes a result of the specific numerical setup. In fact, it may be affected by reflections from the side or top boundaries or, when using nested grids, gravity waves might be directly killed by relaxation regions outside of the innermost grid, where they are generated.

In their reply the authors state that: "On the same note, we would also like to highlight that the numerical implementation that is necessary to drive LES simulation with realistic mesoscale information is essentially the same that we propose, with the only difference that the average reference state changes in time (Allaerts at al. 2020, Allaerts at al. 2023)." They still ignore a significant recent body of work dealing with realistic mesoscale forcing through coupled mesoscale microscale simulations referenced in Haupt et al. 2023.

The referee correctly points out that, in addition to the "profile-assimilation" technique, there exists the "boundary-coupled" simulations in order to provide realistic mesoscale forcing. In the second revision, we have tried to improve the literature review such that it points out the importance of meso-micro scale coupling.

We are currently working on implementing profile-assimilation techniques within TOSCA. Here, instead of a uniform driving pressure gradient and a steady-state potential temperature profile, we would just have to render these quantities time-dependent, possibly including wind height dependency (temperature height dependency is already present). On the boundary-coupled side, TOSCA already features overset-mesh capabilities, so the structure to support nested domains already exists in TOSCA. Similarly to WRF, we use a relaxation region to drive the simulation results from the outer to inner grid. On the other hand, TOSCA does not currently feature a PBL scheme, required to run computations in the mesoscale domain with a grid spacing that falls above the *terra-incognita*. This is certainly a very interesting opportunity in the direction of extending TOSCA's capabilities, and we have to thank the reviewer for pointing it out.

There are a number of instances where the authors did not clarify the statements in the manuscript, but instead provided arguments that frequently are either incomplete or not accurate, for example, the authors claim that (line 231 in the original manuscript, line 823 in the revised manuscript) "If the mesh is uniform, the filtering operation on the face area vectors has no effect. Hence the equality holds exactly. If it is

stretched, this is an approximation as stated." It is not clear how this statement can be supported, filtered product is different than product of filtered quantities. For example, if a filter is idempotent, i.e., yields the same result if it is applied once or multiple times, then the equality in line 822 can hold only if the difference between unfiltered and filtered quantity is exactly zero. Clearly, if equality really holds and it is not just an approximation than this needs an explanation.

We agree to the referee's argument that filtered product is different than product of filtered quantities in general. However, in the case of uniform meshes, face area vectors are constant along each direction. For homogeneous LES filters (as in the case of TOSCA), the value of a filtered constant must be equal to the constant. For this reason, the filtering operation has no effect on the value of the filtered face area vector $\widetilde{\overline{S}}{}_j^k$. Applying this to the product $S_j^k S_{ij}$, we can write

$$\widetilde{\overline{S_j^k S_{ij}}} = \widetilde{\overline{S_j^k}}\,\widetilde{\overline{S}}ij = S_j^k \widetilde{\overline{S}}ij \tag{6}$$

where $\overline{S}{}_j^k$ is identically equal to $S_j^k$ since we cannot compute face area vectors at a resolution smaller than the mesh size. When the mesh is stretched, Eq. 6 does not hold exactly anymore, as face area vectors are now varying along curvilinear directions. Nevertheless, the tilde LES filter has a size of 3 mesh cells in each direction, hence $\overline{S}{}_j^k$ (face area vectors at the central cell) and $\widetilde{\overline{S}}{}_j^k$ (filtered face area vectors within the box) are almost identical provided that mesh grading is smooth enough. For instance, grid stretching over three cells is usually imperceptible, as rapid grid stretching introduces other sources of problems such as diffusion and numerical instabilities when using e.g. second-order centered schemes.

Another example is Marjanovic et al. (2017) reference that they consider of no relevance, however, they did not do a thorough literature search and missed Marjanovic et al. "Implementation of a generalized actuator line model for wind turbine parameterization in the Weather Research and Forecasting model" J. Renewable Sustainable Energy 9, 063308 (2017).

We still believe that the mentioned paper is irrelevant to establish whether or not gravity waves are observed in WRF. A single wind turbine does not produce a momentum deficit that is strong enough to be felt at the inversion height. Hence, as the capping layer is not displaced vertically, wind farm induced gravity waves will not be observed. Moreover, the following sentence mentioned in the paper is incorrect: "Forcing idealized LES with geostrophic wind speed and direction results in inertial oscillations which alter boundary layer characteristics over the first several hours of a simulation. For the TWICS cases (CBL and NBL), a spin-up LES is run for 15 h with the surface heat flux specified (20 and 0 $W/m^2$ for weakly convective and neutral, respectively) to allow the solution to come into balance with the geostrophic wind vector". How can the solution come into balance in 15 h within the free atmosphere if the inertial oscillation period – considering the latitude of Boulder – is roughly 37 h, and the geostrophic equations are the one of an un-damped oscillator? There is no physical term in the governing equations (apart from some additional damping term, like the one proposed in our paper) that can operate in the sense of damping these inertial oscillations. We clearly show in our paper that these oscillations are almost undamped and that they go on indefinitely for a classic precursor simulation setup.

The authors also state that: "Random perturbations do not satisfy continuity and would be killed by the Poisson iteration. We follow the same approach used in SOWFA, where the introduced oscillations are sinusoidal and divergence free." However, random solenoidal field can be generated.

That is correct, but we do not understand the relevance of the comment. Whatever type of initial perturbations can be used provided that they are solenoidal. In fact, these are just used to trigger turbulence in the precursor and their presence is lost after a few recycling time. In particular, the final ABL state does not –

and should not – depend on the type of used turbulence-triggering method.

Finally, based on Table 3 it is still not clear how cases G and S2 differ. This needs to be clearly stated. Based on Table 3 these two cases are identical, this can be confusing for a reader.

Thanks for pointing this out, the caption has been modified. The two cases have in fact the same set-up, but case S2 is from Allaerts and Meyers (2017). Specifically, the difference between these two cases consists in the code used to run them. Moreover, when comparing case S2 with the other cases (Figure 6 in the paper), it should be noted that data from case S2 have been averaged at an earlier stage of the simulation (we did not possess the time history, so we used their published data). For this reason, differences are not only due to the code, but also to the fact that the ABL has evolved less. This is exactly one of the messages that our paper tries to convey: when running idealized CNBLs used to feed wind farm successor simulations aiming at reaching a statistically steady state, results depend on the simulated time if both temperature and velocity are not controlled in the mean. Such drift is not physical and, since simulations are already idealized as the reviewer states, it should be avoided.

[revised manuscript text omitted]